# RNA-peptide interactions tune the ribozyme activity within coacervate microdroplet dispersions

Basusree Ghosh[1,8], Patrick M. McCall [1,2,3,4,5,8], Kristian Kyle Le Vay[6],
Archishman Ghosh [1,2,7], Lars Hubatsch [1], David T. Gonzales[1], Jan Brugués[1,2,3,4],
Hannes Mutschler [6] ✉ & T-Y. Dora Tang[1,4,7] ✉

Membrane-free complex coacervate microdroplets are compelling models for primitive compartmentalisation with the ability to form from biological molecules. However, understanding how molecular interactions can influence physicochemical properties and catalytic activity of membrane-free compartments is still in its infancy. This is important for defining the function of membrane-free compartments during the origin of life as well as in modern biology. Here, we use RNA-peptide coacervate microdroplets prepared with prebiotically relevant amino acids and a minimal hammerhead ribozyme. This is a model system to probe the relationship between coacervate composition, its properties and ribozyme activity. We show that ribozyme catalytic activity is inhibited within the coacervate compared to buffer solution, whilst variations in peptide sequence can modulate rates and yield of the ribozyme within the coacervate droplet by up to 15-fold. The apparent ribozyme rate constant is anti-correlated with its concentration and correlated to its diffusion coefficient within the coacervates. Our results provide a relationship between the physicochemical properties of the coacervate microenvironment and the catalytic activity of the ribozyme where membrane-free compartments could provide a selection pressure to drive molecular evolution on prebiotic earth.

Given that compartmentalisation is a key feature in modern biological systems, it is also pertinent to consider its role in the origin of life. In both instances, compartmentalisation can play a critical role in organising molecules and reactions[1,2]. Membrane-free compartmentalisation driven by liquid-liquid phase separation brings together molecules from a molecular environment based on multivalent interactions and provides chemically distinct reaction hubs that combine molecular cooperativity with compartmentalisation[1,3]. However, it is still not clear how molecular interactions can affect emergent physicochemical properties such as the polymer volume fraction and molecular diffusivities within the compartment, nor how these properties impact localised enzyme reactions. Addressing this will help define the role of membrane-free compartmentalisation in regulating biochemistry during the origin of life and in modern biology, and enable rational engineering of compartment properties using bottom-up approaches.

Complex coacervation, the physical phenomenon of associative phase separation between two oppositely charged polyelectrolytes in

[1]Max Planck Institute of Molecular Cell Biology and Genetics, Dresden, Germany. [2]Max Planck Institute for the Physics of Complex Systems, Dresden, Germany. [3]Center for Systems Biology Dresden, Dresden, Germany. [4]Physics of Life, Cluster of Excellence, TU Dresden, Dresden, Germany. [5]Leibniz Institute of Polymer Research Dresden, Dresden, Germany. [6]Department of Chemistry and Chemical Biology, TU Dortmund University, Dortmund, Germany. [7]Department of Synthetic Biology, University of Saarland, Saarbrücken, Germany. [8]These authors contributed equally: Basusree Ghosh, Patrick M. McCall. ✉e-mail: hannes.mutschler@tu-dortmund.de; dora.tang@uni-saarland.de

solution, provides tractable models for membrane-free compartmentalisation for in-vitro studies. The formation of liquid droplets, as opposed to precipitates, is dependent on factors such as complexation and solvation properties[4–6]. In the 1920s, Alexander Oparin proposed that coacervation would have been important during the origin of life[7] to bring together molecules in the prebiotic soup. In addition, it has been shown that charge interactions that drive coacervation also contribute to liquid-liquid phase separation in biological systems in certain cases[8–10]. Furthermore, there are an increasing number of studies that show that coacervate microdroplets provide a bridge between the origin of cellular life and modern biological systems[11–13]. For example, coacervates can form from small prebiotically relevant molecules such as metabolites[14], oligopeptides[15], nucleotides[16], and fatty acids[17], and larger biological molecules, including intrinsically disordered proteins[18] and mRNA[19]. These coacervates have been shown to support a range of different reactions[20–22], from primitive RNA reactions[23,24] and autocatalytic networks[25] to enzymatic reactions[26], including polyketide synthesis[27] and transcription and translation[28]. However, deconvoluting the effect of coacervates on enzymatic activity is non-trivial, as coacervates have been shown to tune reactions in a variety of ways. For instance, a coacervates ability to support primitive RNA reactions can lead to the slowing down of cleavage rates[23], increased ligation yields[24], reduction in yields for template-directed polymerisation[29], and preferential ligation of linear RNA over circular RNA[30]. Furthermore, the ability for coacervates to be stable within a thermophoretic pore[31] and after wet-dry cycling[32] could have additional implications in tuning enzyme reactions under out-of-equilibrium conditions.

Given the impact of coacervates in tuning reaction kinetics where molecular interactions dictate droplet properties on the microscopic and macroscopic scales, there could be a possible effect of coacervates on the co-evolution of molecular species[33,34]. One interesting example to consider is the co-evolution of RNA and peptides, where direct interactions can facilitate the emergence of functional complexity of RNAs and/or peptides. Not only are RNA-peptide interactions prevalent in modern biological systems, their co-evolution could have been important during the origin of life to fuel the transition from chemistry to biology[35]. In the latter context, it has been shown that on the molecular level, proto-peptides stabilise folded RNA[33] and short peptides enhance ribozyme activity in homogeneous solution[36]. On the microscopic level, coacervates can induce the secondary structure of peptides[34] or alter the tertiary structure of RNA[37], which can affect ribozyme activity. Indeed, previous studies have shown that increasing the length of the polyanion, which forms the coacervate, will increase the fraction of product generated by ribozymes[29]. Furthermore, it has been shown that changing the peptide sequence from polyarginine to polylysine or (RGG)$_n$ - and modifying the length of the peptide can change the material properties of the droplet that correlates to a change in ribozyme activity[30,38,39]. Oligopeptide-based coacervates have additional advantages to homo (poly)peptide systems, as the properties of the different amino acids can impart different physical coacervate environments. For example, increasing hydrophobic moieties in peptides will increase the hydrophobicity of the internal environment and partitioning of hydrophobic molecules into the core[40], while increasing the strength of the interactions will affect the diffusion dynamics and material properties of the droplet[39]. Together, these provide test-tube experiments to demonstrate that RNA and peptide interactions can support the emergence of functional RNAs and the formation of membrane-free compartments that provide distinct biophysical environments that could support molecular evolution[33,34]. However, little is known about how random variations in the chemical sequence of coacervate-forming components, such as peptides, will tune the emergent physicochemical properties of the coacervate and how these together can impact reactions. This could also be relevant in extant biology where random sequence permutations in proteins play a central role in functional diversification.

In this work, we explore the role of molecular interactions via peptide composition on coacervate properties (physicochemical) and reaction kinetics. We used short, arbitrary cationic peptides of variable sequence (13 aa length) constructed from a reduced list of amino acids (12) and a minimal catalytically-active version of the hammerhead ribozyme (HH$_{min}$, Table S1). We show that coacervates prepared from ribozyme and each of the seven peptide sequences result in droplets with distinct chemical compositions i.e., different peptide/ribozyme concentration and stoichiometry. Further, we find differences in the physico-chemical properties of the coacervate microenvironments illustrated by variation in the condensed-phase polymer volume fraction and molecular diffusivity, as well as ribozyme reaction rate and product yield within the dispersion. Specifically, we observe a strong negative correlation between the polymer volume fraction in the coacervate and the ribozyme diffusion coefficient, as well as a positive correlation between the apparent rate constant of the ribozyme and its diffusion coefficient. RNase A footprinting assays show different binding modes of the peptide to the ribozyme depending on the peptide sequence. This suggests that the differences in interaction between RNA and each peptide sequence contribute to the observed differences in droplet properties and activities. We find that sequences with higher net charge lead to denser packing of the ribozyme and peptide and increased total polymer concentrations in the coacervate phase which was concomitant with slower ribozyme diffusion and slower reaction rates. It is important to note that the ribozyme catalytic activity was reduced by 2–3 orders of magnitude within the coacervate compared to free buffer.

Here, we exploit RNA/peptide coacervates as a versatile and robust model system for determining how amino acid sequence can change the physicochemical properties of the coacervate and how these subsequently impact reaction kinetics. Significantly, our study extends our understanding of the effect of compartmentalisation on enzymatic reactions by correlating coacervate properties to ribozyme activity. Furthermore, it marks a step towards unravelling how coacervates formed within a prebiotic soup can affect compartment properties and reaction outcomes across a population of primitive cells.

## Results

### Selection of RNA and peptide sequences for coacervate formation

To investigate the effect of peptide sequence on coacervate properties and ribozyme catalytic activity, we generated a small library of cationic peptides (7) (Supplementary Information Section 2.0; Fig. 1A, and Table S2) from 12 prebiotically plausible proteinaceous amino acids to provide models of prebiotically plausible peptides[41–48]. A random sequence generator script (Supplementary Information Section 2.1) was used to generate six sequences labelled as P-1, P-2, P-3, P-4, P-5, and P-6. The sequences include 50% positive charge (with R and K), 29% long chain aliphatic amino acids (I, V, and L), and 21% acidic, polar, and hydrophobic residues (D, E, S, T, G, A, and P). We also used a uniform positive charge sequence formed from alternating R and K (P-7) as a control peptide. For the negatively charged coacervate-forming component, we used trans-acting hammerhead ribozyme (HH$_{min}$) derived from satellite RNA of tobacco ringspot virus[49] (Fig. 1B). This 39-nt RNA is a modified version of trans-acting hammerhead ribozyme engineered to have a minimum structural fold for efficient substrate cleaving[49] and has been previously reported to be enzymatically active as a client molecule in preformed coacervates[23].

### Droplet formation with HH$_{min}$ and arbitrary cationic peptides

With the positively charged peptides and negatively charged RNA, we tested the ability of the arbitrary cationic peptides to form coacervate

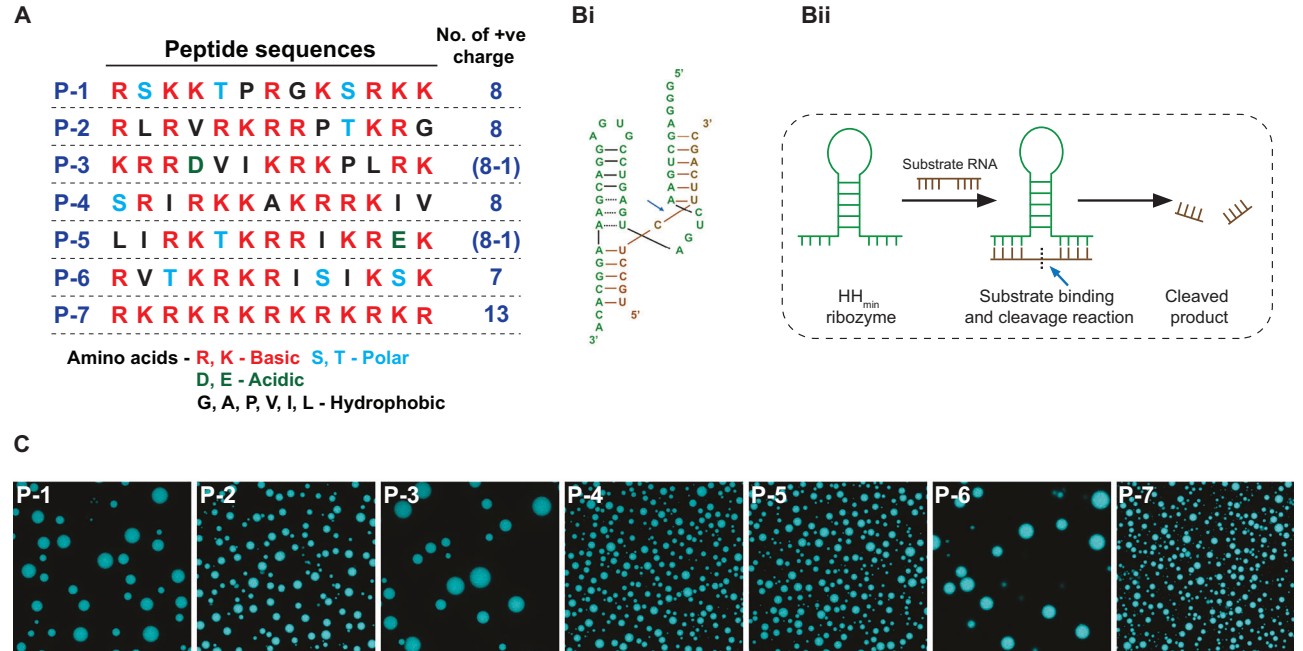

**Fig. 1 | Coacervate formation with peptide and Hammerhead ribozyme (HH$_{min}$).**
**A** Schematic representation showing the seven cationic 13-mer peptide sequences. Sequences labelled P-1 to P-6 contain similar numbers of positive charges, while the P-7 sequence consists of alternating arginines and lysines and is uniformly charged along its length. Colour code shows the chemical properties of the amino acids, red: basic, cyan: polar, green: acidic, and black: hydrophobic. **Bi** Schematic showing the 39-nt hammerhead ribozyme (HH$_{min}$) (green), used in this study, bound with its substrate RNA (brown) and the position of the cleavage site (blue arrow). **Bii** Cartoon representation showing HH$_{min}$ reaction with its substrate and the generation of cleaved products. **C** Representative confocal microscopy images of HH$_{min}$/peptide coacervate droplets prepared with 250 μM HH$_{min}$ mixed with 500 μM peptides in buffer (10 mM Tris, 1 mM MgCl$_2$, pH 8.1). 10% FAM-labelled HH$_{min}$ was mixed with unlabelled HH$_{min}$ to enable droplet imaging by fluorescence. Scale bar 20 μm. The experiment was repeated three times. Source data is provided in the repository.

microdroplets with HH$_{min}$. Polymer (RNA and peptide) concentrations were quoted in units of charge concentrations (molar concentration of polymer * number of charges per polymer) for all the experiments unless otherwise stated. Absorbance spectroscopy was used to observe any increase in turbidity with fixed HH$_{min}$ (500 μM) with increasing peptide concentration (0–5 mM) in 10 mM Tris-HCl, 1 mM MgCl$_2$ buffer at pH 8.1 (Fig. S1A). In all cases, the turbidity increased strongly with increasing peptide concentration towards a maximum near 1 mM, then either saturated or decreased slightly. To confirm that the observed turbidity results from droplet formation, we used confocal microscopy to image RNA-peptide mixtures at a 1:2 RNA: peptide charge ratio, corresponding to the approximate absorbance maximum. To visualise the droplets, HH$_{min}$ was doped with 10% FAM-tagged HH$_{min}$ to render the droplets fluorescent. Thus, 250 μM ribozyme (90% HH$_{min}$ + 10% FAM-HH) was mixed with 500 μM peptides in reaction buffer (10 mM Tris-HCl, 1 mM MgCl$_2$, pH 8.1). Confocal fluorescence imaging (Fig. 1C, for reproducibility, see Fig. S1C, D) revealed that each of the seven peptides formed spherical droplets containing HH$_{min}$ RNA. Given the number density of the droplets within the field of view, we chose to work with ribozyme and peptide at a 1:2 RNA: peptide charge ratio at concentrations of 250 μM ribozyme and 500 μM peptides for all further experiments (unless otherwise stated). For further analysis and discussion regarding the droplets (size and zeta potential (Fig. S1B)), please see Supplementary Information Section 3.2: Note 1. These results show that these randomly generated short 13-mer peptide sequences with at least 50% positive charge form coacervate droplets with HH$_{min}$ ribozyme at sub-millimolar concentrations.

### Peptide-RNA coacervates show variable ribozyme activity
Given that the randomly generated peptide sequences can form coacervates with HH$_{min}$ RNA, we next investigated the ability of the coacervate dispersion to support HH$_{min}$ activity. To do this, we used gel electrophoresis and measured the cleavage reaction of HH$_{min}$ RNA, with peptide (demixed coacervate dispersion) and without peptide (homogeneous buffer solution) (Figs. 2 and S2, S3, S4). We mixed HH$_{min}$ (250 μM) and peptide (500 μM) in buffer and left the sample to rest for 15 min before adding the FAM-tagged HH$_{min}$ substrate (12-nt RNA, Table S1) (50 μM) and gently mixing. The samples were incubated and centrifuged (~3000 × $g$) at specific time intervals to pellet the coacervate droplets into a continuous coacervate phase in equilibrium with a dilute phase or supernatant (see methods). Seven microliters of the supernatant was removed from 10 μL of the sample, leaving 3 μL of pellet composed of the complete coacervate phase in coexistence with some supernatant. The reaction was quenched and, subsequently, the pellet was characterised by gel electrophoresis (Fig. 2A). We observed substrate cleavage and product formation in each peptide coacervate system except in dispersions prepared with P-7, which showed no measurable product band intensity above the background, over 6 h (Figs. 2B and S3). By measuring substrate and product fluorescence band intensities, we calculated the relative product formation (%) of the remaining six systems over time (Fig. 2B, C and S3). We found that the product yield after 6 h varied widely between systems, from 4.6 ± 1.7 % in the P-2 system to 75.7 ± 11.8 % in the P-3 system (Fig. 2B). We also observed differences in the apparent rate constants obtained from fitting the kinetic profiles (Fig. 2B, see methods). Our results showed an approximately 15-fold difference in the slowest coacervate dispersion (P-2) with 0.0005 ± 0.0007 min$^{-1}$ compared to the fastest coacervate dispersion (P-3) at 0.0067 ± 0.0018 min$^{-1}$ (Figs. 2C and S3). Furthermore, when comparing the rate and product yield of HH$_{min}$ between the coacervate dispersions and free buffer solution (without peptides), we observe a significant reduction in the rate and the product yield in the presence of coacervates. The ribozyme is two to three orders of magnitude slower within the coacervate dispersion than the

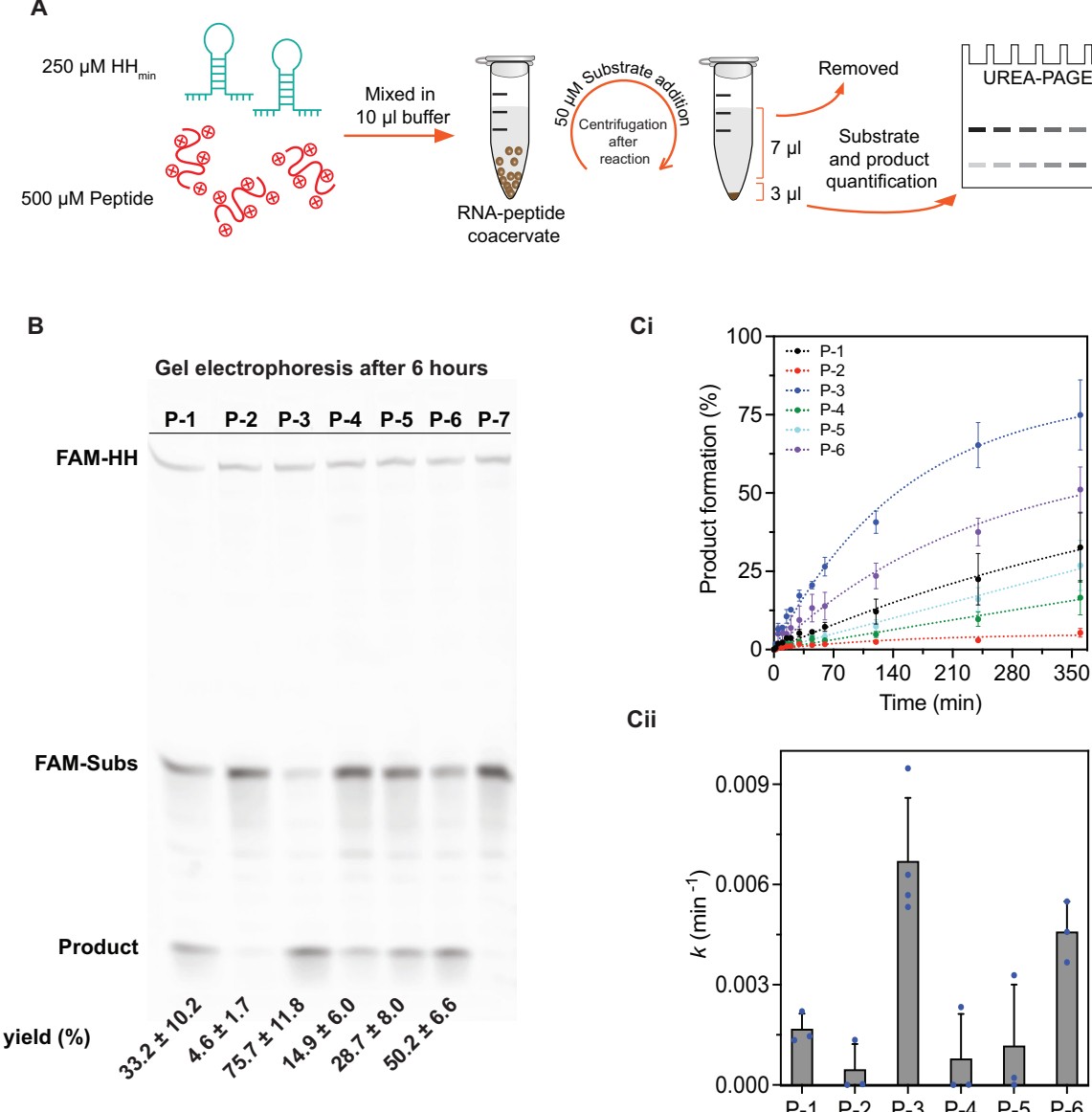

**Fig. 2 | Ribozyme activity within coacervates dispersions. A** Schematic showing the experimental design to measure $HH_{min}$ reaction kinetics inside coacervate dispersions. 250 μM $HH_{min}$ was mixed with 500 μM peptides to generate coacervate droplets. 50 μM FAM-tagged substrate was gently added to the dispersion. After each time point, the mixture was centrifuged, and 7 μL of the supernatant was removed. The pellet containing coacervate was then analysed by 20% urea-PAGE to visualise the cleaved and uncleaved FAM-substrate separated by gel electrophoresis. **B** Representative gel electrophoresis image showing comparative substrate and product band fluorescence intensities after 6 h. $HH_{min}$ was mixed with 10% FAM-tagged $HH_{min}$ to see the relative ribozyme concentration inside each system.

**Ci** Plots showing product formation (%) as a function of time for each of the $HH_{min}$/peptide coacervate systems. Circles are the data points, and the dotted line shows a representative first-order reaction kinetics model fitted to the averaged data points. Error bars indicate the standard deviation from three experimental replicates. **Cii** Apparent rate constant *(k)* comparison between six $HH_{min}$/peptide coacervate systems. The apparent rate constants were obtained from fitting the product formed as a function of time, using first-order reaction kinetics. The bar plot shows the average apparent rate constants from at least three independent experiments and the error bars show the standard deviation. Source data is provided in the repository.

buffer solution (0.5 ± 0.1 min⁻¹, Fig. S4). In addition, the maximum cleavage (92%) that is seen after 20 min in the buffer alone is not achieved within the coacervate dispersion after 6 h of incubation (Figs. S3 and S4). Our results show that the coacervate dispersion has a strong negative effect on the catalytic activity of the ribozyme compared to the free buffer solution. However, we find that variations in the peptide sequence led to large differences in the ribozyme activity—observed through product formation and the apparent rate constant—between the coacervate dispersions. In the case of uniform positive charge from the P-7 peptide, we observe no evidence of RNA activity within the resolution of the experiment.

To determine whether these observed effects could be attributed to the coacervate, we measured the content of the coacervate pellet and the supernatant after 6 h of incubation. In comparison to our previous experiments, we centrifuged and separated the sample immediately after the substrate was added to the coacervate dispersions. Gel electrophoresis was undertaken on 3 μL of the supernatant and 3 μL of the coacervate pellet and showed presence of product in both the coacervate pellet and within the supernatant (Fig. S5Ai). ImageJ 2.3.051 was used to obtain the band intensity of the product and to determine the ratio of product within the pellet versus the supernatant (ratio of product (pellet/supernatant)). Our results show

that in all the peptide systems, more product was produced in the pellet containing coacervate compared to the supernatant, with as much as 50-fold more in the P-1 pellet and as little as 5-fold more in the P-6 pellet (Fig. S5Aii). Given that the pellet contains the coacervate phase as well as residual supernatant (see Materials and Methods), the excess product in the pellet phase could be attributed to the product formation in the coacervate phase. Our results give a clear indication that the rate constant of the ribozyme is regulated by the coacervate dispersion. However, due to the effect of partitioning, it is not always possible to determine where the reaction is taking place. As coacervate droplets equilibrate molecules based on an intrinsic partition coefficient, our results could be attributed to (a) localised reaction within the coacervate and/or (b) RNA produced in the supernatant followed by partitioning into the coacervate. To assess the potential contribution of product formation in the dilute phase in isolated pellet fractions, we performed kinetic simulations of a hypothetical "Scenario B" that was constrained by experimental data (Supplementary Information Section 4.6: Note 2, Fig. S6, S7, S8, and S9 and Tables S3, S4, S5, and S6). In Scenario B, the reaction takes place only in the dilute phase with substrate and product partitioning rapidly between the dilute and coacervate to maintain its equilibrium partition coefficient as the reaction proceeds. The simulations predict substantially lower amounts of product than what we observe in our experiments (Fig. S6). Together, our experimental results and simulations suggest that product is produced inside and outside of the coacervate droplet with a significant contribution coming from ribozyme cleavage within the coacervate.

It is important to note that the rate constant ($k$) we measure is an apparent rate constant for hammerhead ribozyme activity which includes contributions from both dilute and coacervate phases. Due to the two different environments, there can be two different rates of the hammerhead ribozyme activity, and by extension, two rate constants, one outside ($k_{out}$) and one inside ($k_{in}$) of the coacervate phase. The apparent rate constant ($k$) that we measure is a linear combination of these two rates as well as all phenomenon such as diffusion outside, inside and across the phase boundary and the reaction rates inside, outside, and if relevant, at the interface. Due to small volumes of total coacervate phase it can be extremely challenging to determine the extent of the reaction within the coacervate droplet. Thus our measurements from the apparent rate constant obtained from 3 µL of pellet containing coacervate will differ from the rate constant ($k_{in}$) within the coacervate.

To develop an intuition for the relationship between $k$ and $k_{in}$, we undertook two further experiments. Firstly, we compared the rate constant within the coacervate dispersion to the rate constant within the pellet (Fig. S5B). In this case, both samples were treated equally until the end of the incubation period when the latter sample was centrifuged and the pellet separated whilst the former sample was left as a dispersion. Comparisons of the rate constant between the whole dispersion and the pellet shows that the rate constant is lower in the pellet compared to the dispersion for P-1, P-3, P-4, P-5 and P-6 systems (Fig. S5Bii). Secondly, we compared the product in the dilute phase to the pellet from the same dispersion (Fig. S5C), which was separated after incubation. We observed clear product bands in the dilute phase for the P-1, P-3, P-5 and P-6 systems only. Determination of the fraction of product ($F_{prod}^{dil}$) in the dilute phase was highest in the P-3 and P-6 systems, which contained 9.4 ± 2% and 13.2 ± 0.2% of the total product, respectively. We note that the relative percentage of product (~10%) in the supernatant is comparable to the variation in total product in the pellet (with P-3 and P-6) between experimental replicates.

The key observation from our results is that the measured apparent rate constant varies between coacervate dispersions depending on the peptide sequence. Our simulations (Scenario B) provide strong evidence that the majority of the reaction takes place within the coacervate and our experimental results show that the

pellet containing coacervate contains more product relative to the supernatant. We estimate that approximately 9–13% or less of the product is found in the supernatant. These observations could be attributed to the capability of the coacervate to localise RNA by partitioning whilst also supporting RNA cleavage. A high local concentration of substrate is available for cleavage within the coacervate droplet whilst product generated in both phases can, along with substrate, equilibrate through exchange across the interface.

## Coacervates co-localise ribozyme and substrate

We next used optical microscopy to directly observe the spatial localisation of the substrate and product within the coacervate droplets. To do this, we designed a FRET-paired substrate using FAM and Alexa 532 dyes that allows simultaneous imaging of the substrate and product (Figs. 3A and S10). The FRET-substrate (50 µM) was added to preformed ribozyme/peptide droplets and the substrate and product were imaged with confocal microscopy to visualise the uncleaved substrate ($\lambda_{em1} = 614–660$ nm) and the cleaved product ($\lambda_{em2} = 490–509$ nm) (Fig. 3Aii).

In the P-3 coacervate system, which had shown the fastest ribozyme kinetics by gel electrophoresis (Fig. 2C), confocal fluorescence microscopy reveals radial diffusion of the substrate into the droplet at $\lambda_{em1}$ with concomitant fluorescence increase from a cleaved product at $\lambda_{em2}$ over 60 min (Fig. 3B). To confirm that the fluorescence intensity at $\lambda_{em2}$ comes from the RNA product, we used a mutant HH ribozyme with two-point mutations (Mut-HH, Table S1), which can bind to the substrate but does not cleave it (Figure S2). Analysis of the fluorescence increase in the whole droplet as a function of time showed a similar increase in substrate fluorescence ($\lambda_{em1}$), within error, for the active (HH$_{min}$) and inactive (Mut-HH) coacervates (Fig. 3B–D). In comparison, the fluorescence increase of the product ($\lambda_{em2}$) within the coacervate droplets was negligible with the Mut-HH compared to the active ribozyme (Fig. 3B–D). This data confirms that the $\lambda_{em2}$ signal corresponds to product localised within coacervate droplets. Taken together, our results confirm that active ribozyme is driving the formation of product within the dispersions (Fig. 3B, C, Dii). Furthermore, simultaneous detection of the substrate and product within micronsized droplets are consistent with the results obtained from gel electrophoresis analysis and simulations which indicate that product within the coacervate droplet could be predominantly attributed to the activity of the compartmentalised ribozyme (Supplementary Information Section 4.6: Note 2).

We further applied this approach for the remaining coacervate systems (with P-1 - P-6), monitoring both substrate and product fluorescence within the coacervate droplet over 30 min (Fig. S11). In all cases, microscopy images showed a radial diffusion of the substrate into the droplet with the rate of diffusion varying between systems. Comparing P-3 (the fastest reacting coacervate) with P-2 (slowest reacting coacervate), we observe a radial gradient of fluorescent RNA in P-2 coacervates persistent up to 60 min that is not seen in P-3 coacervates. This is commensurate with slower diffusion of the substrate in P-2 coacervates compared to P-3 coacervates (Fig. 3B and S12). Our results show that substrate and product RNA are localised within the coacervate droplets and we confirm that $\lambda_{em2}$ fluorescence intensity is associated with the product and that product formation is driven by active ribozyme. Furthermore, between peptide systems, we observe differences in substrate transport within the droplets. Such variation in substrate diffusion between the peptide coacervate systems could be one of the contributing factors tuning reaction kinetics within the coacervate dispersions.

## Concentrations of ribozyme and peptide in the coacervate and dilute phases

Another potential source of variation in ribozyme activity between systems could arise from differences in ribozyme partitioning and

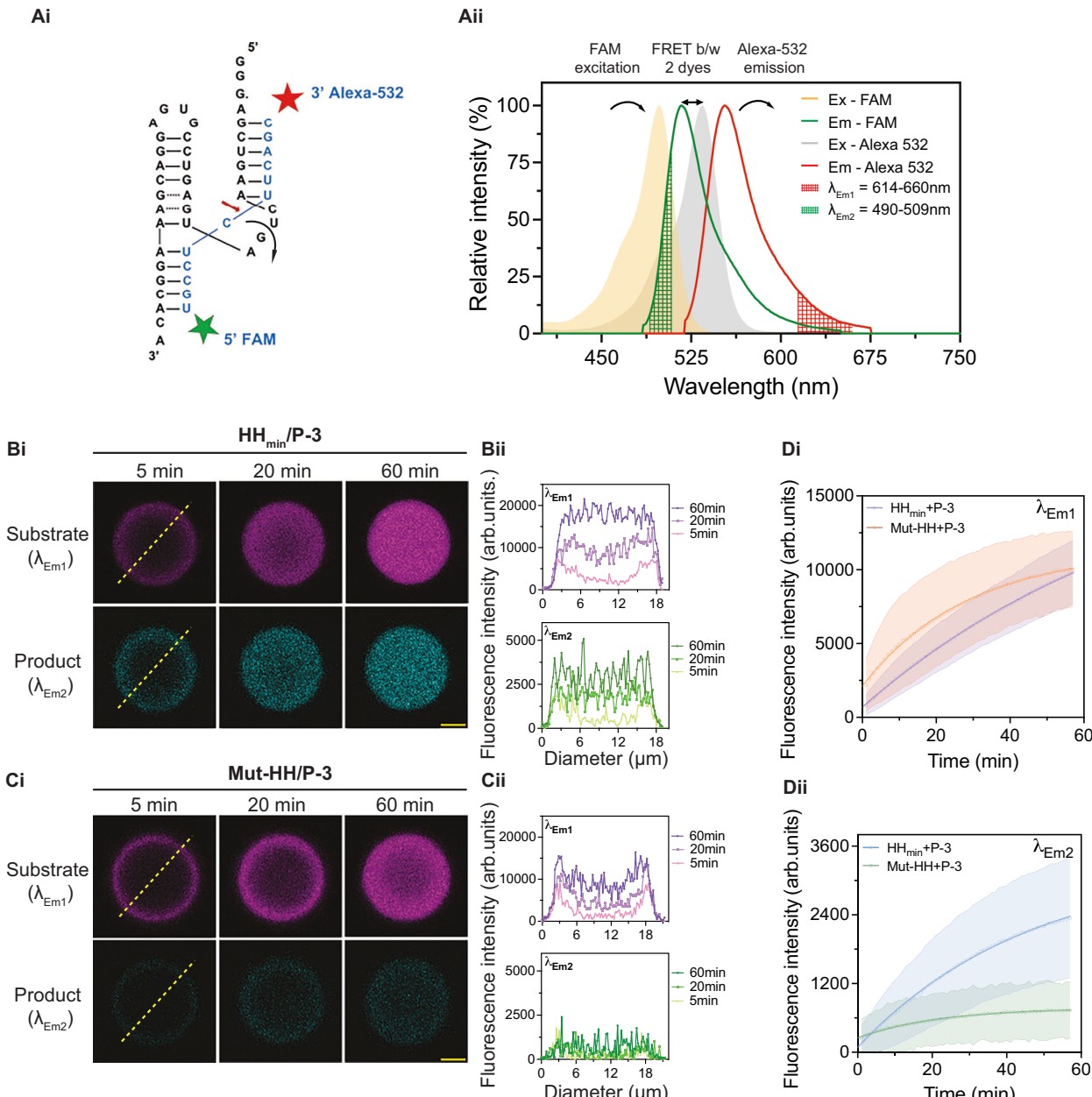

**Fig. 3 | Substrate and product localisation inside HH_min/peptide coacervate droplets. Ai** HH_min structure (in black) shows substrate (in blue) binding and its cleavage location (red arrow). 3′ Alexa 532 (red) and 5′ FAM (green) dyes indicate the position of the FRET pair on the substrate. **Aii** Merged excitation and emission spectra of FAM and Alexa 532 dyes used in the FRET substrate. Emission band filters used to detect the two dyes are $\lambda_{Em1} = 614$–660 nm for the substrate and $\lambda_{Em2} = 490$–509 nm for the product, marked as the red- and green-checked regions with fixed excitation at $\lambda = 488$ nm. **Bi** Confocal microscopy images show the localisation of the FRET-substrate (top: $\lambda_{Em1}$) and cleaved product (bottom: $\lambda_{Em2}$) inside a HH_min/P-3 coacervate droplet 5 min, 20 min and 60 min after the addition of substrate. Scale bar: 5 μm. **Bii** Line profiles showing the fluorescence distribution along the droplet diameter (yellow dotted line of **Bi**). **Ci** Confocal microscopy

images showing localisation of FRET-substrate (top: $\lambda_{Em1}$) and cleaved product (bottom: $\lambda_{Em2}$) inside a Mut-HH/P-3 coacervate droplet 5 min, 20 min and 60 min after the addition of substrate. Fluorescence intensity in the product channel was much weaker with Mut-HH relative to HH_min. Scale bar: 5 μm. **Cii** Line profiles showing the fluorescence distribution along the droplet diameter (yellow dotted line of **Ci**). **Di** Comparative fluorescence intensities of the substrate ($\lambda_{Em1}$) as a function of time obtained from analysis of whole droplet microscopy images of HH_min/P-3 and Mut-HH/P-3 taken under identical conditions. Curves show the average mean intensity of two droplets for each coacervate system and the shaded regions show the standard deviation. **Dii** Same comparison as above for the product ($\lambda_{Em2}$). The experiments were repeated at least three times. Source data is provided in the repository.

subsequent enzyme concentration within the coacervate phase. To test this, we measured the concentrations of HH_min RNA and peptide inside coacervate droplets. As the total condensed-phase volume is only a few nanoliters and thus challenging to quantify, we employed a recently-described approach based on thermodynamic and optical considerations[50] (Fig. 4A, Supplementary Information Section 5 and

Fig. S13A). This method uses tie-line analysis in combination with quantitative phase imaging (QPI) from which it is possible to determine the concentrations of HH_min and peptide within the coacervate phase. The advantage of this technique is that it utilises small amounts of sample and is not reliant on fluorescent methods, which could lead to artifacts[50]. This technique uses ternary phase diagrams which are

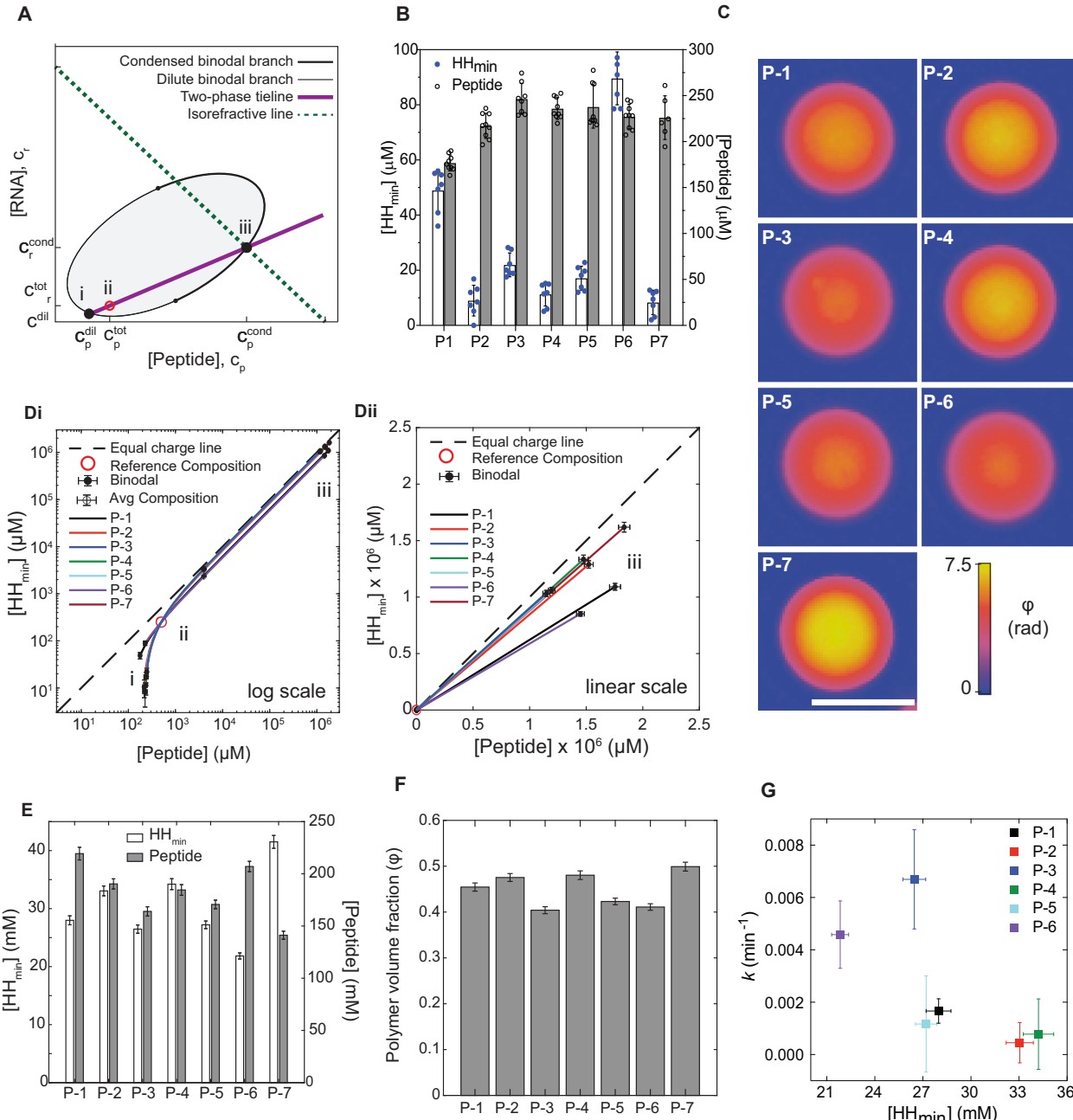

**Fig. 4 | Ribozyme and peptide concentrations inside the coacervate phase.**
**A** Schematic of a ternary phase diagram illustrating the principle used to determine coacervate-phase polymer concentrations. Purple: tie-line; green dotted line: iso-refractive line; (i) dilute-phase concentrations of peptide ($C_p^{dil}$) and $HH_{min}$ ($C_r^{dil}$); (ii) reference concentrations of peptide ($C_p^{tot}$) and $HH_{min}$ ($C_r^{tot}$), (iii) condensed-phase concentrations of peptide ($C_p^{cond}$) and $HH_{min}$ ($C_r^{cond}$). **B** $HH_{min}$/Peptide concentration measurements in the dilute phase. $HH_{min}$ concentration (white) measured in the dilute phase by fluorescence spectroscopy. Dilute phase concentration of peptides (grey bars) measured by BCA assay. Bar charts with error bars show the average and standard deviation from at least 3 repeats (see methods).
**C** Quantitative phase images of similarly-sized peptide/$HH_{min}$ coacervates. Colour bar shows the optical phase shift in radians. Scale bar: 5 µm. **D** Tie-lines for each of the peptide/$HH_{min}$ systems. **Di** Log-log plot showing (i) the dilute-phase binodal for each peptide system from (**B**); (ii) the reference concentrations of $HH_{min}$ (250 µM) and peptide (500 µM) (red circle) and the total concentrations of each system for

QPI measurements (open black symbols); and (iii) the condensed-phase binodal.
**Dii** Linear plot of the same data. Error bars on dilute and total compositions represent the standard deviation of repeat measurements. Error bars on coacervate composition represent error propagation via Jacobians (Supplementary Information Section 5.3.1). QPI measurements represent averages from $N = 161$ (P-1), 1718 (P-2), 1607 (P-3), 541 (P-4), 1355 (P-5), 1440 (P-6), and 1522 (P-7) individual droplets.
**E** Molar polymer concentrations of $HH_{min}$ (white) and peptide (grey) inside the coacervate phase. Error bars are the same as in D - rescaled by polymer net charge.
**F** Fraction of the coacervate phase volume occupied by polymers. Error bars represent propagation of uncertainty from dense-phase $HH_{min}$ and peptide concentrations in (**D**), rescaled by the partial specific volume of each polymer. **G** Plot showing the correlation between the apparent rate constant (*k*, Fig. 2Cii) and the condensed-phase concentration of $HH_{min}$ polymer (from **E**). The origin of the error bars have been described previously (Fig. 3) and (**D**). Source data is provided in the repository.

plots of the phase behaviour with increasing component concentration (Figs. 4A and S13). The transition between a one-phase region (homogeneous solution) to a two-phase region (coacervate plus dilute phase) is given by the binodal (Fig. S13). A tie-line connects the concentrations in the dilute phase (on the dilute binodal branch, Figs. 4A and S13) to the concentrations in the coexisting condensed phase (on the condensed binodal branch, Figs. 4A and S13). For all points along this line, the concentrations of peptide and ribozyme within coacervate phase and in the co-existing dilute phase are fixed (Fig. S13). Although the dense-phase composition lies on the tie-line, knowledge of the tie-line alone is insufficient to specify the precise location of the condensed binodal branch and thus the condensed-phase concentration. To determine where the tie-line meets the condensed binodal branch, quantitative phase imaging (QPI) can be employed to obtain the isorefractive line (Figs. 4A and S14[50]). QPI measures the optical phase shift between the droplet and the surrounding dilute phase and is sensitive to the refractive index difference between these two phases and the local droplet thickness[50,51] (Supplementary Information Section 5.6 and Figs. S13 and S14). Crucially, the refractive index difference ($\Delta n$) is dependent on the peptide and ribozyme concentrations in the condensed phase. Each pair of peptide and ribozyme concentrations that are consistent with the measured refractive index difference lie on an isorefractive line (Fig. 4A). The concentration of peptide and $HH_{min}$ in the coacervate phase is given by the intersection of the tie-line with the isorefractive line. Therefore, by simultaneously solving the isorefractive and tie-line equations, we can determine the condensed-phase concentrations of $HH_{min}$ and peptide for each of the coacervate systems[50].

Using this approach, we first determined the tie-line for each peptide/$HH_{min}$ system (Supplementary Information section 5.2, Fig. S14 and Table S7). The corresponding tie-line was determined from two points: the average concentration point (500 μM peptide, 250 μM $HH_{min}$), which lies in the two-phase region of the phase diagram (Fig. 4), and the dilute-phase composition point, which lies on the dilute binodal branch (Fig. 4 and Supplementary Information Section 5). The latter was determined by direct measurement of isolated dilute phase using spectroscopic methods (Supplementary Information Section 5.2.1). The dilute-phase concentrations differed for each peptide/$HH_{min}$ pair, varying from 175 μM to 250 μM of peptide in the dilute phase of the P-1 and P-3 systems, respectively, and 12 μM–90 μM of ribozyme in the dilute phase of the P-2 and P-6 systems, respectively (Fig. 4B). These differences in dilute-phase composition suggest that the extent of the two-phase region, and thus the strength of the thermodynamic driving force for coacervation, differs between the systems. For further analysis and discussion of the tie-lines and variation of phase behaviour between peptide sequences, please see Supplementary Information Section 5.4: Note 3 and Fig. S14).

Next, we determined the refractive index difference using QPI. To increase the accuracy of the measurement by QPI, we increased the average droplet number and size by preparing samples at higher total concentrations. To ensure that these larger droplets had the same thermodynamic properties (density) as those used for our previous experiments ($HH_{min}$: peptide, 250 μM: 500 μM), we used the measured tie-lines to determine the concentration of $HH_{min}$ which, when combined with 4 mM peptide, would produce coacervates of the same properties as those generated at 8-fold lower concentrations for the kinetic measurements (Supplementary Information Section 5.5: Note 4; Tables S7 and S8 and Fig. S15). Comparing the optical phase shift of different peptide droplets of similar size showed considerable variation between the different peptide systems (Fig. 4C), indicating substantial sequence-dependent variation in the refractive index difference between the droplet and the surrounding dilute phase (Fig. S16 and Table S9). From the QPI and tie-line measurements (Fig. 4d and Tables S10 and S11), we find that the molar concentration of ribozyme in the coacervate phase differs across the systems, ranging from

21.8 mM ± 0.5 for P-6 to 41.5 ± 1.1 mM for P-7. With these concentration measurements in each pair of coexisting phases, we were able to determine the partition coefficients of $HH_{min}$ and peptide (Fig. S17 and Supplementary Information Section 5.7). Incredibly, we find that the ribozyme is up-concentrated by as much as ~200,000-fold in the condensed phase over the dilute phase in the P-7 coacervate system. To compare the macromolecular content of these complex multi-component coacervates, we calculate the polymer volume fraction ($\phi_{poly}$) in the condensed phase for each system (Supplementary Information Section 5.8). We find that the condensed-phase polymer volume fractions in these systems vary from 0.4040 ± 0.0077 for P-3/$HH_{min}$ to 0.4991 ± 0.0097 for P-7/$HH_{min}$ (Fig. 4F and S18), with solvent and small ions occupying the remaining 50–60% of coacervate volume. Differences in measured $\phi_{poly}$ between pairs of systems are larger than the measurement uncertainty in most cases (Fig. 4F and Supplementary Information Section 5.3.1). We further note that, at such high concentrations, even modest changes in polymer content can strongly influence solution physical properties like viscosity[52].

Having calculated the concentration of $HH_{min}$ and peptide in the condensed phase, we sought to determine if the ribozyme concentration alone was enough to explain the variations in ribozyme activity within the dispersion. We plot the apparent rate constant, $k$, in the dispersion as a function of the molar concentration of $HH_{min}$ within the coacervates (Fig. 4G). Interestingly, we find that the apparent rate constant decreases with increasing ribozyme concentration in the coacervates. This contradicts the typical expectation from the law of mass action in dilute solution[53,54] where increasing enzyme concentration increases the reaction rate. This observed anti-correlation suggests that, despite the up-concentration of the ribozyme, the coacervate environment plays a dominant role in tuning enzyme reactions. For the peptide/$HH_{min}$ systems explored here, the variation in coacervate environment depends on the peptide chemistry. Our observations indicate that the coacervate composition can vary depending on differences in the peptide sequence, and this can modulate the overall activity of the ribozyme.

## Diffusion coefficient of ribozyme in coacervates

To explore how coacervate density, as measured by polymer volume fraction, influences molecular mobility, we next estimated the diffusion coefficient of $HH_{min}$ within coacervate droplets (formed from 250 μM $HH_{min}$ and 500 μM peptide in buffer). For this, we monitored mobility (in droplets 10-12 μm in diameter) using fluorescence recovery after photobleaching (FRAP) for each coacervate system (Methods, Supplementary Information Section 6.0). Recovery profiles showed partial recovery of $HH_{min}$ within 200 s for six peptides (P-1–P-6) and no detectable recovery with P-7, even after 10 min (Fig. 5A and S19). These results indicate that $HH_{min}$ mobility is strongly suppressed in P-7 coacervates compared to those formed from the other peptides.

Analysis of the recovery profiles with single-exponential fits gave diffusion coefficients of $HH_{min}$ between 0.00097 ± 0.0003 μm² s⁻¹ for P-2 coacervates and 0.006 ± 0.001 μm² s⁻¹ for P-3 coacervates (Fig. 5B and S19). We find a strong anti-correlation between the coacervate-phase $HH_{min}$ diffusion coefficient and the coacervate-phase polymer volume fraction (Fig. 5C). The slower diffusion we observed in denser coacervates is qualitatively consistent with two expectations. First, the diffusion coefficient is inversely proportional to the solution viscosity, in accordance with the Stokes-Einstein relation. Second, the viscosity of a polymer solution increases with polymer concentration[52].

Interestingly, we find a power law relationship between the reaction rate measured in the dispersion and the diffusion coefficient measured in the coacervate (Fig. 5D). This is consistent with the intuition for a diffusion-limited reaction, where a reduction in mobility necessarily slows the reaction rate. Taken together with this anti-correlation between local polymer volume fraction and diffusion

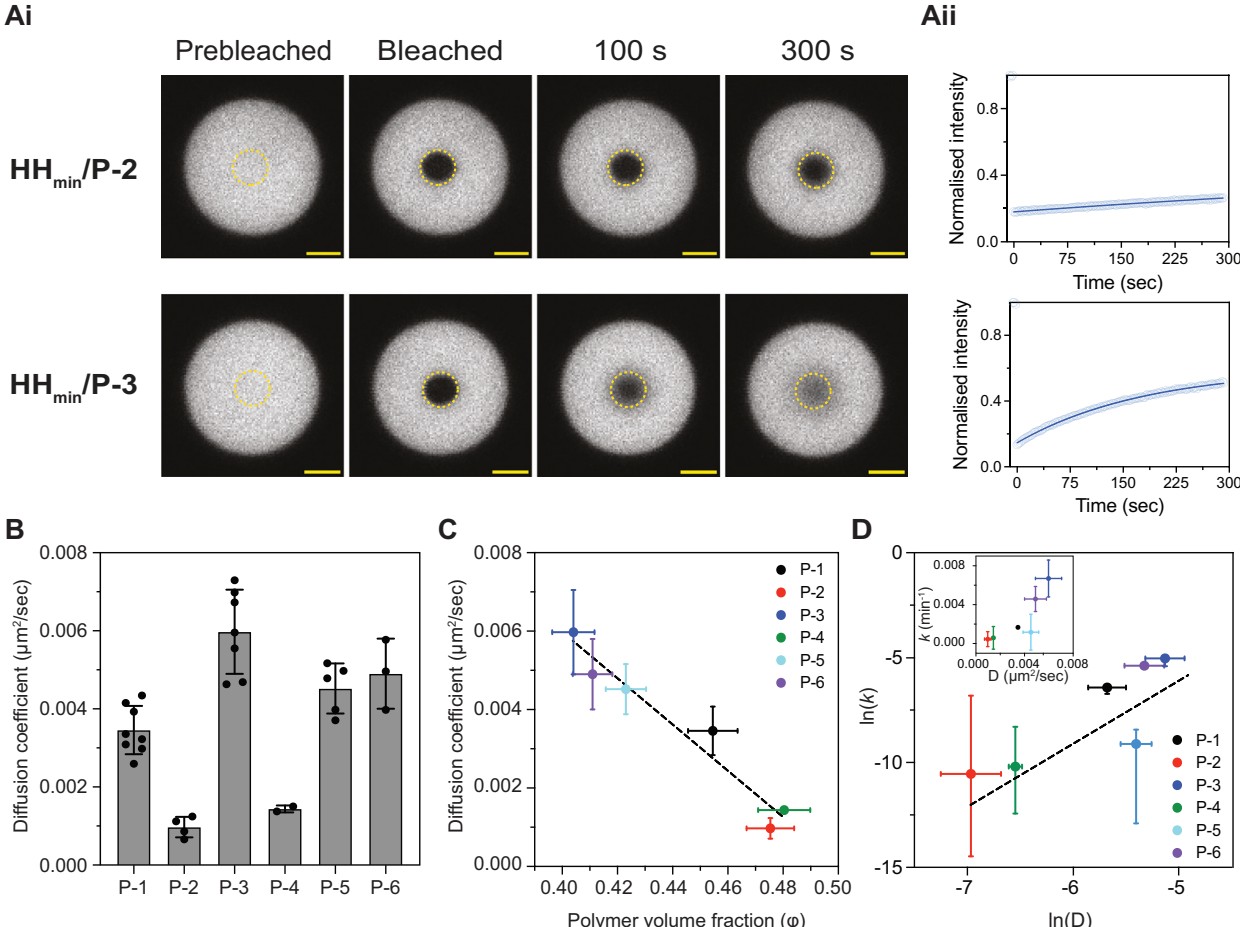

**Fig. 5 | Diffusion coefficients of ribozyme within coacervate droplets. Ai** Representative droplet images from FRAP measurement of $HH_{min}$ using confocal fluorescence microscopy. 10% FAM-tagged $HH_{min}$ was mixed with unlabelled $HH_{min}$ to measure the ribozyme diffusion coefficient inside the coacervates. FRAP images before and after bleaching for single coacervate droplets are shown for $HH_{min}$/P-2 (top) and $HH_{min}$/P-3 (bottom) coacervate systems. The yellow circle indicates the bleached region and the region from which the intensity of fluorescence recovery was measured. Scale bar: 5 μm. **Aii** Representative examples of normalised fluorescence intensity changes after bleaching obtained from microscopy images. **B** $HH_{min}$ diffusion coefficient obtained from FRAP for six $HH_{min}$/peptide systems (P-1 to P-6). Bar plots represent the average diffusion coefficient, and the error bars

represent the standard deviation obtained from 3 repeat experiments where multiple droplets were sampled per experiment. **C** Plot of $HH_{min}$ diffusion coefficient versus total polymer volume fraction ($\phi$) inside coacervates that reveals a linear relationship. Error bars for the diffusion coefficient are as in (**B**). Error bars for the total polymer volume fraction are as in Fig. 4F. **D** Plot of the natural log of the apparent rate constant ($k$) versus the diffusion coefficient ($D$). The dotted line shows a linear fit to the data indicating a linear relation between the two parameters. Inset shows the apparent rate constant versus the diffusion coefficient on linear scales. Average and error of $k$ and $D$ are as described in Figs. 2 and (**B**), respectively. Source data is provided in the repository.

coefficient, our data suggest increased molecular crowding or density can lead to slower molecular diffusion and also lower reaction rates, which is consistent with previous theoretical studies[55]. In conclusion, these results suggest that molecular mobility plays a crucial role in tuning reaction kinetics in the coacervate dispersion.

**Peptide-RNA interactions determined by an RNase A footprinting assay**

To determine whether the observed differences in the emergent properties of the coacervate systems correlated with variation in molecular level interactions, we sought to identify differences in molecular interactions between the peptide and RNA. To do this, we used a Ribonuclease A (RNase A) footprinting assay to identify the accessibility of specific areas of the ribozyme[56]. RNase A cleaves single-stranded RNA after pyrimidine nucleotides, i.e., cytosine (C) and uracil (U)[57]. Therefore, incubation of RNase A with ribozyme/peptide solutions can be used to identify RNA-RNA and RNA-peptide interaction sites that might hide or restrict access to particular parts of the ribozyme from RNase A with single nucleotide resolution (Fig. 6Ai). Folded

$HH_{min}$ in the buffer has nine cleavage sites where RNase A digestion can occur (Fig. 6Aii). To test the interactions of each peptide with $HH_{min}$, we used diluted $HH_{min}$ and peptide mixtures (25 μM $HH_{min}$: 50 μM peptide) in the buffer. We chose these lower concentrations of ribozyme and peptide to focus on the RNA-peptide interaction alone. Whilst the interactions within the coacervate droplet may not be exactly the same, it was critical to work at concentrations where droplets were not present but where RNA-peptide interactions are present within small clusters[58,59]. This was important to minimise any issues that could inhibit enzymatic activity, from the charged and crowded environment of the coacervate phase. This could include inhibition of the RNase A due to restricted motion, diffusion and/or accessibility to the reaction site, for example. This assay can provide invaluable insights into the RNA-peptide interactions that cannot be accessed within the droplet. FAM-HH/peptide solution was briefly treated with 100 nM RNase A (see methods) and the reaction mixture was characterised by gel electrophoresis to detect bands from the cleaved RNA fragments, from which the relative band intensity was determined (Figs. 6B and S20, Methods, Supplementary Information Section 7.0).

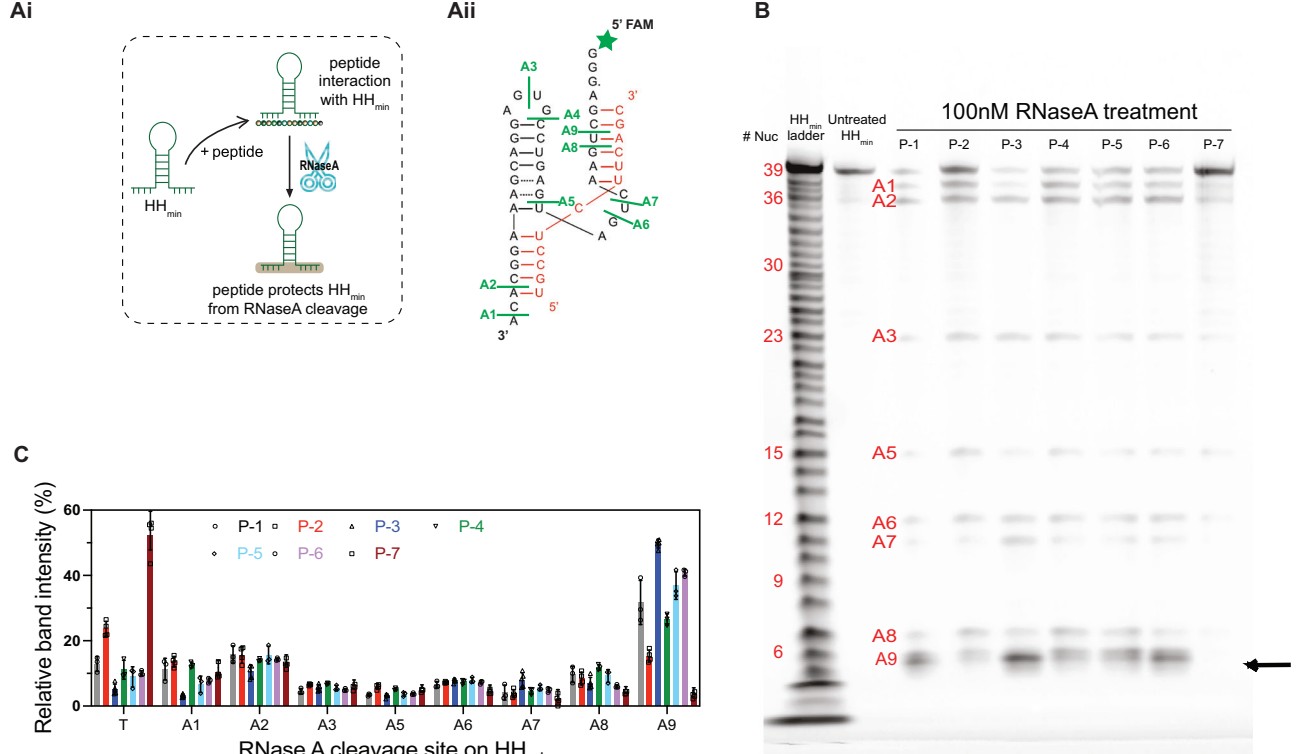

**Fig. 6 | RNase A footprinting assay to detect peptide-ribozyme binding sites without the substrate. Ai** Schematic showing hypothesised competitive binding between peptide and the substrate with $HH_{min}$. In $HH_{min}$/peptide complexes, if the peptide binds and blocks the substrate binding site, the site would be protected from RNase A cleavage. **Aii** Identification of RNase A cleavage sites on free $HH_{min}$ in the absence of substrate (substrate sequence marked in red). Green lines mark RNase A cleavage sites (A1-A9) starting from the 3′ site in the $HH_{min}$ unfolded region. **B** Fluorescence image of gel electrophoresis showing relative band intensities of RNase A cleavage sites on $HH_{min}$ bound with peptides ($HH_{min}$: peptide; 25 μM:50 μM). From left to right, the columns show the $HH_{min}$ ladder generated by alkaline hydrolysis of $HH_{min}$ with individual degraded nucleotides (37 of 39 were detected), untreated $HH_{min}$ and 100 nM RNase A treated $HH_{min}$ with each of the seven peptides. Cleaved sites are marked according to the nucleotide numbers and positions indicated in (**Aii**). RNase A cleavage is significantly different at the A9 position (marked with a black arrow) when compared across samples. **C** Bar chart showing the average relative band intensities for each of the RNase A cleavage sites (A1-A9 and T for uncut $HH_{min}$ 39-nt on $HH_{min}$ in the presence of each peptide from four replicates. Error bars show the standard deviation from four independent experiments. Source data is provided in the repository.

Analysis of the relative band intensities showed that solutions prepared with $HH_{min}$ and P-7 showed the maximum intensity from the uncleaved $HH_{min}$ (full-length: $T = 54.4 \pm 5.9\%$). In comparison, relative band intensities in the cleavage sites (A1-A9) were low (<18%). This could be attributed to strong interactions between P-7 and $HH_{min}$ that restrict RNase A access to the ribozyme or the inhibition of RNase A by the peptide.

Among the other six peptides, P-2 showed the least amount of $HH_{min}$ cleavage by RNase A ($T = 23.6 \pm 2.3$ %) compared to the other peptide/RNA complexes. Interestingly, the cleavage regions A3, A5, A6 and A7 in the folded region of $HH_{min}$ have a lower relative band intensity (%) compared to A9. The differences in band intensities between all six peptide systems are smaller in A3, A5, A6 and A7 compared to the variation between peptide systems at A9. The results indicate that RNase A may have restricted access to the folded region of the ribozyme in the presence of these peptides (Figure 6Aii). Together with our observation that RNA cleavage can take place in the coacervates, it suggests that the ribozyme may retain its secondary structure in the presence of peptide. Aside from P-7, we observed large band intensities at the A9 region (Figs. 6B, C and S20). The A9 region is the end fragment located on the $HH_{min}$−unfolded arm and, along with A2 and A8, serves as a substrate binding site of the ribozyme[23,49]. In the A9 region, band intensities lie between 3% for P7 and 50% for P-3. Furthermore, comparisons of the relative band intensities for peptides P-1 to P-6 showed maximal cleavage for $HH_{min}$/P-3 at A9, indicating that the overall RNA/peptide interaction is more dynamic for P-3 than for the other peptides. After $HH_{min}$/P-7, $HH_{min}$/P-2 showed minimum band intensity for the A9 region. This corresponds to a lower degree of cleavage of $HH_{min}$ by RNase A which suggests restricted accessibility of the ribozyme to RNase A through comparatively tight binding; a higher degree of RNA bound to peptide; or inhibition of RNase A by the peptide. It is interesting to note that, whilst the footprinting assay was undertaken in reduced concentration conditions, we are able to correlate the trends with our earlier results. For example, from the coacervates prepared with P-1 to P-6, the P-3 coacervates was the most active system with the fastest diffusion of the ribozyme and potentially more labile interactions at the substrate binding site (from RNase footprinting). In comparison, the P-2 system was the least active in this subset, with the slowest diffusion of the ribozyme and potentially stronger interactions at the substrate binding site.

Taken together, our results show that the accessibility of RNase A to the $HH_{min}$ ribozyme in the presence of peptide in homogeneous solution potentially correlates to the diffusive properties of the ribozyme within the droplet and its activity within the coacervate dispersion. Furthermore, variations in the strength of interaction between the RNA and peptide could potentially explain the differences in polymer volume fraction in the droplets, where stronger binding leads to better packing of the molecules and increased polymer volume fraction. This suggests that particularly strong $HH_{min}$/peptide interactions could contribute significantly to the markedly slow ribozyme mobility and catalytic activity we observe in the P-7 system, for example (Figs. S19, 2B, S5A, and S6B). Taken together, our results

indicate that the molecular interactions between the coacervate forming components modulate the physicochemical properties of the coacervate such polymer volume fraction which directly affects ribozyme diffusion and reaction kinetics.

## Discussion

In this work, we show how the physicochemical properties of membrane-free coacervate droplets influence and modulate ribozyme catalytic activity. Our study demonstrates that macroscopic droplet properties that emerge from molecular co-operativity directly impact ribozyme reaction kinetics.

In brief, we show that each peptide from our small library of seven cationic peptides will form coacervate droplets with the minimal hammerhead ribozyme. The ribozyme is functionally active within the coacervate dispersion, albeit with much slower rates compared to the buffer solution. Small variations in short, arbitrary cationic peptide sequences can lead to observable differences in the rate and yield of substrate cleavage, which we attribute predominantly to the coacervate micro-environment. Significantly, between 6 of the short peptides, we observed a 15-fold difference in both the rate and yield of the ribozyme reaction within coacervate dispersions.

Despite the slower reaction rate in the coacervate compared to the buffer solution, we measured an increase in $HH_{min}$ concentration within the coacervate up to ~200,000-fold. Furthermore, a weakly correlated decrease in the reaction rate with increasing ribozyme concentration inside the coacervates contradicts our general expectation of enzyme kinetics by mass action, where increased enzyme concentration is expected to increase the reaction rate. Our results agree with a recent theoretical study, which shows that this expectation could break down within the droplet environment[44]. This decrease in rate with increasing ribozyme concentration could be due to molecular or macroscopic effects. For example, on the molecular level, RNA-RNA interactions driven by high local concentrations could lead to substrate binding between two neighbouring ribozymes which could create a population that contains unproductive bound-states which attenuates the reaction. Increased packing (increased density and concentration) of the ribozyme within the coacervate can lead to changes in the secondary structure that could lead to loss of catalytic activity. In addition, peptide-RNA interactions could block RNA substrate binding sites. Here, stronger binding between the ribozyme and peptide can lead to denser droplets with increased concentrations but reduced catalytic activity. Further, some of the ribozyme within the droplet could be rendered inactive by coacervate formation. On the macroscopic level, molecular diffusion is slower in denser droplets, which can reduce reaction rates. Therefore, rate of ribozyme cleavage could be affected by substrate partitioning and diffusion through the coacervate as well as to changes in ribozyme activity, for instance from interactions with peptide. In addition, changes to local pH and salt can directly affect the ribozyme rate as well as modulate the properties of the coacervate which could contribute to changes in ribozyme rates (Supplementary Information Section 8.0: Note 5).

We also observed differences in the emergent physicochemical properties between the droplets, where the polymer volume fraction and the ribozyme diffusion coefficient were linearly correlated across the peptide sequences. It was possible to correlate the apparent rate constant of the ribozyme with an emergent property - namely the polymer volume fraction or molecular diffusion coefficient regardless of the details of the sequence. This can be important when considering biological systems, where it can be challenging to obtain precise chemical compositions. Our results suggest that by correlating measurable parameters such as diffusion coefficients to reaction rates one can still obtain new insights about biological systems. It is important to note that we have used seven peptides with a small subset of amino acids that provide a small sub-range of polymer volume fractions. Thus, expanding the amino acid set and library of sequences could provide further insights into the significance of polyelectrolyte chemistry on coacervate properties and its impact on tuning enzyme kinetics.

In the presence of substrate, the strength of the peptide-ribozyme interaction can affect the ability for the substrate to access the ribozyme binding site. In previous studies, it has been shown that tuning the molecular grammar of proteins can affect their phase behaviour[60–62]. In this study, it has not been possible to extract the exact contribution of the peptide sequence or its molecular grammar on the emergent properties of the coacervate droplet and its catalytic activity. Seven sequences are likely too few to extract a detailed molecular grammar for cationic peptide/HH min systems. However, we have identified three potentially predictive characteristics among the cationic peptide sequences that are worth discussing. First, the distribution of hydrophobic residues between positively charged amino acids could distribute the charge along the peptide sequence. This could lead to patches of increased charge density and stronger charge-charge interaction with $HH_{min}$ as seen in P-2 and P-4 peptides if, consequently, the number of charged residues are accumulated together (Fig. 6). Second, across all seven peptide sequences, we found that the tie-lines collapse into two families when plotted in units of charge concentration (Figs. 4 and S14). This confirms that charge units remain a useful basis for comparing phase behaviour in these systems even though the peptides contain many un-charged residues (Supplementary Information Section 5.4: Note 3). Further, the length of the tie-line increases with the net charge per peptide chain (Fig. 4), consistent with previous results on coacervates formed from homopolymers[63,64]. This shows that the negatively charged amino acids (D, E) change the phase behaviour of the coacervate. Finally, the number of polar non-charged sidechains (S, T) could influence coacervate phase behaviour as we observed two distinguishable groups of tie-lines in which P-1 and P-6 (each consisting of 3 polar non-charged sidechains) had a shallower slope compared to the remaining peptides (Supplementary Information Section 5.4: Note 3). This, along with previous observations with proteins[60–62] and peptides[40,65], shows how coacervate formation can be influenced by sequence. Taken together, it could be possible to engineer peptide sequences to modify material properties and the catalytic activity of coacervates or biomolecular condensates[66].

Our previous study showed that in pools of total RNA (from human induced pluripotent stem cells; iPSCs)[19], coacervates within the population can retain distinct chemical compositions. Together with the results presented here, which show that different peptide sequences can tune dispersion activities, this suggests that coacervate droplets could provide a selection pressure on molecular reactions. For example, pools of chemically diverse molecules such as metabolites, peptides, nucleic acids, or lipids could lead to coacervate droplets with different chemical compositions and diverse reaction outcomes within a population. Within an origin-of-life context, this presents an intriguing scenario for the evolution of molecular complexity and cooperativity in a prebiotic soup, while also offering insights for engineering compartments with controllable functions in synthetic cell design.

## Methods

### Coacervate formation

$HH_{min}$ and peptide concentrations were used as charge concentrations (molar concentration of polymer * number of charges per polymer) for coacervate preparation. Unless stated otherwise, 250 μM $HH_{min}$ was mixed with 500 μM peptide in 40 μL buffer (10 mM Tris, 1 mM $MgCl_2$ at pH 8.1) to produce coacervate droplets.

## Droplet imaging using confocal microscopy

For fluorescence microscopy imaging, 250 μM HH$_{min}$ (with 10% FAM-HH) was mixed with 500 μM peptide in 40 μL buffer to form droplets. Droplets were settled under gravity within Ibidi 18-well bottomless μ-Slide chambers with self-adhesive undersides attached to custom-made PEGylated glass coverslips (24 × 60 mm). Droplets were loaded and left to settle for 15–30 min before imaging. All imaging was performed using a Zeiss LSM 880 Airyscan inverted laser scanning confocal microscope equipped with a 63x/1.4 Plan-Apochromat, oil-immersion, DIC-compatible objective unless otherwise stated. For fluorescence imaging of FAM, $\lambda_{ex} = 488$ nm/$\lambda_{em} = 499$–561 was used. The temperature was maintained at 25 °C. Whole field-of-view Z-stacks were taken with a 1 μm slice interval.

Imaging with the FRET substrate (FRET-subs) involved adding 50 μM of FRET-subs to a dispersion of HH$_{min}$–peptide droplets. Immediately after addition, confocal Z-stack images (0.3 μm slice interval, 15 slices) in a time-lapse series with 1 min intervals for 60 min were acquired using $\lambda_{ex} = 488$ nm. Two emission wavelength ranges provided the ability to visualise the uncleaved substrate ($\lambda_{em1} = 614$–660 nm) that detects FRET transfer between the two dyes within the substrate and the cleaved product ($\lambda_{em2} = 490$–509 nm). The emission bands were carefully chosen to minimise spectral overlap between the emission spectra of the two dyes.

For FRAP studies 10% labelled FAM-HH (25 μM) was mixed with 90% unlabelled HH$_{min}$ (225 μM) and peptides (500 μM). Fluorescence recovery after photobleaching (FRAP) was measured with droplets of approximately 10–12 μm in diameter and an almost 1/4 diameter circle in the center of the droplet was selected for bleaching using 405 nm (100%) and 488 nm (100%) laser powers with two iterations. Bleaching was initiated after three initial scans; thereafter, images were captured every second, using the 488 nm laser ($\lambda_{ex} = 488$ nm, $\lambda_{em} = 499$–561 nm) for 200–300 s or, in the case of the slower dynamics for the P-7 system, for 600 s. Images were processed using ImageJ 2.3.051, and fluorescence intensities were plotted using GraphPad Prism 9.4.

## Assessment of ribozyme activity in the coacervate pellet and dispersion by gel electrophoresis

250 μM HH$_{min}$ was mixed with 500 μM peptides in centrifuge tubes and settled for 15 min before adding 50 μM FAM-subs to a final volume of 10 μL. Immediately after the substrate addition, the solution was gently mixed by pipetting and either incubated at room temperature without further agitation or immediately subjected to centrifugation. In the latter case, 3 μL and then 4 μL of the supernatant were removed. 3 μL of the supernatant and the remaining 3 μL of the pellet containing coacervate were incubated for 6 h. For samples which did not undergo immediate centrifugation, seven sets of samples (HH$_{min}$ mixed with seven peptides) with eleven different time points (2 min, 5 min, 10 min, 15 min, 20 min, 30 min, 45 min, 60 min, 2 h, 4 h and 6 h) were prepared in individual tubes. After each time point, the centrifuge tube was quickly spun down at 3000–5400 × g for 30 s, and 7 μL supernatant was withdrawn and removed. To the remaining 3 μL pellet, which contained the total condensed phase and some supernatant, 15 μL of 2x RNA loading dye mixed with 20 mM sodium hexametaphosphate was added to quench the reaction and an additional 10 μL buffer solution was added to dilute it further. The solution was stored at −20 °C until analysis by gel electrophoresis.

Analysis of ribozyme reactions performed within the whole solution was treated in the same way, but without the centrifugation and supernatant removal step, unless otherwise stated. For determining the reaction kinetics in the whole solution, 10 μL (250 μM HH$_{min}$, 500 μM peptide) of coacervate dispersion was prepared. After the addition of substrate (50 μM), the samples were incubated. After each time point (2 min, 7 min, 15 min, 30 min, 60 min, 2 h, 4 h, and 6 h), 10 μL of 2x of RNA loading dye with 20 mM sodium hexametaphosphate was added to the mixture, and an additional 20 μL of 50%

glycerol buffer was used to dilute the sample. The solution was stored at −20 °C until analysis by gel electrophoresis. 2 μL of sample was loaded onto a denaturing 20% UREA-PAGE and run at 300 V with 1X TBE buffer to monitor product formation. Fluorescence intensity from the substrate and the product was monitored using a Typhoon 9500 Fluo Phospho Imager (GE Healthcare Life Sciences) with the 473 nm laser and the filter model -BPB1/530F-20 with emission bandwidth 520–540 nm. Band fluorescence intensities (FI) were quantified using ImageJ 2.3.051 and HH$_{min}$ product formation was calculated as the fraction of total fluorescence intensity given by

$$\text{Product formation}(\%) = \frac{FI_{product}}{FI_{substrate} + FI_{product}} \times 100 \qquad (1)$$

The product formation (%) curve was fit to

$$Y = Y_{max}\left(1 - e^{-kt}\right) \qquad (2)$$

using GraphPad Prism 9.4 software, and the first-order rate constant ($k$) was obtained from the fit. Here, $Y$ is the fraction of product present at time $t$, $Y_{max}$ is the plateau value, $k$ is the apparent rate constant expressed in reciprocal of the time units and $t$ is time in minutes.

## Determination of HH$_{min}$ and peptide concentrations

Dense-phase concentrations of HH$_{min}$ and peptide were determined by application of a method as described in previous work[50]. A mathematical constraint equation for the tie-line can be obtained by linear interpolation of the two points defined by dilute-phase and average composition of the system for each peptide/HH$_{min}$ pair. This constraint equation is of the form

$$c_r^{cond} = m_{TL}c_p^{cond} + b, \qquad (3)$$

where $c_r^{cond}$ is the concentration of the RNA in the condensed phase, $c_p^{cond}$ is the concentration of peptide, the gradient of the tie-line slope is given by $m_{TL}$, and $b$ is the y-intercept. This equation provides one of two constraints that are solved simultaneously to determine the condensed-phase composition.

The second physical constraint is that the concentration differences between the thermodynamic phases must be consistent with the refractive index difference between the phases. Following previous work[50], here we employ a simple linear model to describe the compositional dependence of the refractive index difference, $\Delta n$, between the condensed and dilute-phases,

$$\Delta n = \frac{dn}{dc_p}\left(c_p^{cond} - c_p^{dil}\right) + \frac{dn}{dc_r}\left(c_r^{cond} - c_r^{dil}\right), \qquad (4)$$

where superscripts distinguish concentrations in the condensed and dilute phases and the subscripts $p$ and $r$ denote peptide and RNA, respectively. This second constraint equation provides a quantitative relationship between the unknown peptide and RNA concentrations in the condensed phase in terms of $\Delta n$, the refractive index increments $dn/dc_p$ and $dn/dc_r$, and the known dilute-phase concentrations of peptide, $c_p^{dil}$ and RNA, $c_r^{dil}$. The refractive index increments $dn/dc_i$ characterise the linear change in refractive index, $n$, of a solution with solute concentration and may be reliably estimated (see Tables S10 and S11). To measure the remaining free parameter, $\Delta n$, we used quantitative phase imaging[50,67] (Supplementary Information Section 5.6).

## RNase A footprinting assay

In centrifuge tubes, 25 μM FAM-HH and 50 μM peptide were mixed in 10 μL buffer and allowed to settle for 15 min. After adding RNase A (total final concentration: 100 nM), the reaction mixtures were

incubated for 12 min at room temperature. 20 μL of 2X RNA loading dye mixed with 20 mM sodium hexametaphosphate was added to the solution and immediately transferred to a −20 °C freezer. Samples were run on a denaturing 20% UREA-PAGE gel in 1x TBE buffer and imaged using a Typhoon 9500 Fluo Phospho Imager, with the 473 nm laser and the filter model -BPB1/530F-20 with emission bandwidth 520–540 nm. Band intensities were quantified using ImageJ 2.3.051 and plot using GraphPad Prism 9.4. The cleavage percentage $Y_{A_i}$ at location $A_i$ was obtained by:

$$Relative\ Band\ intensity(\%) = \frac{FI_{A_i}}{FI_T + \sum_{i=1}^{9} FI_{A_n}} \times 100 \qquad (5)$$

where FI is the band fluorescence intensity and the subscript denotes the band ($T$ = uncut and $A_i$ = respective cleaved fragments).

### Reporting summary

Further information on research design is available in the Nature Portfolio Reporting Summary linked to this article.

## Data availability

The processed and raw data generated in this study are available in the EDMOND database under accession code https://doi.org/10.17617/3.G6KSU2. Supplementary material is provided. Source data has been provided with this paper.

## Code availability

All code will be available at the following https://doi.org/10.17617/3.G6KSU2.

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

## Acknowledgements

We acknowledge financial support from the MaxSynBio Consortium, which is jointly funded by the Federal Ministry of Education and Research (Germany) and the Max Planck Society (T-YDT). We thank the Deutsche Forschungsgemeinschaft (DFG, German Research Foundation) under Germany´s Excellence Strategy – EXC-2068 – 390729961 – Cluster of Excellence Physics of Life of TU Dresden (T-YDT, JB) and EXC-1056 – Center for Advancing Electronics Dresden (T-YDT); and the Volkswagen Stiftung (grant numbers: 92772, 92857, T-YDT, HM) for generous funding. Co-funded by the European Union (ERC, MinSynCell, 101088834, T-YDT) and (ITN, DARCHEMDN Grant number 101119956, T-YDT). Views and opinions expressed are, however, those of the author(s) only and do not necessarily reflect those of the European Union or the European Research Council; neither the European Union nor the granting authority can be held responsible for them. PMM was supported by Volkswagen 'Life' grant number 96827 awarded to JB and subsequently by the Biocondensate Emerging Topic at the Leibniz Institute of Polymer Research Dresden. We thank A. Schwager and S. Ernst for preparation of PEGylated coverglass. We thank the Light Microscopy Facility (LMF), Scientific Computing Facility, and Protein expression, purification and characterisation Facility (PEPC) of the Max Planck Institute of Molecular Cell Biology and Genetics (MPI-CBG) for their technical support and useful discussions. We further thank Frank Jülicher, Tyler Harmon, and Christoph Weber for their valuable discussions and Carl Modes for help with proofreading.

## Author contributions

H.M., B.G., K.L.V., and T.-Y.D.T. designed the study. B.G. and P.M.M. designed and performed the experiments and analyzed the data. L.H., K.L.V., A.G., D.G., and J.B. contributed to experimental design, methodology, and formal analysis. P.M.M. performed the simulations. A.G. contributed to the simulations of pH. All authors contributed to conceptualisation, resources, and writing the manuscript.

## Funding

## Competing interests
The authors declare no competing interests.
