## [Transparent Peer Review file · Nature Communications]

RNA-peptide interactions tune the ribozyme activity within coacervate microdroplet dispersions

Corresponding Author: Dora Tang

Version 0:

Reviewer comments:

Reviewer #1

(Remarks to the Author)
Summary

This study investigates the effect of short peptide sequences (13 amino acids) on different physico-chemical properties of coacervates that are formed with an RNA, specifically a hammerhead ribozyme. It is impressive that the ribozyme is up-concentrated as much as 200,000-fold in the coacervate phase. Additionally, the effects on the activity, and the secondary structure of the ribozyme are studied. A strong correlation is found between the coacervate's polymer volume fraction, and the diffusion constant inside the coacervate droplets, with the catalytic rate of the ribozyme in the coacervate droplets (figure 5). Additionally, an RNase A protection assay shows that the peptide sequences have different effects on the ribozyme's secondary structure (figure 6). These are important results because they link the properties of the coacervate with catalytic activity within, providing important data for origin-of-life scenarios. Most experiments are well-executed and very insightful for the understanding of RNA/peptide coacervates.

The importance of the data is somewhat reduced by the information that the ribozyme activity is inhibited in the coacervates by 2-3 orders of magnitude - which means that the effects of the different peptides are different levels of inhibiting the ribozyme. This important context needs to be mentioned, and discussed in every context that relates the results to origin-of-life scenarios. Additionally, the title and abstract should be less vague and focus on the results, the discussion should discuss alternative explanations whenever relevant, words and phrases should be used in the way they are usually used, and the interpretation of the results needs to be more careful with respect to prebiotic relevance. Similarly, there is insufficient discussion of results from other groups that have already published important work on the exact topic of this manuscript, the influence of peptide side chains on the physico-chemical properties of coacervates, and the activity of a ribozyme in the coacervates (e.g. by the Keating & Bevilacqua labs).

The study may be suitable for Nat. Commun. after points 1-7 below are addressed, and the minor comments are addressed at the author's discretion.

Comments on the manuscript:

(1) As described in lines 165-172, the peptides reduce the ribozyme reaction rate and yield by 2-3 orders of magnitude. This important finding needs to be represented in the abstract and the last paragraph of the introduction, otherwise it seems that undesired results would be hidden. In addition, this observation influences a number of interpretations throughout the manuscript that need to be done more carefully (see below).

Paragraph L178 to L196: It is unclear why only an 'underestimate' is considered and not an 'overestimate'. Indeed, an overestimate seems more likely since the reaction in the supernatant is significantly more efficient than in the coacervate (L165-172), and an exchange between supernatant and coacervate phase, or carryover of supernatant with the coacervate phase, could give the impression of a reaction in coacervates even if all reaction would occur in the supernatant. The differences between samples could then arise because specific peptide sequences may mediate more exchange, or more carryover. This means that (a) the word 'underestimate' needs to be replaced by a word that covers both types of systematic errors, and (b) it needs to be discussed in this paragraph why the ribozyme reaction yields measured for the coacervate phase are not overestimates (similar to the first sentence of the paragraph in L176-177).

This also means that it is unclear how it can be concluded in L193-195 "that the cleavage reaction may occur primarily within the coacervate phase" based on the observation that "the vast majority of all RNA species are associated to the

coacervate phase": Since the reaction in the supernatant is 2-3 orders of magnitude faster / higher amplitude than in the coacervate, the same results could be obtained if there is an exchange between supernatant and coacervate, or if there is a carryover of supernatant in the procedure that separates pellet and supernatant. Correspondingly, in figure 2 please insert data on the reaction in the supernatant (for example, as an additional column in Ci, perhaps with a 'broken' column. It is unclear how a carryover of supernatant in the 3 uL of 'pellet' (figure 2A) was avoided: If a small volume of supernatant was carried over with the 'pellet fraction' then it could invalidate all ribozyme activity data in this study. This is an important issue because this conclusion affects the interpretation of most data in this study. For example, the observation in L213-215 (FRET paired substrate binding to ribozyme) could be explained differently (from L215-220) by an exchange between the inside and outside of coacervates, with the reaction proceeding exclusively outside coacervates.

(2) In a few cases, a discussion leaves out important information, resulting in a lack of credit given to other researchers, or in the impression that inconvenient, alternative explanations are not considered.

L98-103: The sentences from "This system provides ... primitive cells" make it appear as if such work has not been done before - which would be incorrect. The work of Christine Keating and Phil Bevilacqua, among others, has already provided groundwork to this understanding. It is therefore necessary to better specify what is the new insight, and use a phrase such as 'expand our understanding' to make clear what work has been done before on peptide/RNA interactions and their effects on RNA catalysis in coacervates, and what is new in this study.

(3) Especially in the abstract and introduction, the manuscript should describe more clearly what are the new findings of this study. Please see my summary for the findings I was able to identify.

(4) Words and phrases are often used unconventionally. To make the content more clear (and thereby increase the manuscript's impact), it is important to use words and phrases in the usual way, and to simply sentences whenever possible. Some examples are below, and some additional, minor examples are given under 'minor comments':

Throughout the manuscript, the term 'random' is used incorrectly; it should be replaced with the words 'arbitrary' or 'specific' (the word 'representative' may not match because six of the seven sequences are not designed to represent classes of sequences). While a random sequence generator was used to come up with six specific sequences this does not make the sequences random, it makes them arbitrary. In contrast, the word 'random' implies that the sequence of peptides is at least partially randomized is used - but the used seven peptides have defined sequences.

L67-76: The sentences from 'Despite' to 'by biology' describe that 'RNA-peptide interactions in coacervates are important to consider for the properties of the individual molecular species'; this could which could be expressed more clearly and succinctly in a single (or at most two) sentences.

L91: The statement "coacervates prepared from ribozyme and each of the seven peptide sequences result in droplets with distinct chemical compositions." does not contain information: The compositions *are* 'ribozyme + peptide 1' vs. "ribozyme + peptide 2" etc. Instead of 'chemical compositions, should it be 'physico-chemical properties'? Is the intention to say that each peptide, when mixed with the ribozyme, has its own optimal peptide - ribozyme stoichiometry (this would overlap with the information of the next sentence)?

L90-103: Consider re-writing the sentences from "We show ..." to "... primitive cells" much more concisely, with a focus on the gained new information, and insight. For example, these sentences in lines 90-103 could be replaced - without the loss of important information - much more concisely by "Coacervates prepared from ribozyme and each of the seven peptide sequences showed differences in the condensed-phase polymer volume fraction and molecular diffusivity, as well as ribozyme reaction rate and product yield. Peptide sequences with higher net charge led to denser packing of the ribozyme and peptide and slower ribozyme diffusion and slower reaction rates. This study expands our understanding of how coacervates formed within a prebiotic soup can affect the properties of compartments and catalysts within."

L342: Consider removing unnecessary wording from the sentence "Despite having shown that the rate of the reaction for coacervate systems is driven by the emergent properties of the coacervate, it cannot be ignored that these properties could also be affected by the underlying molecular interaction between charged RNA and peptide that is driven by the RNA/peptide sequence." After reading this sentence five times I realized that its message was very simple, and it could be added as a side sentence to the first sentence of the next paragraph. Note that the sentence in L347 only makes sense of such a side sentence is included.

(5) The abstract and introduction claim that the used peptides are 'prebiotic' (e.g. line 25). However, there are two problems with this statement: First, we will never know whether certain peptides were 'prebiotic'; we can only assume that they were 'prebiotically plausible'. Second, the amino acid composition described in lines 112-114 is in my opinion not prebiotically plausible: All peptides contain more than 50% arginine / lysine, and neither of these two amino acids seem prebiotically plausible at a quantity required for the formation of such peptides. While two laboratories have devised a synthetic route where arginine could have been generated prebiotically, the expected concentrations would not be in a range that would allow the formation of peptides containing 50% of arginine or lysine. To address this issue, either an argument needs to be presented up-front (the latest in the first paragraph of the results section) that these peptides are indeed 'prebiotically plausible' or the claim of prebiotic plausibility is dropped and the peptides serve as model compounds that are closer to prebiotically plausible peptides than previous studies. I believe that the latter would be more appropriate.

(6) L1: Please clarify the title. In the title, it is not clear what the term 'activity' refers to: droplets are usually not associated with an 'activity' (this word is mostly used in the context of catalysis). However, the manuscript describes effects on ribozyme activity and physico-chemical properties of the droplets.

(7) The counter-intuitive finding that the ribozyme-catalyzed reaction is slower at higher ribozyme concentration in the coacervates needs to be critically evaluated, and alternative explanations discussed. For example, the same anti-correlation would be expected if a given substrate molecule in the droplet could be bound by the arms from two different hammerhead

ribozymes at increased ribozyme concentration. That arrangement would not allow the formation of the hammerhead ribozyme's catalytic core, and thereby lead to lower activity.

Minor comments:

L20: I think that there should be a 'they' inserted so that it reads "... biological molecules; they concentrate molecules ...".

L41: would 'environment' be more clear than 'milieu'?

L43: I believe that it is 'cooperation' and not 'co-operation'.

L44: convert to 'physico-chemical properties of the compartment such as the polymer volume fraction and ...'?

L49/50: the current description leaves it unclear what determines the formation of coacervates as opposed to larger aggregates / precipitation - because both processes can be a result of the stated polyelectrolytes. If this would be a more involved explanation then it would be helpful to include a statement such as 'the distinction between the formation of small coacervate droplets and large aggregates or precipitates depends on subtle changes in molecular properties and solution conditions, as described in [reference].'

L53: "there is an increasing number of studies" (singular).

L62: The words 'in an origin of life context' can be removed to simplify the sentence. This works because the following statements are general statements and not restricted to an OoL scenario.

L67-70: The statement 'there have been a limited number of studies' should be supported by citing those studies.

L87: what does the word 'chemistry' mean here? Does it mean 'chemical reactions'. Similarly, L310 & L311 would be better served by a different word such as 'amino acid sequence' or 'amino acid composition and sequence'. The sentence "the coacervate composition can vary depending on the chemistry of the coacervate-forming components," does not contain information because the coacervate composition is defined by the coacervate-forming components.

L91: A very minor point: Please consider using a different term than 'emergent' because this term is used in many, quite different contexts in OoL literature. The term 'emergent' is usually used to describe higher-order phenomena that are not necessarily expected from the physico-chemical properties of the individual components. However, the listed results seem to be just 'properties of the mixture' that are not necessarily unexpected.

L94: It is unclear what information is in that sentence. Independent of the intended meaning of the word 'underpinned', since the only polyelectrolytes in the mixture are RNA and peptides, and the only difference between the mixtures is the sequence of the peptide, it is trivial that

Figure 1A: The circles around each amino acid make the figure unnecessarily complicated. I suggest removing the circles. Additionally, I suggest using slightly more space between the rows than between the columns. In the current format, it seems that there amino acids above each other are as connected as amino acids beside each other; that visual impression should be avoided.

Figure Bii: the number of product molecules gives the impression of multiple turnover. I don't remember seeing a description of multiple turnover in the text. If there is multiple turnover, please describe it clearly in text and legend; if there is no multiple turnover please reduce the number of product fragments to two.

L133: The term 'number density' is unfamiliar to me. please clarify.

L141: Should 'sub molar' be 'sub millimolar'? The peptides seem to be used at 0.5 mM, which would be sub-millimolar.

Figure 2A: Please show all ribozyme schematics in the same size, and all peptides in the same size. Otherwise the impression is raised that peptides of different length are employed, and that impression should be avoided. I suggest converting the 'circular' text to a straight text. The 'circular text' makes it harder to read.

Figure 2Bii: I suggest removing the squares with different gray tones and the 'gray tone scale'. These two elements are unnecessary / redundant and make the figure more complicated than necessary. The numbers below the squares are sufficient, together with the gel image.

Figure 2Cii: It is unusual to display catalytic rates as column graph because the value is a discrete number, not an amount. Shouldn't the 'k' be lowercase? I believe that kinetic values are lowercase; thermodynamic values are uppercase.

L302: I suggest converting 'obtained' to 'estimated' or 'calculated' because the RNA and peptide concentrations are derived from somewhat complicated models as opposed to direct measurements.

L352: Please add that this RNase A experiment not only identifies direct RNA interactions between RNA and peptide but also indirectly identifies whether annealing with another RNA strand occurs.

L355: It is unclear why the "lower concentration of the RNA and peptide" would allow "to focus on the RNA-peptide interaction alone and to avoid any issues that may arise from RNaseA diffusion through the coacervate." Does that mean that coacervate formation is avoided? If so, wouldn't this mean that the interactions could be very different with single peptide molecules and RNA as opposed to aggregated peptide molecules in the coacervate with RNA? Please clarify.

Figure 4: The schematics in (A) should use the same axis labels as the graphs containing the measured data in Bi and Bii. Please plot the schematics in the same linear scale (not log) as the graphs with measured data.

I did not understand the text corresponding to figure 4, or the figure itself. While this is in part my educational background, an effort to make the text and figure accessible to a broader audience would be appreciated.

L397 and many other instances: I suggest replacing the word 'tune' to describe the influence of the coacervate on ribozyme activity because 'tuning' usually requires an agent with an intention such as an experimenter or engineer. I would find 'influence' more neutral and appropriate.

L369: there should be a space between 'RNase' and 'A'

Figure 6: Please remove sub-panel 6D because it adds no information and makes the figure more confusing / less clear. The introduction of the gray scale is redundant with the height of columns in A9 of panel C. If the authors don't deem the columns in sub-panel C clear enough then the sub-panel C should be increased in size. The averages and error bars shown in sub-panel D are unnecessary because they are represented in the columns and their error bars in sub-panel C.

L402: I suggest refraining from the use of the word 'library' because this implies that the peptides were used as a mixture. I suggest calling them 'seven peptides'.

L407: This is usually called a '15-fold difference'.

L447: Please clarify what 'total RNA' means. Is that total RNA from an organism?

L450: Please add a few words to explain what is meant by 'chemically diverse molecules', perhaps by adding 'such as x, y, z.'

Reviewer #2

(Remarks to the Author)

In this report different combinations of RNA hammerhead ribozyme and a small peptide library are studied for their coacervation performance and catalytic activity. Because the peptide library is cationic in charge, all combinations give coacervates. There are differences in density, which is also related to the activity of the ribozyme.

This work is a continuation of earlier work by the group and the findings are to be expected, also in line with for example the work by the Keating group. As such, the results are rather incremental. The peptide library used is moderate of size and although some design results are extracted from the results in the discussion, these aspects are not further corroborated. Besides this general reservation about the novelty of the research, a number of technical comments should be addressed.

First of all, the coacervates are made in 10 mM Tris-136 HCl, 1 mM MgCl₂, pH 8.1. How sensitive are the coacervates to varying pH and electrolyte concentration?

What is the pH in the microenvironment of the different coacervates? This could also have an effect on the catalytic activity of the ribozyme.

Furthermore, there is a sharp change between P3 and P5 in fig 4F when changing the ribozyme concentration. This cannot be merely explained by the concentration.

How does the particle size distribution affect the results obtained?

Reviewer #3

(Remarks to the Author)

In the manuscript "RNA-peptide co-operativity tunes the activity of coacervate microdroplets dispersions," Ghosh et al. present that coacervates composed of randomly generated sequences of peptides can alter ribozyme kinetics. The authors randomly generated small variation in sequences of 13-mer peptides with a fixed percentage of amino acid types and used them with hammerhead (HH) ribozyme to produce the coacervate phase. While this peptide library is rather modest size, they showed that the overall ribozyme rate constant in coacervate phase is anti-correlated to the ribozyme concentration and diffusion coefficient of ribozyme.

Although discussions have arisen regarding the testing of randomly generated peptides in the emergence of life, no reported works have attempted to use such peptides for coacervation, to my knowledge, due to potential difficulties and limitations in coacervation and interpretation. In this context, this work is intriguing, offering valuable insights into the ability of small randomly generated variation in sequence of peptides, devoid of biased design or inspiration from biological proteins, to produce coacervates with RNA that vary in composition and chemical properties, influencing RNA reactivity within. The

manuscript is original and logically ordered, supported by experimental results. However, there are some concerns regarding substrate diffusion, local ribozyme reactivity, and the rationale behind choosing a 10x higher initial concentration for polymer content estimation. Given the fact that this manuscript can potentially have a broader impact by establishing a linkage between the physicochemical properties of membrane-free compartments and their molecular reactivity, I would recommend publishing in Nature Communications after addressing these concerns. Details are provided below:

1. While I understand the authors' intent to maintain cationic amino acid composition for coacervation with HH ribozyme, I couldn't fully follow the reasoning for choosing these composition ratios in amino acid types. For example, in the provided Python code in SI, each amino acid for acidic, polar, and hydrophobic amino acids has only a 0.03 probability each, while hydrophobic amino acids have a 0.1 probability each. As this could provide some bias in amino acid selection from amino acids pools given to the code, provide any reasons to support the fact that these are "randomly" generate peptide sequences.
2. The cited papers in the main text (ref [36] – [39]) seem like they do not mention arginine nor lysine. Please cite papers producing amino acid analogues containing amine groups to justify arginine and lysine. Longo et al ("Primordial emergence of a nucleic acid-binding protein via phase separation and statistical ornithine-to-arginine conversion" PNAS (2020) 117, 27, 15731-15739) and Plankensteiner et al ("Amino acids on the rampant primordial Earth: Electric discharges and the hot salty ocean", Molecular Diversity (2006) 10, 3–7) may be helpful.
3. A HH ribozyme mutant is a great control for the FRET experiment.
4. I agree that the overall reactivity rate of the ribozyme can be limited by the diffusion of substrates, as it takes ~1 hour to fully reach the center of droplets. However, in Fig 3 and S7, the product is well colocalized with the substrate in the snapshots of 5 mins and 20 mins, although the overall reaction rates from gel analysis in Fig 1 are extremely low (e.g., 0.006/min for P3). A discussion of substrate partitioning and diffusion in relation to local vs overall ribozyme reactivity and coacervate properties needs addressing, linking it to the comparison of interactions between peptide/ribozyme versus RNA/substrate if possible.
5. Peptide and RNA composition analysis via quantitative phase microscopy is interesting and has great potential in this community. Perhaps citing the paper by Kim et al. (Nat Commun 14, 2425 (2023)) as another example of this method would be appropriate.
6. However, on pg. 7, the last paragraph, the reasoning to use 10x higher peptide and ribozyme concentration is unclear. On a phase diagram, it is not guaranteed that a 10x higher initial concentration of peptide and ribozyme would lead to the same trend as the ones used for ribozyme reactivity experiments. This is critical for comparing their relationship in Figure 5D. Two suggestions:
 - 1) If the increase in 10x peptide and ribozyme concentration was to increase the number of droplets and droplet sizes, consider making a larger sample size and letting them sit for a longer time to produce more coacervates on the cover glass (assuming that coacervate droplets would get bigger via gravity and coalescence at a fixed surface area of the cover glass).
 - 2) While I appreciate the raw data in S11, consider showing the raw data from 1x peptide and ribozyme concentration and state in the SI the limitations of this measurement. Try 2x or 5x of two extreme cases of peptides (perhaps P6 and P4) to ensure that the peptide and ribozyme concentration in the coacervate phase changes somewhat linearly with their initial concentration.
7. The interpretation of comparing the A9 site of HH ribozyme cleavages in the presence of different peptides seems valid. However, to support the assumption that RNase cannot cleave the ribozyme sites if they interact with peptides or proteins in the RNaseA footprinting assay, it is important to include some comparison of total uncut RNA between the negative control of RNaseA reaction of HH ribozyme without peptides (as in Fig S15) and the positive controls of RNaseA reaction in the presence of peptides (as in Fig S6). Although they are not in the same gel, this should be addressed.
8. In the last paragraph on pg. 11, "sequence-specific between peptide and ribozyme interaction" could imply a specific sequence motif of the protein (in this case, peptide) interacting with a specific sequence of ribozymes coordinated by hydrogen bonding, cation-pi interaction, etc., which is not supported by the experiment here. I suggest rephrasing this.
9. Additionally, it is unclear why peptides with a distribution of hydrophobic residues between positively charged amino acids lead to increased charge density. The pattern of charged amino acids seems similar (e.g., 1-1-4-2 for P2 vs. 1-3-4 for P4). Do you mean the local hydrophobicity by hydrophobic amino acid residues can change the effective charge? Please clarify this.

Version 1:

Reviewer comments:

Reviewer #1

(Remarks to the Author)

The manuscript is much improved over the first submission, and I support publication after one critical alteration have been made. I do not support publication with the current title.

Specifically, the title of the manuscript is still misleading by using the term 'RNA-peptide cooperativity tunes...'. The term 'RNA-peptide cooperativity' would be understood that ribozyme activity is enhanced by the peptides. However, the major effect of the peptides is that ribozyme activity is inhibited by 2-3 orders of magnitude, in peptide coacervates as compared to the free ribozymes. The minor effect is that some peptides are up to 15-fold less inhibitory than others. The effect of 'less inhibition by some peptides' is still interesting but we should not present a misleading picture. I request that the title is modified to "RNA-peptide interactions tune the ribozyme activity...".

(Remarks on code availability)

Reviewer #2

(Remarks to the Author)

The authors have extensively answered to the comments raised by the reviewers. Because of the answers my initial concerns regarding novelty and interpretation of the results have been lifted. I feel the manuscript is now suitable for publication

(Remarks on code availability)

Reviewer #3

(Remarks to the Author)

The authors have done substantial works to address major concerns of the reviewers, making this manuscript suitable for publication in Nature Communications.

(Remarks on code availability)

We thank the editor and the reviewers for their constructive comments with respect to our manuscript previously titled **“RNA-peptide co-operativity tunes emergent properties and activity of coacervate microdroplet dispersions”** now titled **“RNA-peptide co-operativity tunes the activity of coacervate microdroplet dispersions”**. We address each comment in a point-by-point fashion, please find our responses to the review comments below. The original reviewer comments are **in bold** and every change and addition to the main manuscript and supplemental text has been highlighted in **yellow** in our response and the resubmission.

REVIEWER COMMENTS

Reviewer #1 (Remarks to the Author):

Summary

This study investigates the effect of short peptide sequences (13 amino acids) on different physico-chemical properties of coacervates that are formed with an RNA, specifically a hammerhead ribozyme. It is impressive that the ribozyme is up-concentrated as much as 200,000-fold in the coacervate phase. Additionally, the effects on the activity, and the secondary structure of the ribozyme are studied. A strong correlation is found between the coacervate's polymer volume fraction, and the diffusion constant inside the coacervate droplets, with the catalytic rate of the ribozyme in the coacervate droplets (figure 5). Additionally, an RNase A protection assay shows that the peptide sequences have different effects on the ribozyme's secondary structure (figure 6). These are important results because they link the properties of the coacervate with catalytic activity within, providing important data for origin-of-life scenarios. Most experiments are well-executed and very insightful for the understanding of RNA/peptide coacervates.

The importance of the data is somewhat reduced by the information that the ribozyme activity is inhibited in the coacervates by 2-3 orders of magnitude - which means that the effects of the different peptides are different levels of inhibiting the ribozyme. This important context needs to be mentioned, and discussed in every context that relates the results to origin-of-life scenarios.

Additionally, the title and abstract should be less vague and focus on the results, the discussion should discuss alternative explanations whenever relevant, words and phrases should be used in the way they are usually used, and the interpretation of the results needs to be more careful with respect to prebiotic relevance.

Similarly, there is insufficient discussion of results from other groups that have already published important work on the exact topic of this manuscript, the influence of peptide side chains on the physico-chemical properties of coacervates, and the activity of a ribozyme in the coacervates (e.g. by the Keating & Bevilacqua labs).

The study may be suitable for Nat. Commun. after points 1-7 below are addressed, and the minor comments are addressed at the author's discretion.

Comments on the manuscript:

(1) As described in lines 165-172, the peptides reduce the ribozyme reaction rate and yield by 2-3 orders of magnitude. This important finding needs to be represented in the abstract and the last paragraph of the introduction, otherwise it seems that undesired results would be hidden. In addition, this observation influences a number of interpretations throughout the manuscript that need to be done more carefully (see below).

We have modified the abstract and the introduction to highlight the decrease in the ribozyme reaction rate within the coacervate droplet; in the abstract, from:

“We show that small variations in peptide sequence can tune rates and yield of the ribozyme up to 15 times.”

To

Ribozyme activity is inhibited within the coacervate compared to buffer solution. However, variations in peptide sequence can modulate rates and yield of the ribozyme within the coacervate droplet by up to 15-fold.

In the introduction from:

“We show that coacervates prepared from ribozyme and each of the seven peptide sequences result in droplets with distinct chemical compositions.”

To:

“We show that coacervates prepared from ribozyme and each of the seven peptide sequences result in droplets with distinct chemical compositions i.e. different peptide/ribozyme concentration and stoichiometry..... It is important to note that the ribozyme activity was reduced by 2-3 orders of magnitude within the coacervate compared to free buffer.”

Paragraph L178 to L196: It is unclear why only an 'underestimate' is considered and not an 'overestimate'. Indeed, an overestimate seems more likely since the reaction in the supernatant is significantly more efficient than in the coacervate (L165-172), and an exchange between supernatant and coacervate phase, or carryover of supernatant with the coacervate phase, could give the impression of a reaction in coacervates even if all reaction would occur in the supernatant. The differences between samples could then arise because specific peptide sequences may mediate more exchange, or more carryover. This means that (a) the word 'underestimate' needs to be replaced by a word that covers both types of systematic errors, and (b) it needs to be discussed in this paragraph why the ribozyme reaction yields measured for the coacervate phase are not overestimates (similar to the first sentence of the paragraph in L176-177).

This also means that it is unclear how it can be concluded in L193-195 "that the cleavage reaction may occur primarily within the coacervate phase" based on the observation that "the vast majority of all RNA species are associated to the coacervate phase": Since the reaction in the supernatant is 2-3 orders of magnitude faster / higher amplitude than in the coacervate, the same results could be obtained if there is an exchange between supernatant and coacervate, or if there is a carryover of supernatant in the procedure that separates pellet and supernatant.

Correspondingly, in figure 2 please insert data on the reaction in the supernatant (for example, as an additional column in Ci, perhaps with a 'broken' column. It is unclear how a carryover of supernatant in the 3 uL of 'pellet' (figure 2A) was avoided: If a small volume of supernatant was carried over with the 'pellet fraction' then it could invalidate all ribozyme activity data in this study. This is an important issue because this conclusion affects the interpretation of most data in this study. For example, the observation in L213-215 (FRET paired substrate binding to ribozyme) could be explained differently (from L215-220) by an exchange between the inside and outside of coacervates, with the reaction proceeding exclusively outside coacervates.

We thank the reviewer for this comment. We agree that it is important to clarify the points regarding the location of the reaction and the under/over-estimation of the rate constant from the measurement within the coacervate pellet. However, it can be exceedingly challenging to identify the precise location of the reaction due to the ability of the coacervates to share material between the droplet and outer aqueous phase and the difficulty in isolating just the coacervate phase alone.

Despite this, we have undertaken additional experiments to clarify whether the coacervate environment can facilitate the product formation. Firstly, we sought to determine the increase or decrease in the product when the supernatant had been separated from the pellet. To do this, we prepared coacervate droplet dispersions from HH_{min} (250 μM) and each of the peptides (500 μM) in 10 mM Tris and 1 mM MgCl₂ at pH 8.1 as previously described. Immediately after the addition of substrate each dispersion was subjected to centrifugation and 3 μL of the supernatant was removed and placed in a separate centrifuge tube and incubated. From the remaining 7 μL, 4 μL of supernatant was removed and the remaining 3 μL which contained the coacervate pellet and supernatant was incubated. 3 μL of the supernatant and the coacervate pellet were incubated for 6 hours at room temperature quenched using 2x loading dye and 20mM Sodium hexametaphosphate. The samples were loaded onto a 20% UREA-PAGE gel and run at 300V as described previously.

The ImageJ gel analysis function was used to obtain band intensity of the product in the pellet and the supernatant to obtain the relative difference between the coacervate pellet and the supernatant. Please note that both the pellet and supernatant samples are 3 μL in volume with the only difference between the samples being that the pellet contains additionally the coacervate phase to the supernatant. Therefore comparing 3 μL of pellet with the supernatant provides the difference attributed to the coacervate dense phase. Our analysis was used to estimate the relative product in the pellet vs the supernatant.:

$$RD_p = BI \left(\frac{Pellet}{Supernatant} \right) \text{ or}$$

Where RD_p is the relative difference of the pellet and RD_s is the relative difference in the supernatant. BI is the product band intensity.

Our results show that in all cases, RD_p > 1 which indicates that more product was present in the pellet compared to the supernatant. RD_p varied across the peptide coacervate systems with P-1 showing the greatest difference of approximately 50x +/- 20x and to the smallest difference of 5x +/- 2.3 in the pellet with P-6.

Figure R1: Gel image showing HH_{min} reaction in the supernatant (3 μL) separated immediately after substrate addition (right panel) and incubated for 6 hours along with 3 μL of pellet containing coacervate phase (left panel) also isolated from the supernatant right after substrate addition. The supernatant and the coacervate pellet were incubated for 6 hours and run on 20 % Urea-Page gel. Analysis of the product bands using ImageJ show that more of the reaction taking place within the coacervate pellet.

We used our additional data to estimate the amount of product that would be produced if the reaction only took place in the supernatant and then partitioned into the coacervate phase (a hypothetical scenario B) using a theoretical approach. Our theoretical approach provided the ability to estimate the amount of product in the pellet phase under scenario B. Comparisons of the amount of product in the pellet phase from experiments and from scenario B simulations, show that there were orders of magnitude more product in the pellet, obtained experimentally compared to the prediction of a scenario B. This result provides some confirmation that the reaction takes place within the coacervate droplet.

Figure S6: **A.** Schematic illustrating the essential features of the Scenario B model. Condensed-phase HH_{min} is catalytically inactive while HH_{min} in the dilute phase converts substrate to product with rate constant $k_{I,eff}$. Substrate and product exchange between condensed and dilute phases with rate constant k_{ex} . **B.** Comparison of product amounts in the pellet observed experimentally (blue) to those predicted by simulation of the Scenario B model after 360 min (orange). In each case, the observed amounts exceed the predictions of Scenario B by several-fold. These data strongly suggest ribozyme is catalytically active in the dense phase.

Secondly, to confirm that the ribozyme rate constant is underestimated within the pellet, we directly compared the reaction within 3 μ L of the coacervate pellet with the whole dispersion. To do this, we prepared dispersions of peptide/ RNA coacervates as described previously and added substrate to trigger the reaction. We use gel electrophoresis to obtain the substrate and product ratios as a function of time for each peptide systems. Analysis of the kinetic profiles to obtain rate constants from 3 μ L of pellet show that the rate constant was the same, within error, or lower than the rate constant obtained from the whole dispersion. Given that in all peptide systems, more product is formed in the coacervate pellet compared to the supernatant (Figure R2) this underestimation is most likely attributed to the partitioning of the product from the coacervate droplets to the supernatant.

Taken together, our additional results confirm that our quoted rates (Figure 2) are underestimated, where the majority of the ribozyme cleavage takes place in the coacervate phase and the product RNA partitions out into the supernatant.

In response to the reviewer's comments regarding the 2-3 order-of-magnitude increase in reaction within the buffer. We agree that, if the reaction is more efficient in the supernatant than the coacervate then exchange between the two phases would lead to an overestimation of the RNA in the pellet. However, it is important to note that the supernatant of the coacervate phase is not equivalent to the buffer. The supernatant of the coacervate dispersion will contain RNA and peptides (see figure 4 and supplementary figure S10). In comparison the reaction in the buffer solution will only contain the RNA. As peptide and RNA are likely to form complexes the rates of ribozyme activity in the supernatant will be affected by peptide complexation and are unlikely to be equivalent to free ribozyme reaction in buffer. Consistent with this, we see that most of the substrate obtained from supernatant that was isolated immediately following substrate addition was unreacted after 6 hrs (Figure R1). This is in strong contrast to the near complete reaction of HH_{min} and the substrate in buffer (i.e. absence of peptides) within 20 min.

Figure R2: Comparison between ribozyme cleavage rates from the whole dispersion and from the coacervate phase containing pellet. **A**. Representative gel electrophoresis experiments of whole dispersion reaction kinetics for each of the 6 peptides. **B**. Kinetic profiles of product formation in whole dispersions for each of the peptide/ RNA systems obtained from gel electrophoresis error bars from 3 repeats. (ii) Comparisons of reaction rates obtained from a dispersion of coacervates (grey) vs the centrifuged coacervate pellet (3 μL) from the coacervate containing pellet (blue) (data from figure 2). Our new results from the whole dispersion show that in most cases the reaction rate has been underestimated within the coacervate pellet.

The reviewer is absolutely correct in his assertion that the 3 μL of coacervate phase will also contain some supernatant. It is very difficult, with these volumes to ensure complete removal of the

supernatant. For this reason, we estimated an error in the measured reaction rate, by quantifying the amount of product in the supernatant after 6 hours of incubation (supplementary figure 5). In all the samples this underestimation lay between 9 and 13%.

Again, we thank the reviewer for the important comments, we have consequently revised the manuscript and the supplementary information. We have included the data from new experiments into the supplementary figure and included additional explanations in the manuscript. Further, we have included a full explanation and derivation for the modelling of scenario B in a new supplementary note 2. We decided to include the new data into a new supplementary figure S5 rather than in the main text (as suggested by the reviewer) to avoid confusion between the gels that provide information on the product yield in the coacervate pellet after incubation (main figure 2) and the additional experiments we undertook for the pellet and supernatant which had been centrifuged and separated directly after addition of substrate and then further incubated (new supplementary figure 5). We have further included a statement in the discussion to specify the requirement in future studies to determine ribozyme rates within the coacervate and dilute phase separately.

New supplementary 5:

Figure S5: Ribozyme is active within coacervate dispersions: **Ai.** Gel image showing HH_{min} reaction in the coacervate pellet and the supernatant. The supernatant (3 μ L) was separated from the coacervate pellet (3 μ L) immediately after substrate addition by centrifugation. Each of the separated samples were incubated for 6 hours and then run on gel electrophoresis. **Aii.** Analysis of the product bands of the coacervate pellet (left) and the supernatant (right) using imageJ provide the RD_p which is the relative difference between the product band intensities of the coacervate pellet over the supernatant. The results show that in all cases there is more product produced within the coacervate pellet compared to the supernatant. Error bars are obtained from at least 2 repeats. **B.** Kinetic profiles of product formation in whole dispersions for each of the peptide/ RNA systems obtained from gel electrophoresis (ii) Comparisons of rate constants obtained from a dispersion of coacervates (grey) with rate constants obtained from 3 μ L of coacervate pellet (blue) (data from figure 2). The results show that the reaction rates are underestimated within the coacervate containing pellet. **C** Gel electrophoresis (20 % urea-PAGE) images showing HH_{min} reaction inside HH_{min}/peptide coacervate systems after 6 hours. Reaction in the dilute phase (left panel) refers to the supernatant separated from the reaction mixture after centrifugation and mixed with an equal volume of 2x loading dye before running on the gel. The reaction mixture from the dilute phase (5x diluted) (middle panel) and the reaction mixture from the coacervate phase (right panel) were diluted five times and run on the gel. The gel shows negligible concentration

of FAM-substrate, product and HH_{min} (with 10% FAM-HH_{min}) in the dilute phase compared to the coacervate phase when diluted in the same ratio. We observed product band intensity in the undiluted dilute phase (left panel). To determine the amount of product in the dilute phase for P3 and P6, we used ImageJ to determine the fluorescence intensity of the product band. We calculated the fraction of product ($F_{\text{pdt}}^{\text{dil}}$) in the dilute phase (details in the method). We observed for the P-3 dilute phase product fraction was 9% of the total product and for the P-6 dilute phase product fraction was 13% of the total product.

And amended the text in the manuscript in the following places:

To determine whether these observed effects could be attributed to the coacervate, we measured the content of the coacervate pellet and the supernatant after 6 hours of incubation. In comparison to our previous experiments, we centrifuged and separated the sample immediately after the substrate was added to the coacervate dispersions. Gel electrophoresis was undertaken from 3 μL of the supernatant and 3 μL of the coacervate pellet and showed presence of product in both the coacervate pellet and within the supernatant. ImageJ was used to obtain the band intensity of the product and to determine the relative difference of product (RDp) within the pellet vs the supernatant. Our results (Supplementary figure 5) showed that in all the peptide systems more product was produced in the pellet compared to the supernatant, as much as 50x more in P-1 coacervates and as little as 5x in P-6. Given that the pellet contains both residual supernatant and coacervate phase (see Materials and Methods), this relative difference can be attributed to the small volume of the coacervate phase.

Our results indicate the rate constant of the ribozyme is regulated by the coacervate dispersion. As coacervate droplets equilibrate molecules based on an intrinsic partition coefficient, this result could be attributed to a) localized reaction within the coacervate or b) RNA produced in the supernatant followed by partitioning into the coacervate. Due to the effect of partitioning, it can be challenging to determine where the reaction is taking place. To this end, we performed a simulation to predict the amount of product that could be produced under "scenario b" where the reaction takes place only in the supernatant phase and the product will partition into the coacervate phase to maintain its equilibrium partition coefficient (see supplementary note 2). The simulation predicts orders of magnitude lower amounts of product than what we observe in our experiments (Supplementary figure S6). Together, our results and simulations suggest that the product is being produced inside and outside of the droplet with a significant contribution coming from ribozyme cleavage within the droplet.

Given that we typically measured 3 μL of pellet from 10 μL of dispersion to obtain the rate constant, this could lead to an underestimation of the amount of product RNA that is being produced due to product partitioning from the droplet into the outer aqueous dispersion. Indeed, comparisons of the reaction rate from 3 μL of pelleted coacervate phase compared to the whole coacervate dispersion show that in all of the peptide coacervates there was an underestimation of the reaction rate within the coacervate pellet compared to the whole dispersion (Supplementary Figure 5B). To determine the approximate error in our measurement by gel electrophoresis, we determined the amount of RNA (ribozyme, substrate and product) within the supernatant after 6 hours of incubation. To do this, we compared the band intensity of the RNA in the coacervate and dilute phases after the addition of substrate and six hours of incubation (Supplementary Figure S5). The results showed that in the dilute phase, which had been diluted to the same amount as the pellet (5x dilution), there was a very faint band associated to the product for P-3 and P-6. However, in the undiluted supernatant, we observed product band intensity associated with P-1, P-3, P-5 and P-6 systems and no observable substrate/product bands associated with any other systems. Calculation of the fraction of product ($F_{\text{pdt}}^{\text{dil}}$, see supplementary methods) in the dilute phase showed that it contained $9.4 \pm 2\%$ and $13.2 \pm 0.2\%$ of the total product for P-3 and P-6 systems, respectively. Given that the relative percentage of product (~10%) in the supernatant is comparable to the variation in total product in the pellet (with P-3 and P-6) between experimental replicates, we find that the underestimation of the product formation (%) is small.

The key observation from our results is that coacervates prepared from different peptides can demonstrate variability in the apparent rate of the ribozyme within the coacervate dispersions. It is important to note that we measure the apparent rate constant for hammerhead ribozyme activity. Due to the differences in the microenvironment of the reactants between the coacervate and dilute phases, there can be two different rates of the hammerhead ribozyme activity, and by extension, two rate constants, one outside (k_{out}) and one inside k_{in} . The apparent rate constant (k) that we measure would be a linear combination of these two rates and would include all phenomenon such as diffusion outside, inside and across the phase boundary and the reaction rates inside, outside, and if relevant, at the interface. Due to small volumes of total coacervate phase it can be extremely challenging to determine the extent of the reaction within the coacervate droplet. Despite this, our simulation provides strong evidence that the majority of the reaction takes place within the coacervate and our experimental results

show that the presence of the coacervate phase leads to increased amounts of product relative to the supernatant and this varies depending on the peptide. This is likely attributed to the coacervate providing a source of RNA by partitioning. The increased concentration of substrate is available for cleavage within the coacervate droplet and outside of the droplet where the flux of the RNA allows the free exchange of substrate and product across the interface. Partitioning of the RNA product into the supernatant will lead to an underestimation of approximately 9-13% of the measured rate constants.

Further, we have amended the text and interpretation from the FRET study:

Furthermore, simultaneous detection of the substrate and product within micron-sized droplets are consistent with the results obtained from gel electrophoresis analysis and simulations which indicate that product within the coacervate droplet could be predominantly attributed to compartmentalized ribozyme activity. (Please also see supplementary note 2).

...and that product formation is driven by active ribozyme. Furthermore, between peptide systems, we observe differences in substrate transport within the droplets. Such variable substrate diffusion within the coacervate droplet

Additional information for the materials and methods:

either incubated at room temperature without further agitation or immediately subjected to centrifugation. In the latter case, 3 μ L and then 4 μ L of the supernatant was removed. 3 μ L of the supernatant and the remaining 3 μ L of the coacervate pellet was incubated for 6 hours. For samples which did not undergo immediate centrifugation, seven sets of samples (HH_{min} mixed with seven peptides) with eleven different time points (2 min, 5 min, 10 min, 15 min, 20 min, 30 min, 45 min, 60 min, 2 hours, 4 hours and 6 hours) were prepared in individual tubes. After each time point, the centrifuge tube was quickly spun down at 1000-2000 rpm for 30 seconds and 7 μ L supernatant was withdrawn and removed. To the remaining 3 μ L which contained the total condensed phase and some supernatant, 15 μ L of 2x RNA loading dye mixed with 20 mM Sodium hexametaphosphate was added to quench the reaction and an additional 10 μ L buffer solution was added to dilute it further. The solution was stored at -20 °C until analysis by gel electrophoresis. For determining the reaction kinetics in the whole solution, 10 μ L (250 μ M HH-ribozyme, 500 μ M of peptide) of coacervate dispersion was prepared. After the addition of substrate (50 μ M) the samples were incubated. After each time point (2 min, 7 min, 15 min, 30 min, 60 min, 2h, 4h, 6h) 10 μ L of 2x of RNA loading dye with 20 mM sodium hexametaphosphate) was added to the mixture and an addition 20 μ L of 50% glycerol buffer was used to dilute the sample and the solution stored at -20 °C until analysis by gel electrophoresis.

New supplementary note 2:

Supplementary information note 2: Confirmation of product generation in the dense phase.

Due to the intrinsic challenges in determining the location of a reaction within a phase separated droplet we opted to use a theoretical approach in combination with our experimental data to indirectly ascertain the location of the reaction. Here, we describe a model to describe a scenario where the product is exclusively produced in the dilute-phase portion of the pellet fraction of a sample. We refer to this model as "Scenario B" as an alternative to Scenario A where the reaction takes place in both phases present in the pellet i.e. the coacervate phase and the dilute phase. The Scenario B model gives an estimate of the maximum amount of product that could be generated if the reaction was restricted to the dilute phase only (Figure S6).

Within this note, we provide a description of the model and define the key parameters. We then specify how parameter values and initial conditions were determined. We also comment on the rationality of the parameter choices and the approximations made.

In summary, comparing the results from a Scenario B simulation to our experimental results, the amount of product in Scenario B is less in all peptide systems compared to the experimental data

(Figure S6b). Where the observed amounts exceed the predictions of Scenario B by several-fold and provides a strong indication that the ribozyme is catalytically active in the dense phase.

Figure S6: **A.** Schematic illustrating the essential features of the Scenario B model. Condensed-phase HH_{min} is catalytically inactive while HH_{min} in the dilute phase converts substrate to product with rate constant $k_{I,eff}$. Substrate and product exchange between condensed and dilute phases with rate constant k_{ex} . **B.** Comparison of product amounts in the pellet observed experimentally (blue) to those predicted by simulation of the Scenario B model after 360 min (orange). In each case, the observed amounts exceed the predictions of Scenario B by several-fold. These data strongly suggest ribozyme is catalytically active in the dense phase.

The Scenario B model

In Scenario B, we model the conversion of substrate into product in the $V_{pel} = 3 \mu\text{L}$ “pellet fraction” obtained following centrifugation of a $V_{tot} = 10 \mu\text{L}$ sample immediately upon substrate addition. The pellet fraction is composed of two coexisting phases, the dilute and condensed phases, which we denote by roman numerals I and II, respectively. The volume of the condensed phase is given by

$$V_{II} = \phi_{II} V_{tot}, \quad (6)$$

where

$$\phi_{II} = \frac{c_{peptide}^{tot} - c_{peptide}^I}{c_{peptide}^{II} - c_{peptide}^I} \quad (7)$$

is the volume fraction of the condensed phase and is calculated from the concentrations of peptide in each coexisting phase as well as the total peptide concentration averaged over the entire $10 \mu\text{L}$ sample.

The volume of dilute phase in the pellet is

$$V_I^* = V_{pel} - V_{II}. \quad (8)$$

Note that this is distinct from the volume of dilute phase (V_I) in the system as a whole as we consider only $3 \mu\text{L}$ of the total volume that contains the condensed coacervate phase and dilute phase,

$$V_I = V_{tot} - V_{II}. \quad (9)$$

Under Scenario B, the reaction $HH_{min} + Sub \rightarrow HH_{min} + Prod$ is assumed to take place exclusively in the dilute phase. Additionally, substrate and product molecules are assumed to move between phases in order to satisfy the equilibrium partition coefficients for each species i ,

$$P_i^{eq} = \frac{c_{i,eq}^{II}}{c_{i,eq}^I}. \quad (10)$$

Thus, Scenario B models the dynamics of the substrate and product concentrations in each phase (4 concentrations in total) subject to two processes: reaction in the dilute phase and molecular exchange between phases.

Here, we describe these dynamics with a master equation formalism by the following set of 4 ordinary differential equations:

$$\frac{dc_{sub}^I}{dt} = -k_I c_{HHmin}^I c_{sub}^I + \frac{\Delta N_{sub}}{V_I^*} k_{ex} \quad (11.1)$$

$$\frac{dc_{sub}^{II}}{dt} = -\frac{\Delta N_{sub}}{V_{II}} k_{ex} \quad (11.2)$$

$$\frac{dc_{prod}^I}{dt} = +k_I c_{HHmin}^I c_{sub}^I + \frac{\Delta N_{prod}}{V_I^*} k_{ex} \quad (11.3)$$

$$\frac{dc_{prod}^{II}}{dt} = -\frac{\Delta N_{prod}}{V_{II}} k_{ex}, \quad (11.4)$$

where k_I is the reaction rate constant in the dilute phase, c_{HHmin}^I is the ribozyme concentration in the dilute phase, and k_{ex} characterizes the rate of molecular exchange between phases. k_I , c_{HHmin}^I , and k_{ex} are assumed here to be constant in time. ΔN_{sub} and ΔN_{prod} represent the number of molecules needed to move from the condensed phase to the dilute phase in order to satisfy the equilibrium partition coefficients of each species, and these numbers change with time as the reaction proceeds. At each timepoint, ΔN_{sub} and ΔN_{prod} satisfy the relation

$$\frac{c_i^{II}(t) - \frac{\Delta N_i}{V_{II}}}{c_i^I(t) + \frac{\Delta N_i}{V_I^*}} = P_i^{eq}, \quad (12)$$

where i indexes the substrate and product species. Rearranging to solve for ΔN_i gives

$$\Delta N_i = \frac{c_i^{II}(t) - c_i^I(t) P_i^{eq}}{\frac{1}{V_{II}} + \frac{P_i^{eq}}{V_I^*}}. \quad (13)$$

Finally, the Scenario B expectation for the number of product molecules in the pellet fraction at time t is calculated as $N_{prod}^{Scen.B}(t) = V_I^* c_{prod}^I(t) + V_{II} c_{prod}^{II}(t)$.

The dynamics of equations 11.1-11.4 were solved numerically via forward integration in MATLAB using ode45.m given suitable parameter values and initial conditions. Below, we describe our process for selecting parameter values and initial conditions. In most cases, these values reflect constraints obtained directly from experimental design or data.

Determination of parameter values

This model for Scenario B includes 7 explicit parameters: k_{ex} , k_I , c_{HHmin}^I , P_{sub}^{eq} , P_{prod}^{eq} , V_I^* , and V_{II} . We determined numerical values for these parameters as follows, using experimental data when possible.

Volumes V_I^* , and V_{II} :

V_{II} is given by equations 6-7 using $c_{peptide}^{tot} = 500 \mu\text{M}$, $c_{peptide}^I$ and $c_{peptide}^{II}$ as determined from dilute phase and QPI analysis (as described previously) respectively, and $V_{tot} = 10 \mu\text{L}$. With V_{II} in hand, V_I^* is given by equation 8 where $V_{pel} = 3 \mu\text{L}$. The volumes V_I^* and V_{II} are reported for each system in Table S3.

Equilibrium partition coefficient P_{sub}^{eq} :

To estimate the equilibrium partition coefficient of the substrate between the dense and dilute phases (equation 10) we estimate the substrate concentration in the condensed phase at early time points in the reaction. Given that 3 μL of pellet contains both the condensed phase and the dilute phase, we estimate P_{sub}^{eq} using gel electrophoresis data of the pellet at $t=360$ min and mass conservation of the phase volumes and an estimate of the substrate concentration in the dilute phase from supernatant gel data.

First, we measure the integrated intensities (after background subtraction) of the substrate and product bands in each lane of each gel. These values are proportional to the number of molecules of each species present at $t = 360$ min:

$$\text{int}I_i^\alpha = \theta N_i^\alpha \quad (14)$$

where α indexes the pellet and supernatant fractions, i indexes the substrate and product species, $\text{int}I$ is the integrated intensity, N is the number of molecules, and θ is the proportionality constant that depends on the fluorophore and imaging conditions. Since both the substrate and product carry the same FAM fluorophore, and since both the pellet and supernatant fractions were imaged on the same gel, θ is the same for all 4 variants of equation 14. As the reaction converts substrate into product, the number of substrate molecules observed at $t = 360$ min is lower than the amount present when the supernatant and pellet fractions were separated at the beginning of the reaction.

We note that, in each case, the number of ribozyme molecules in the supernatant fraction is significantly larger than the number of product molecules observed after 6 hrs (Table S4). This suggests that product inhibition is unlikely to contribute significantly to the kinetics over this timescale, and validates our choice to neglect product inhibition in the model used to describe kinetics in the supernatant equation 27.

To estimate the number of substrate molecules present at the time of separation we approximate the number of product molecules at time zero

$$N_{sub}^\alpha(0) \approx N_{sub}^\alpha(360) + N_{prod}^\alpha(360). \quad (15)$$

Given that the reaction proceeds on the hours-timescale and the separation via centrifugation requires only a few minutes, this approximation appears reasonable. Values of $N_{sub}^\alpha(0)$ are reported for each system in Table S4.

Next, we estimate the number of substrate molecules present in the condensed phase at early times, $N_{sub}^{II}(0)$. From mass conservation,

$$N_{sub}^{pellet}(0) = N_{sub}^{I^*}(0) + N_{sub}^{II}(0), \quad (16)$$

where I^* and II refer to the dilute- and condensed-phase portions of the pellet fraction, respectively. At the time of separation, we assume that the concentration of substrate in the dilute-phase portion of the pellet fraction is equal to the concentration of substrate in the supernatant fraction because both consist of pure dilute phase:

$$\frac{N_{sub}^{I^*}(0)}{V_i^*} \equiv c_{sub}^{I^*}(0) = c_{sub}^I(0) = c_{sub}^{super}(0) \equiv \frac{N_{sub}^{super}(0)}{V_{super}}, \quad (17)$$

where $V_{super} = 3 \mu\text{L}$ is the volume of the supernatant fraction loaded onto the gel. Consequently, the number of molecules in the dilute-phase portion of the pellet fraction is

$$N_{sub}^{I^*}(0) = \frac{V_i^*}{V_{super}} N_{sub}^{super}(0) = \left[1 - \phi_{II} \left(\frac{V_{tot}}{V_{pellet}} \right) \right] N_{sub}^{super}(0). \quad (18)$$

Combining equations 18 and 20, the number of substrate molecules in the condensed phase can be expressed in terms of the substrate amounts in the pellet and supernatant fractions at early times as

$$N_{sub}^{II}(0) = N_{sub}^{pellet}(0) - \left[1 - \phi_{II} \left(\frac{V_{tot}}{V_{pellet}}\right)\right] N_{sub}^{super}(0). \quad (19)$$

Next, we calculate the ratio of substrate concentrations in the condensed and dilute phases at early times. To obtain concentrations, we divide the number of substrate molecules in each phase in the pellet fraction by the respective volumes of those phases in the pellet fraction. For the condensed phase, this yields

$$c_{sub}^{II}(0) = \frac{N_{sub}^{II}(0)}{V_{II}} = \frac{N_{sub}^{pellet}(0) - \left[1 - \phi_{II} \left(\frac{V_{tot}}{V_{pellet}}\right)\right] N_{sub}^{super}(0)}{\phi_{II} V_{tot}}. \quad (20)$$

For the dilute phase, equation 17 gives

$$c_{sub}^I(0) = c_{sub}^{super}(0) \equiv \frac{N_{sub}^{super}(0)}{V_{super}}. \quad (21)$$

The ratio of substrate concentrations in the two phases at early times is then given by

$$\Omega_{sub} \equiv \frac{c_{sub}^{II}(0)}{c_{sub}^I(0)} = \left[\frac{N_{sub}^{pellet}(0)}{N_{sub}^{super}(0)} - 1 + \frac{\phi_{II} V_{tot}}{V_{pellet}} \right] \left(\frac{V_{super}}{\phi_{II} V_{tot}} \right). \quad (22)$$

Recalling Equation 14, this can also be written explicitly in terms of the integrated intensities measured from the gels as

$$\Omega_{sub} = \left[\frac{int_{sub}^{pellet}}{int_{sub}^{super}} - 1 + \frac{\phi_{II} V_{tot}}{V_{pellet}} \right] \left(\frac{V_{super}}{\phi_{II} V_{tot}} \right). \quad (23)$$

Values of Ω_{sub} are reported for each system in Table S5.

Finally, we use this concentration ratio as an estimate for the equilibrium partition coefficient of the substrate between the two phases:

$$P_{sub}^{eq} \cong \Omega_{sub}. \quad (24)$$

This approximation is best in the limit where the timescale for substrate partitioning to equilibrate is shorter than timescale for the separation of the pellet and supernatant fractions, and both are much shorter than the characteristic timescale of the reaction, i.e.

$$\tau_{partitioning} < \tau_{centrifugation} \ll \tau_{reaction}. \quad (25)$$

In our systems, we anticipate that $\tau_{partitioning}$ is limited primarily by the rate of diffusion in the condensed-phase. Our FRET experiments (Figure 3) demonstrate that this timescale varies between different peptide/HH_{min} systems and can be on the order of 10s of minutes. This indicates that substrate partitioning may not yet have equilibrated by the time the supernatant and pellet fractions are separated. Since the substrate is predominantly in the dilute-phase upon addition to the sample, Ω_{sub} will lie below the true value of P_{sub}^{eq} in the slow-partitioning limit. Intuitively, this is because the concentration of substrate in the condensed phase begins near zero and not enough time has elapsed for it to rise to the equilibrium level throughout the entire condensed phase.

To assess the impact of deviations of the partition coefficients on our conclusions, we ran simulations with 10-fold higher partitioning (Figure S5). In each case, we found that increasing the partition coefficient reduces the amount of product generated through Scenario B. So, while our estimate of P may lie below the true equilibrium partition coefficient, using a more accurate (i.e. larger) value further reduces product formation relative to our prediction. Our prediction is an upper-bound on the contribution of Scenario B to overall production.

Equilibrium partition coefficient P_{prod}^{eq} :

The estimation of the equilibrium partition coefficient for the product is given by

$$P_{prod}^{eq} \cong \frac{\Omega_{sub}}{2}, \quad (26)$$

reported for each peptide-coacervate system in Table S5. This choice (equation 26) is motivated by the observation that the partition coefficient generally increases with chain length for strong polyelectrolytes in coacervate-forming systems^{1, 2}, given that the product is about half the size of the substrate, we use equation 21 to provide a qualitative estimate of the relationship between the partitioning of the substrate and product. Whilst it is unlikely that the product partitioning is half of the substrate, the particular choice of P_{prod}^{eq} does not impact the conclusions we draw from the Scenario B model. This is because the reaction rate is assumed to be independent of the local product concentration, (i.e. no product inhibition). As a result, the reaction terms in Eq. (11.1-11.4) are independent of local product concentrations and therefore independent of P_{prod}^{eq} . Since the number of ribozyme molecules in the dilute-phase portion of the pellet is, in most cases, less than the total amount of product observed within the pellet fraction (Table S4), product inhibition could influence the rate of product formation in the dilute phase. However, we note that incorporating product inhibition into the model would further reduce the product generated by Scenario B and would not increase it. Neglecting product inhibition is thus consistent with our use of the Scenario B model to place an upper-bound on product formation in the dilute-phase portion of the pellet fraction.

Effective rate constant $k_{I,eff} = k_I \times c_{HH_{min}}^I$:

Note that the parameters k_I and $c_{HH_{min}}^I$ appear in the model only as the product $k_I c_{HH_{min}}^I = k_{I,eff}$. It is therefore sufficient to specify this product alone, from which the rate constant k_I can be determined using measured values of $c_{HH_{min}}^I$ (Figure 4B) if desired.

For each peptide/HH_{min} system, we estimate the effective dilute-phase rate constant $k_{I,eff}$ by fitting a minimal kinetic timecourse of product formation in the supernatant fraction. We assume that the supernatant fraction is an homogeneous solution without any dense phase. We model the HH_{min}-catalyzed conversion of substrate to product in the homogeneous supernatant fraction as a single-step reaction governed by the master equations

$$\frac{dc_{sub}^{super}}{dt} = -k_I c_{HH_{min}}^I c_{sub}^{super} = -k_{I,eff} c_{sub}^{super} \quad (27.1)$$

$$\frac{dc_{prod}^{super}}{dt} = +k_I c_{HH_{min}}^I c_{sub}^{super} = +k_{I,eff} c_{sub}^{super}. \quad (27.2)$$

Following integration, this pair of equations has the simple set of solutions

$$c_{sub}^{super}(t) = c_{sub}^{super}(0) e^{-k_{I,eff} t} \quad (28.1)$$

$$c_{prod}^{super}(t) = c_{prod}^{super}(0) + c_{sub}^{super}(0) [1 - e^{-k_{I,eff} t}]. \quad (28.2)$$

We specify the initial conditions by approximating the product concentration at the time the supernatant fraction is separated from the pellet fraction to be zero,

$$c_{prod}^{super}(0) = \frac{N_{prod}^{super}(0)}{V_{super}} \approx 0, \quad (29)$$

implying via Equation 15 that

$$c_{sub}^{super}(0) = \frac{N_{sub}^{super}(0)}{V_{super}} \approx \frac{N_{sub}^{super}(360) + N_{prod}^{super}(360)}{V_{super}}. \quad (30)$$

Finally, we determine $k_{I,eff}$ by fitting the normalized product concentration

$$y(t) = c_{prod}^{super}(t) / c_{sub}^{super}(0) \quad (31)$$

at two timepoints, $t = 0$ (assumed zero) and 360 min (determined from the gel), to the normalized version of Equation 30.2 with the initial conditions (31-32)

$$y(t) = [1 - e^{-k_{I,eff}t}]. \quad (32)$$

The fits are shown in Figure S7, and the values of $k_{I,eff}$ obtained through this fitting procedure are reported in Table S6.

Constant characterizing rate of molecule exchange between phases k_{ex} :

This is a phenomenological parameter used to characterize the partitioning process in a simple way. In general, we expect that the timescale for a molecular species to equilibrate concentrations between two phases depends on the size distribution of coacervate droplets present in the sample as well as the diffusion coefficient of the species in the coacervate phase. For simplicity, we do not attempt to model these dependencies explicitly or to account for system-specific differences in the exchange kinetics beyond the level of differences in the total volume of condensed phase. Rather, we set a relatively fast value of $k_{ex} = 1/\text{min}$ as a default value for all peptide/HH_{min} systems. Depending on the system, this value is ~1000-fold to 30,000-fold larger than $k_{I,eff}$, corresponding to a limit in which $\tau_{partitioning} \ll \tau_{reaction}$. As mentioned above, we observe the partitioning timescale can be on the order of 10s of minutes (Figure 3), suggesting that a rate closer to that of $k_{I,eff}$ may be more accurate.

To assess the impact of the fast exchange assumption on our conclusions, we also ran simulations with 10-fold and 1000-fold lower values of k_{ex} . In each case, we found that decreasing k_{ex} reduces the amount of product generated through Scenario B (Figure S9). We note that the dependence of the product amount generated at $t = 6$ hrs is very weak. Over the range of parameters examined here, this indicates that the amount of product generated through Scenario B is much less sensitive to the choice of k_{ex} than to the choice of P_{sub}^{eq} (Figure S9). Taken together, these observations indicate that our choice of $k_{ex} = 1/\text{min}$ is again consistent with our use of the Scenario B model to place an upper-bound on product generation in the dilute-phase portion of the pellet fraction.

Determination of initial conditions for Scenario B model

To be fully-specified, the Scenario B model given in Equations 11.1-11.4 requires initial conditions for each of the four dynamical variables c_{sub}^I , c_{sub}^{II} , c_{prod}^I , and c_{prod}^{II} . Consistent with the approximations used to determine parameter values and discussed above, we approximate the number of product molecules in the pellet at time $t = 0$ to be zero, such that Equation 15 holds. $c_{sub}^I(0)$ and $c_{sub}^{II}(0)$ are thus given by Equation 23 and 22, respectively, while $c_{prod}^I(0) = c_{prod}^{II}(0) = 0$.

Conversion between integrated intensity on the gel and number of molecules, θ :

Aside from assessing the potential impact of product inhibition, none of the analysis described above nor conclusions drawn from it requires precise knowledge of θ , defined in Equation 14. In particular, the Scenario B master equations are readily re-written in terms of dynamical integrated intensities instead of concentrations and lead to equivalent conclusions.

That said, we chose to report amounts in units of picomole rather than integrated intensity owing to the former's clear physical meaning and potential transferability beyond the current work. We determined θ for both HH_{\min} and substrate as

$$\theta_i = \frac{\langle intI_i^{tot} \rangle_n}{c_i^{tot} v_{tot} (q_i^{tot})^{-1}} \quad (33)$$

where c_i^{tot} is the total concentration of species i present in the system (in units of charge equivalents; 50 μM for substrate, 250 μM for HH_{\min}), q_i^{tot} is the formal charge of species i , $intI_i^{tot}$ is the total integrated intensity expected for the species in the entire system at $t = 0$ given as

$$intI_i^{tot} = \begin{cases} intI_{HH_{\min}}^{pel} + \left(\frac{v_{tot} - v_{pel}}{v_{super}} \right) intI_{HH_{\min}}^{super}, & \text{for } i = HH_{\min} \\ (intI_{sub}^{pel} + intI_{prod}^{pel}) + \left(\frac{v_{tot} - v_{pel}}{v_{super}} \right) (intI_{sub}^{super} + intI_{prod}^{super}), & \text{for } i = sub \end{cases} \quad (34)$$

and $\langle \dots \rangle_n$ denotes an average over the $n = 7$ peptide/ HH_{\min} systems. We found $\theta_{sub} = 39.3 \pm 4.8$ and $\theta_{HH_{\min}} = 5.7 \pm 1.4$. The ~ 10 -fold lower value for $\theta_{HH_{\min}}$ reflects the different fractions of FAM-labeled molecules in the experiment ($f_{HH_{\min}}^{label} = 0.1$, while $f_{sub}^{label} = 1$).

Figure S7: Fit to kinetics in supernatant fraction for Scenario B model. a, Minimal time-courses (circles) constructed from electrophoresis measurements at $t = 360$ min (Figure S5A) and the assumption of zero product at $t = 0$. Solid curves represent fits to Equation 32. The only free parameter is the effective reaction rate constant, whose values are reported in Table S6.

Figure S8: Scenario B model of the kinetics in the pellet fraction. A Simulated timecourses of reaction and partitioning for Scenario B model ($k_{ex} = 1$, other parameters as in Tables S3,5,6) for each peptide/HH_{min} system. Yellow curve is the sum of the other four. **B** Partitioning dynamics from the simulations in Figure S6. While substrate partitions at equilibrium levels throughout, equilibration of product partitioning proceeds with a delay owing to its initial absence from the system.

Figure S9: Sensitivity of Scenario B model output to parameters. Comparison of product predictions after 360 min from Scenario B models upon variation of the least-certain model parameters: the exchange rate (k_{ex}) and equilibrium partition coefficients (P_i^{eq}) for the substrate and product. Partition coefficients are quoted as multiples of the concentration ratios between pellet and supernatant fractions, Ω , values of which are listed in Table S5. The model shown previously in Figure S6b with $k_{ex} = 1$ and $P_i^{eq} = \Omega_i$ (red-orange) is reproduced here for comparison. For fixed partition coefficient, product amounts decrease slightly as the exchange rate is reduced 1000-fold. For fixed exchange rate, product amounts decrease ~ 8 -fold upon 10-fold increase in equilibrium partition coefficient. These results suggest that revising parameter values to more accurately reflect the timescale of partitioning will tend to reduce the generation of product under Scenario B. In this sense, the parameters used in Figure S8 serve as an upper-bound on product generation under Scenario B.

Table S3 – Volume parameters in Scenario B model

System	V_{II} (μL)	V_I^* (μL)
P-1/HH _{min}	0.00184	2.99816
P-2/HH _{min}	0.00186	2.99814
P-3/HH _{min}	0.00221	2.99779
P-4/HH _{min}	0.00179	2.99821
P-5/HH _{min}	0.00219	2.99781
P-6/HH _{min}	0.00188	2.99812
P-7/HH _{min}	0.00149	2.99851

Table S4 – Amounts in gel from Fig. S5A^a

System	Pellet (pmol)				Supernatant (pmol)			
	$N_{HH_{min}}^{pel}$	$N_{Sub}^{pel}(36)$	$N_{prod}^{pel}(360)$	$N_{Sub}^{pel}(0)$	$N_{HH_{min}}^{super}$	$N_{Sub}^{super}(36)$	$N_{prod}^{super}(360)$	$N_{Sub}^{super}(0)$
P-1/HH _{min}	61.28	25.57	11.70	37.27	0.726	1.661	0.296	1.957
P-2/HH _{min}	82.87	29.49	1.54	31.03	0.417	1.648	0.081	1.728
P-3/HH _{min}	57.39	21.61	13.69	35.29	1.163	3.353	0.702	4.055
P-4/HH _{min}	59.85	29.80	2.55	32.35	0.701	2.485	0.166	2.651
P-5/HH _{min}	81.92	27.02	5.91	32.94	1.193	2.801	0.190	2.991
P-6/HH _{min}	33.60	22.61	8.33	30.95	3.972	6.187	2.386	8.573
P-7/HH _{min}	50.10	30.57	1.16	31.73	1.136	3.758	0.048	3.806

^a All quantities listed in units of pmol polymer. Numbers in parenthesis indicates the timepoint (in minutes) to which the value applies. Deviations of the sum $N_{sub}^{pel}(360) + N_{prod}^{pel}(360)$ from $N_{sub}^{pel}(0)$ within a row represent rounding error.

Table S5 – Partition coefficient estimates for substrate and product in Scenario B model^a

System	Ω_{sub}	Ω_{prod}
P-1/HH _{min}	29396	14698
P-2/HH _{min}	27372	13686
P-3/HH _{min}	10462	5231
P-4/HH _{min}	18788	9394
P-5/HH _{min}	13685	6843
P-6/HH _{min}	4158	2079
P-7/HH _{min}	14749	7375

^a All values are dimensionless. Partition coefficients for substrate are estimated as equal to the concentration ratios calculated from Equation 24. Partition coefficients for product are estimated as $\frac{1}{2}$ the value for substrate in the same system. Deviations of Ω_{prod} from $\Omega_{sub}/2$ represent rounding error.

Table S6 – Summary of fits to reaction kinetics in supernatant used in Scenario B model^a

System	$k_{I,eff}$ (min ⁻¹)	$\delta k_{I,eff}$ (min ⁻¹)
P-1/HH _{min}	4.555E-04	1.140E-08
P-2/HH _{min}	1.328E-04	1.275E-08
P-3/HH _{min}	5.280E-04	2.700E-10
P-4/HH _{min}	1.796E-04	1.840E-07
P-5/HH _{min}	1.822E-04	4.552E-09
P-6/HH _{min}	9.061E-04	3.975E-09
P-7/HH _{min}	3.525E-05	3.911E-07

^a As each fit is to a two-point time-course, the fit Adj. $R^2 = 1$ in each case. Correspondingly, the quoted uncertainty in effective dilute-phase rate constant, $\delta k_{I,eff}$, reflects numerical precision of the fitting rather than statistical uncertainty.

Comment: (2) - In a few cases, a discussion leaves out important information, resulting in a lack of credit given to other researchers, or in the impression that inconvenient, alternative explanations are not considered. L98-103: The sentences from "This system provides ... primitive cells" make it appear as if such work has not been done before - which would be incorrect. The work of Christine Keating and Phil Bevilacqua, among others, has already provided groundwork to this understanding. It is therefore necessary to better specify what is the new insight, and use a phrase such as 'expand our understanding' to make clear what work has been done before on peptide/RNA interactions and their effects on RNA catalysis in coacervates, and what is new in this study.

Thank you for this comment. To ensure that we have not left out important information or not credited previous studies we have included additional references, descriptions and explanations.

On the microscopic level, coacervates, can induce the secondary structure of peptides³ or alter the tertiary structure of RNA⁴ which can affect ribozyme activity. Indeed, previous studies have shown that increasing the length of the polyanion which forms the coacervate will increase the fraction of product generated by ribozymes⁵. Furthermore, it has been shown that changing the peptide sequence from polyarginine to polylysine and (RGG)_n and modifying the length of the peptide can lead to differences

in material properties of the droplet which then impact the ribozyme activity⁶⁻⁸. Oligopeptide based coacervates have additional advantages to homo (poly)peptide systems as the properties of the different amino acids can impart different physical coacervate environments. For example, increasing hydrophobic moieties in peptides will increase the hydrophobicity of the internal environment and partitioning of hydrophobic molecules into the core,⁹ while increasing the strength of the interactions will affect the diffusion dynamics and material properties of the droplet⁸.

With new references:

38. Cakmak, F.P., Choi, S., Meyer, M.O., Bevilacqua, P.C. & Keating, C.D. Prebiotically-relevant low polyion multivalency can improve functionality of membraneless compartments. *Nature communications* **11**, 1-11 (2020).
39. Poudyal, R.R. et al. Template-directed RNA polymerization and enhanced ribozyme catalysis inside membraneless compartments formed by coacervates. *Nature communications* **10**, 1-13 (2019).
40. Iglesias-Artola, J.M. et al. Charge-density reduction promotes ribozyme activity in RNA-peptide coacervates via RNA fluidization and magnesium partitioning. *Nat Chem* **14**, 407-416 (2022).
41. Baruch Leshem, A. et al. Biomolecular condensates formed by designer minimalistic peptides. *Nat Commun* **14**, 421 (2023).
42. Akahoshi, Y. et al. Phase-Separation Propensity of Non-ionic Amino Acids in Peptide-Based Complex Coacervation Systems. *Biomacromolecules* **24**, 704-713 (2023).
43. Baruch Leshem, A. et al. Biomolecular condensates formed by designer minimalistic peptides. *Nature Communications* **14**, 421 (2023).

Furthermore, alternative explanations for the observed variations in reaction rate have been provided. However, deconvoluting the effect of coacervates on enzymatic activity is non-trivial as coacervates have been shown to tune reactions in a variety of ways. For instance, a coacervates' ability to support primitive RNA reactions can lead to the slowing down of cleavage rates¹⁰, increased ligation yields¹¹, reduction in yields for template-directed polymerisation¹² and preferential ligation of linear RNA over circular RNA⁶.

We have added additional text at the end of the introduction. The text has been revised to provide a better description of the new insights from the study taking into account previous studies, from:

“This system provides a versatile and robust model for determining how peptide chemistry can change the physicochemical properties of the coacervate and how these subsequently impact reaction kinetics. Our study provides a physical framework for understanding how emergent properties of biological condensates can tune biochemistry within the cell and marks a step toward unravelling how coacervates formed within a prebiotic soup can affect compartment properties and reaction outcomes across a population of primitive cells. “

To:

We exploit RNA/peptide coacervates as a versatile and robust model for determining how amino acid sequence can change the physicochemical properties of the coacervate and how these subsequently impact reaction kinetics. Significantly, our study extends our understanding of the effect of compartmentalization on enzymatic reactions by correlating coacervate properties to ribozyme activity. Furthermore, it marks a step toward unravelling how coacervates formed within a prebiotic soup can affect compartment properties and reaction outcomes across a population of primitive cells.

Comment (3) - Especially in the abstract and introduction, the manuscript should describe more clearly what are the new findings of this study. Please see my summary for the findings I was able to identify.

We have included additional text to the abstract:

“a relationship between the physicochemical properties of the coacervate microenvironment and the catalytic activity of the ribozyme in the dispersion.”

Additional text has been included in the introduction:

“Specifically, we observe a strong negative correlation between the polymer volume fraction in the coacervate and the diffusion coefficient and a positive correlation between the reaction rate constant of the ribozyme and its’ diffusion coefficient. RNase footprinting assays show different binding modes of the peptide to the ribozyme depending on the peptide sequence. This suggests that the differences in interaction between RNA and each peptide sequence contribute to the observed differences in droplet properties and activities.”

“Significantly, our study extends our understanding of the effect of compartmentalization on enzymatic reactions by correlating coacervate properties to ribozyme activity.”

Comment (4) - Words and phrases are often used unconventionally. To make the content more clear (and thereby increase the manuscript's impact), it is important to use words and phrases in the usual way, and to simply sentences whenever possible. Some examples are below, and some additional, minor examples are given under 'minor comments': Throughout the manuscript, the term 'random' is used incorrectly; it should be replaced with the words 'arbitrary' or 'specific' (the word 'representative' may not match because six of the seven sequences are not designed to represent classes of sequences). While a random sequence generator was used to come up with six specific sequences this does not make the sequences random, it makes them arbitrary. In contrast, the word 'random' implies that the sequence of peptides is at least partially randomized is used - but the used seven peptides have defined sequences.

We thank the reviewer for this comment and have changed the word random to arbitrary to better describe the peptide sequences.

L67-76: The sentences from 'Despite' to 'by biology' describe that 'RNA-peptide interactions in coacervates are important to consider for the properties of the individual molecular species'; this could which could be expressed more clearly and succinctly in a single (or at most two) sentences.

Following the recommendation of the reviewer, we have shortened the text from:

“Despite the impact of coacervates in tuning reaction kinetics, there have been a limited number of studies that have focused on the potential effect of coacervation on the co-evolution of molecular species, where the capability to bring molecules in close proximity can change the physicochemical properties of the droplet that then impact molecular reactions housed within the compartment. For example, the emergent properties of the compartment are determined by molecular level interactions. Thus, interactions and properties on the molecular and microscopic scales could in principle provide a selection pressure on any compartmentalized reaction. One interesting example to consider is the co-evolution of RNA and peptides. Not only are RNA - peptide interactions prevalent in modern biological systems, their co-evolution could have been important during the origin of life to fuel the transition from chemistry to biology.”

To:

Despite the impact of coacervates in tuning reaction kinetics, there have been a limited number of studies^{13, 14} that have focused on the potential effect of coacervation on the co-evolution of molecular species. Molecular interactions dictate droplet properties on the microscopic and macroscopic scales that in turn impact compartmentalized reactions.

L91: The statement "coacervates prepared from ribozyme and each of the seven peptide sequences result in droplets with distinct chemical compositions." does not contain information: The compositions *are* 'ribozyme + peptide 1' vs. "ribozyme + peptide 2" etc.

Instead of 'chemical compositions, should it be 'physico-chemical properties'? Is the intention to say that each peptide, when mixed with the ribozyme, has its own optimal peptide - ribozyme stoichiometry (this would overlap with the information of the next sentence)?

We thank the reviewer for this comment. By “distinct chemical composition” we mean the identity as well as concentrations of components within the coacervate. The information in the next sentence refers to physico-chemical properties of the coacervate which are again different between the peptide coacervate systems. To improve clarity, we have modified the text from:

“We show that coacervates prepared from ribozyme and each of the seven peptide sequences result in droplets with distinct chemical compositions”

To:

We show that coacervates prepared from ribozyme and each of the seven peptide sequences result in droplets with distinct chemical compositions i.e. different peptide/ribozyme concentrations and stoichiometry;

And from: Further, we find differences in emergent properties of the coacervate microenvironments

To: Further, we find differences in the physico-chemical properties of the coacervate

L90-103: Consider re-writing the sentences from "We show ..." to "... primitive cells" much more concisely, with a focus on the gained new information, and insight. For example, these sentences in lines 90-103 could be replaced - without the loss of important information - much more concisely by "Coacervates prepared from ribozyme and each of the seven peptide sequences showed differences in the condensed-phase polymer volume fraction and molecular diffusivity, as well as ribozyme reaction rate and product yield. Peptide sequences with higher net charge led to denser packing of the ribozyme and peptide and slower ribozyme diffusion and slower reaction rates. This study expands our understanding of how coacervates formed within a prebiotic soup can affect the properties of compartments and catalysts within."

We thank the reviewer for this comment and have rewritten the text so that the new information is better articulated. Unfortunately given the previous comments by the reviewer which asks for more clarity in explanation within this paragraph it was not possible to make it more concise. However, we have modified the text as described below as recommended by the reviewer. The text is changed (L90-L103) from: We show that coacervates prepared from ribozyme and each of the seven peptide sequences result in droplets with distinct chemical compositions. Further, we find differences in emergent properties of the coacervate microenvironments including the condensed-phase polymer volume fraction and molecular diffusivity, as well as ribozyme reaction rate and product yield. Our results show that the differences in the droplet properties and activities of the droplet are underpinned by the interaction between the RNA and peptide. We find that the sequence with the higher net charge led to denser packing of the ribozyme and peptide and increased total polymer concentrations in the coacervate phase that was concomitant with slower ribozyme diffusion and slower reaction rates. This system provides a versatile and robust model for determining how peptide chemistry can change the physicochemical properties of the coacervate and how these subsequently impact reaction kinetics. Our study provides a physical framework for understanding how emergent properties of biological condensates can tune biochemistry within the cell and marks a step toward unravelling how coacervates formed within a prebiotic soup can affect compartment properties and reaction outcomes across a population of primitive cells.

To: We show that coacervates prepared from ribozyme and each of the seven peptide sequences result in droplets with distinct chemical compositions i.e. different peptide/ribozyme concentration and stoichiometry. Further, we find differences in the physico-chemical properties of the coacervate microenvironments including the condensed-phase polymer volume fraction and molecular diffusivity, as well as ribozyme reaction rate and product yield within the dispersion. Specifically, we observe a strong negative correlation between the polymer volume fraction in the coacervate and the diffusion

coefficient and a positive correlation between the reaction rate constant of the ribozyme and its' diffusion coefficient. RNase footprinting assays show different binding modes of the peptide to the ribozyme depending on the peptide sequence. This suggests that the differences in interaction between RNA and each peptide sequence contribute to the observed differences in droplet properties and activities. We find that the sequence with the higher net charge led to denser packing of the ribozyme and peptide and increased total polymer concentrations in the coacervate phase that was concomitant with slower ribozyme diffusion and slower reaction rates. It is important to note that the ribozyme activity was reduced by 2-3 orders of magnitude within the coacervate compared to free buffer.

We exploit RNA/peptide coacervates as a versatile and robust model for determining how amino acid sequence can change the physicochemical properties of the coacervate and how these subsequently impact reaction kinetics. Significantly, our study extends our understanding of the effect of compartmentalization on enzymatic reactions by correlating coacervate properties to ribozyme activity. Furthermore, it marks a step toward unravelling how coacervates formed within a prebiotic soup can affect compartment properties and reaction outcomes across a population of primitive cells.

L342: Consider removing unnecessary wording from the sentence "Despite having shown that the rate of the reaction for coacervate systems is driven by the emergent properties of the coacervate, it cannot be ignored that these properties could also be affected by the underlying molecular interaction between charged RNA and peptide that is driven by the RNA/peptide sequence." After reading this sentence five times I realized that its message was very simple, and it could be added as a side sentence to the first sentence of the next paragraph. Note that the sentence in L347 only makes sense of such a side sentence is included.

The text has been changed from: Despite having shown that the rate of the reaction for coacervate systems is driven by the emergent properties of the coacervate, it cannot be ignored that these properties could also be affected by the underlying molecular interaction between charged RNA and peptide that is driven by the RNA/peptide sequence. Therefore, we wanted to determine if there were differences in the molecular interaction between RNA and each of the peptides. To determine the effect of molecular interactions between the peptides and RNA, we used a RNase (ribonuclease) footprinting assay to identify the accessibility of specific areas to the ribozyme.

To: To determine whether the observed differences in the emergent properties of the coacervate systems were correlated to molecular level interactions we sought to identify differences in molecular interactions between the peptide and RNA. To do this

(5) The abstract and introduction claim that the used peptides are 'prebiotic' (e.g. line 25). However, there are two problems with this statement: First, we will never know whether certain peptides were 'prebiotic'; we can only assume that they were 'prebiotically plausible'. Second, the amino acid composition described in lines 112-114 is in my opinion not prebiotically plausible: All peptides contain more than 50% arginine / lysine, and neither of these two amino acids seem prebiotically plausible at a quantity required for the formation of such peptides. While two laboratories have devised a synthetic route where arginine could have been generated prebiotically, the expected concentrations would not be in a range that would allow the formation of peptides containing 50% of arginine or lysine. To address this issue, either an argument needs to be presented up-front (the latest in the first paragraph of the results section) that these peptides are indeed 'prebiotically plausible' or the claim of prebiotic plausibility is dropped and the peptides serve as model compounds that are closer to prebiotically plausible peptides than previous studies. I believe that the latter would be more appropriate.

We thank the reviewer for this comment and have included new references (48-51) which demonstrate the prebiotically plausible synthesis of lysine or arginine (as suggested by reviewer 3). The text has now been changed from: "12 prebiotic proteinaceous amino acids which have been proposed to be abundant during early life¹⁵⁻¹⁸"

To : "from 12 prebiotic proteinaceous amino acids to provide models of prebiotically plausible peptides⁴⁸⁻⁵¹"

New references:

48. Longo, L.M. et al. Primordial emergence of a nucleic acid-binding protein via phase separation and statistical ornithine-to-arginine conversion. *Proc Natl Acad Sci U S A* **117**, 15731-15739 (2020).
49. Plankensteiner, K., Reiner, H. & Rode, B.M. Amino acids on the rampant primordial Earth: electric discharges and the hot salty ocean. *Mol Divers* **10**, 3-7 (2006).
50. Thoma, B. & Powner, M.W. Selective Synthesis of Lysine Peptides and the Prebiotically Plausible Synthesis of Catalytically Active Diaminopropionic Acid Peptide Nitriles in Water. *J Am Chem Soc* **145**, 3121-3130 (2023).
51. Blanco, C., Bayas, M., Yan, F. & Chen, I.A. Analysis of evolutionarily independent protein-RNA complexes yields a criterion to evaluate the relevance of prebiotic scenarios. *Current Biology* **28**, 526-537. e525 (2018).

(6) L1: Please clarify the title. In the title, it is not clear what the term 'activity' refers to: droplets are usually not associated with an 'activity' (this word is mostly used in the context of catalysis). However, the manuscript describes effects on ribozyme activity and physico-chemical properties of the droplets.

We modified the title focusing more on ribozyme activity modulation by short peptide and coacervate composition. Title changed from: "RNA-peptide co-operativity tunes the activity of coacervate microdroplet dispersions". To "RNA-peptide cooperativity tunes ribozyme activity within coacervate microdroplets"

(7) The counter-intuitive finding that the ribozyme-catalyzed reaction is slower at higher ribozyme concentration in the coacervates needs to be critically evaluated, and alternative explanations discussed. For example, the same anti-correlation would be expected if a given substrate molecule in the droplet could be bound by the arms from two different hammerhead ribozymes at increased ribozyme concentration. That arrangement would not allow the formation of the hammerhead ribozyme's catalytic core, and thereby lead to lower activity.

We thank the reviewer for the comment. We agree that the ability for substrate molecules to bind to two different ribozymes, at high concentrations could lead to a reduction in activity especially as the ribozymes will be close proximity to each other within the coacervate. It is unclear whether this would happen as peptide-RNA interactions at the active site could inhibit this. However, given that the peptide concentration is higher than the ribozyme in the coacervate but the binding interaction is not 1:1 there could still be some available substrate binding sites on the ribozyme. We have revised the text to discuss alternative explanations for the observed anti-correlated behaviour between ribozyme rate and concentration.

Further, this decrease in rate with increasing ribozyme concentration, could be due to molecular or mesoscopic effects. For example, on the molecular level, RNA-RNA interactions driven by high local concentrations could lead to substrate binding between two neighboring ribozymes which reduces ribozyme activity. Increased packing (increased density and concentration) of the ribozyme within the coacervate can lead to changes in the secondary structure that could lead to loss of activity. In addition, peptide-RNA interactions could block RNA substrate binding sites. Here stronger binding between the ribozyme and peptide can lead to denser droplets with increased concentrations but reduced activity. On the mesoscopic level, molecular diffusion is slower in denser droplets, which can reduce reaction rates. Therefore, rate of ribozyme cleavage could be affected by the substrate partitioning and diffusion through the coacervate as well as to changes in ribozyme activity, for instance from interactions with peptide.

Minor comments:

L20: I think that there should be a 'they' inserted so that it reads "... biological molecules; they concentrate molecules ...".

The text has been changed to “to form from biological molecules, they concentrate molecules”

L41: would 'environment' be more clear than 'milieu'?

“Molecular milieu” is changed to “molecular environment”.

L43: I believe that it is 'cooperation' and not 'co-operation'.

“cooperation” is inserted in place of “co-operation”.

L44: convert to 'physico-chemical properties of the compartment such as the polymer volume fraction and ...'?

The sentence is now corrected from - Despite this, it is still not clear how molecular co-operation through non-covalent interactions can affect emergent physicochemical properties such as the polymer volume fraction of the compartment and how this impacts localized enzyme reactions.

To - Despite this, it is still not clear how molecular cooperation through non-covalent interactions can affect emergent physicochemical properties such as the polymer volume fraction and molecular diffusivities within the compartment nor how these properties impact localized enzyme reactions..

L49/50: the current description leaves it unclear what determines the formation of coacervates as opposed to larger aggregates / precipitation - because both processes can be a result of the stated polyelectrolytes. If this would be a more involved explanation then it would be helpful to include a statement such as 'the distinction between the formation of small coacervate droplets and large aggregates or precipitates depends on subtle changes in molecular properties and solution conditions, as described in [reference].'

We thank the reviewer for this comment and have slightly modified the suggested text and included it into the manuscript. Furthermore, we include the word “complex” to coacervation to add precision into the phase separation process.

Complex coacervation, the physical phenomenon of associative phase separation between two oppositely charged polyelectrolytes in solution, provides tractable models for membrane-free compartmentalization for *in-vitro* studies. The formation of liquid droplets as opposed to precipitates is dependent on factors such as complexation and solvation properties^{4,6}.

4. Chollakup, R., Smitthipong, W., Eisenbach, C.D. & Tirrell, M. Phase Behavior and Coacervation of Aqueous Poly(acrylic acid)–Poly(allylamine) Solutions. *Macromolecules* **43**, 2518-2528 (2010).
5. Perry, S.L. et al. Chirality-selected phase behaviour in ionic polypeptide complexes. *Nature Communications* **6** (2015).
6. Priftis, D. & Tirrell, M. Phase behaviour and complex coacervation of aqueous polypeptide solutions. *Soft Matter* **8**, 9396-9405 (2012).

L53: "there is an increasing number of studies" (singular).

Grammatically, this could go either way, we have opted to remain with “are” to coincide with the word “studies”

L62: The words 'in an origin of life context' can be removed to simplify the sentence. This works because the following statements are general statements and not restricted to an OoL scenario.

The words 'in an origin of life context' have been removed in the modified text as suggested.

L67-70: The statement should be supported by citing those studies.

The citations had already been included and placed when the specific examples were described. For clarity we have now included some of the same references after “there have been a limited number of studies^{33,34}”

L87: what does the word 'chemistry' mean here? Does it mean 'chemical reactions'. Similarly, L310 & L311 would be better served by a different word such as 'amino acid sequence' or 'amino acid composition and sequence'. The sentence "the coacervate composition can vary depending on the chemistry of the coacervate-forming components," does not contain information because the coacervate composition is defined by the coacervate-forming components.

The word chemistry means composition and sequence. We have changed the text on line 87 to peptide composition. Changes have also been made to the text on lines 310-311 from: "This indicates not only that the coacervate composition can vary depending on the chemistry of the coacervate-forming components, but that this can modulate the overall activity of the ribozyme."

To : "This indicates not only that the coacervate composition can vary depending on **differences in the peptide sequence**, but that this can modulate the overall activity of the ribozyme."

L91: A very minor point: Please consider using a different term than 'emergent' because this term is used in many, quite different contexts in OoL literature. The term 'emergent' is usually used to describe higher-order phenomena that are not necessarily expected from the physico-chemical properties of the individual components. However, the listed results seem to be just 'properties of the mixture' that are not necessarily unexpected.

We agree with the definition of the term emergent. We have replaced the term emergent on line 91 with physicochemical properties from: "To explore the role of molecular co-operation via peptide composition on emergent coacervate and reaction kinetics."

To : "To explore the role of molecular cooperation via peptide **composition** on the **coacervate properties** (physicochemical) and reaction kinetics."

L94: It is unclear what information is in that sentence. Independent of the intended meaning of the word 'underpinned', since the only polyelectrolytes in the mixture are RNA and peptides, and the only difference between the mixtures is the sequence of the peptide, it is trivial that Unfortunately we could not find the rest of the sentence here. We have modified the text in line with other comments by the reviewer, this text has now been removed and changed.

Figure 1A: The circles around each amino acid make the figure unnecessarily complicated. I suggest removing the circles. Additionally, I suggest using slightly more space between the rows than between the columns. In the current format, it seems that there amino acids above each other are as connected as amino acids beside each other; that visual impression should be avoided.

Figure Bii: the number of product molecules gives the impression of multiple turnover. I don't remember seeing a description of multiple turnover in the text. If there is multiple turnover, please describe it clearly in text and legend; if there is no multiple turnover please reduce the number of product fragments to two.

Figure 1 has been modified as suggested, in Figure 1A: The circles around the amino acids are removed and also appropriate spaces between rows are added and in figure Bii, the product number has been modified and reduced to two.

Figure 1: Coacervate formation with peptide and Hammerhead ribozyme (HH_{min})

L133: The term 'number density' is unfamiliar to me. please clarify.

The number density means the number of droplets. The text has been modified to "number of droplets, within a field of view".

L141: Should 'sub molar' be 'sub millimolar'? The peptides seem to be used at 0.5 mM, which would be sub-millimolar.

Yes, sub-millimolar is appropriate in this case, this has been modified.

Figure 2A: Please show all ribozyme schematics in the same size, and all peptides in the same size. Otherwise the impression is raised that peptides of different length are employed, and that impression should be avoided.

I suggest converting the 'circular' text to a straight text. The 'circular text' makes it harder to read.

The figure 2A schematic is modified, both ribozyme and peptides are now changed so that they are the same size. Some but not all of the circular text was modified due to space constraints.

Figure 2Bii: I suggest removing the squares with different gray tones and the 'gray tone scale'. These two elements are unnecessary / redundant and make the figure more complicated than necessary. The numbers below the squares are sufficient, together with the gel image.

The squares with gray tones (figure 2Bii) are removed and the numbers (product yield %) are added to the gel image. Figure 2Bi and 2Bii are merged to 2B.

Figure 2Cii: It is unusual to display catalytic rates as column graph because the value is a discreet number, not an amount.

Shouldn't the 'k' be lowercase? I believe that kinetic values are lowercase; thermodynamic values are uppercase.

We thank the reviewer and have made the modifications so that we use lowercase 'k' to figures 2, 4 and 5, as shown below. We have chosen to keep the bar chart as is it.

Figure 2: Ribozyme activity within coacervates

Figure 4: Determination of RNA and peptide concentration inside the coacervate phase

Figure 5: Diffusion coefficients of ribozyme within coacervate droplets

L302: I suggest converting 'obtained' to 'estimated' or 'calculated' because the RNA and peptide concentrations are derived from somewhat complicated models as opposed to direct measurements.

“obtained” is changed to “calculated” in the text.

L352: Please add that this RNase A experiment not only identifies direct RNA interactions between RNA and peptide but also indirectly identifies whether annealing with another RNA strand occurs.

The text is modified from - “Therefore, incubation of RNase A with RNA/peptide dispersions can be used to identify RNA-peptide interaction sites that might hide or restrict access to particular parts of the ribozyme from RNase A with single nucleotide resolution.”

To - “Therefore, incubation of RNase A with RNA/peptide dispersions can be used to identify RNA-RNA and/or RNA-peptide interaction sites that might hide or restrict access to particular parts of the ribozyme from RNase A with single nucleotide resolution.”

L355: It is unclear why the "lower concentration of the RNA and peptide" would allow "to focus on the RNA-peptide interaction alone and to avoid any issues that may arise from RNaseA diffusion through the coacervate." Does that mean that coacervate formation is avoided? If so, wouldn't this mean that the interactions could be very different with single peptide molecules and RNA as opposed to aggregated peptide molecules in the coacervate with RNA? Please clarify.

Coacervate formation is avoided to limit any effects of the coacervate environment hindering RNase A interaction with the RNA. The charged crowded environment of the coacervate could prevent RNase A diffusion through the droplet and RNA accessibility. Therefore, it was important to undertake RNA degradation in a non-coacervated state but where the peptide and RNA would had formed complexes.

This condition is expected at low concentration of RNA and peptide (below the critical coacervation concentration) and is driven by the intrinsic binding constant between RNA and peptide. As the RNA and peptide molecules are small and the peptide does not form secondary structure (unpublished CD spectroscopy) we do expect the molecular interactions in RNA-Peptide complexes to be equivalent to those within the coacervate. It could be possible that within the coacervate when the RNA is at high concentration and close proximity to one another, peptides could interact with two different ribozymes, as the reviewer has previously suggested. This could change the RNA secondary structure but would not change the underlying molecular interactions between the RNA and peptide between coacervate droplets and complexes.

To provide more explanation, we have included additional text and reference that describes the formation of complexes at concentrations below the critical coacervation concentration:

Whilst the interactions within the coacervate droplet may not be exactly the same, it was critical to work at concentrations where droplets were not present but where RNA-peptide interactions are present within small “clusters”^{19, 20}. This was important to avoid any issues that could inhibit enzymatic activity from the charge and crowded environment of the coacervate phase. This could include inhibition of the RNase A due to restricted motion, diffusion and accessibility to the reaction site, for example. This assay can provide invaluable insights into the RNA-peptide interactions that cannot be accessed within the droplet. insights into the RNA-peptide interactions that cannot be accessed within the droplet.

60. Kar, M. et al. Phase-separating RNA-binding proteins form heterogeneous distributions of clusters in subsaturated solutions. *Proc Natl Acad Sci U S A* **119**, e2202222119 (2022).

61. Chowdhury, A. et al. Driving forces of the complex formation between highly charged disordered proteins. *Proc Natl Acad Sci U S A* **120**, e2304036120 (2023).

Figure 4: The schematics in (A) should use the same axis labels as the graphs containing the measured data in Bi and Bii. Please plot the schematics in the same linear scale (not log) as the graphs with measured data. I did not understand the text corresponding to figure 4, or the figure itself. While this is in part my educational background, an effort to make the text and figure accessible to a broader audience would be appreciated.

We thank the review for this comment and have modified the figure, supplementary information and text around the figure 4 to make it more accessible to the broader audience. The schematics in A have now been moved to the SI and replaced with a phase diagram schematic to demonstrate the concept of the approach. A is a reminder of text book information and was placed into the SI as a reminder of phase diagrams theory. We chose to plot the graphs for the old figure Bi in both log and linear scale for the charge ratios to demonstrate the co-incident point at 250 μ M RNA / 500 μ M peptide as one of the data points used to determine the tie-line as well as to make it easier to visualize the tie-lines. As A has now been moved to the SI there should be no confusion with the different descriptions in one figure.

Furthermore, Figure 4 has now been modified to include a schematic illustrating how the concentrations in the condensed phase are determined from the intersection of a tie-line with an isorefractive line. The schematic introduces many of the quantities presented in the rest of the figure, such as the dilute phase concentrations (Figure 4B) and subsequent tie-lines (Figure 4C), Q-phase microscopy data (Figure 4D) from which it was possible to obtain the condensed-phase concentrations (E) and polymer volume fractions (F). Furthermore, we have simplified the main text.

The main text now reads:

This method uses tie line analysis in combination with quantitative phase microscopy (QPM) from which it is possible to determine the concentrations of RNA and peptide within the coacervate phase. The advantage of this technique is that it utilizes small amounts of sample and is not reliant on fluorescent methods, which could lead to artifacts²¹. This technique utilizes ternary phase diagrams which are plots of the phase behaviour with increasing component concentration (Supplementary Figure S13). The transition from a one phase region (homogeneous solution) to a 2-phase region (coacervate plus dilute phase) are given by the binodals (Supplementary figure S13). A tie-line connects the concentrations in the dilute phase (on the dilute binodal branch (Figure 4A, Supplementary figure

S13) to the concentrations in the coexisting condensed phase (on the condensed binodal branch, Figure 4A, supplementary figure S13). For all points along this line, the concentrations of peptide and ribozyme within coacervate phase and in the co-existing dilute phase are fixed (Supplementary figure 13). Although, the dense-phase composition lies on the tie line, the tie line alone is insufficient to specify the precise location of the condensed binodal branch and thus the condensed-phase concentration. To determine where the tie line meets the condensed binodal branch, quantitative phase imaging (QPI) can be employed to obtain the isorefractive line (Figure 4A and Supplementary figure 14). QPI provides the optical phase shift between the droplet and the surrounding dilute phase and is sensitive to the refractive index difference between these two phases and the local droplet thickness^{21, 22} (see Supplementary Materials and Methods and Supplementary figure S13 and S14). Crucially, the refractive index difference (Δn) is dependent on the peptide and ribozyme concentrations in the condensed phase. Each pair of peptide and ribozyme concentrations that are consistent with the measured refractive index difference lie on an “isorefractive line” (Figure 4A). The concentration of peptide and HH_{min} in the coacervate phase is given by the intersection of the tie line with the isorefractive line. Therefore, by simultaneously solving the isorefractive and tie-line equations, we can determine the condensed-phase concentrations of RNA and peptide for each of the coacervate systems.

Utilizing this approach we first determined the tie-line for each peptide/RNA system

The figure has been modified from:

Figure 4: Measuring RNA and peptide concentration inside the coacervate phase and their comparison. **A.** Schematic phase diagram of an associative ternary mixture in the composition plane spanned by two polymer solutes, denoted p and r , on log scales. System-average (total) compositions in the gray region are thermodynamically unstable and demix, resulting in an equilibrium coexistence of two phases located compositionally (named the 2-phase region) at the boundary of the demixed regime (binodal) and connected to the system-average composition by a tie-line (dashed). Points A-F (dashed circles) represent distinct samples prepared on the same tie-line with increasing total concentrations of both polymers. Each of these unstable systems demixes, giving rise to coexisting dilute (light red, polymer concentrations (p^I, r^I)) and coacervate (red, (p^{II}, r^{II})) phases. As one proceeds along the tie-line from A to F, the coacervate phase occupies an increasing fraction of the total system volume, while the concentrations of p and r in each phase do not vary. Note that the coacervate phase

volume was calculated from the phase diagram by the lever rule for $p^{\text{II}}/p^{\text{I}} = 1000$. **Bi.** Composition of the coexisting phases for each peptide/HH_{min} system on log-log-scales. The system-average (total) concentrations for the dilute-phase measurements were the same for each system (open red circle). Open black symbols (prepared at $[p] = 500 \mu\text{M}$) denote the total concentrations at which the samples were prepared for QPI. Error bars on dilute and total compositions represent the standard deviation of repeat measurements, while error bars on coacervate composition represent error propagation via Jacobians (see methods). **Bii.** Data plotted on linear scales to visualize the condensed-phase compositional differences between systems. **C.** Quantitative phase images of peptide-HH_{min} coacervates (selected with radii between $2.67 \pm 0.04 \mu\text{m}$). The colour bar indicates the optical phase shift in radians, which is proportional to the product of local droplet thickness and the refractive index difference (Δn) between the coexisting phases. Scale bar $5 \mu\text{m}$. **D.** The molar concentrations of HH_{min} (white) and peptide (grey) inside the condensed phase for each system. Error bars are the same as in B but rescaled by net polymer charge. **E.** Fraction of the coacervate phase volume occupied by polymers. The remaining volume is occupied by solvent and salt ions. Error bars are the same as in B and D, rescaled by the partial specific volume of each polymer. **F.** Plot showing the correlation between the rate constant (K) and the concentration of HH_{min} (in molar concentration, from D) within the coacervate for each system.

To:

Figure 4: Measuring RNA and peptide concentration inside the coacervate phase and their comparison. **A** Schematic representation showing the geometry of a ternary phase diagram for determination of polymer concentrations in the coacervate. The method uses a combination of tie-line analysis (purple line) with quantitative phase microscopy (QPM) to obtain the isorefractive line (dotted line). The tie line is firstly determined by the dilute phase concentration (i) and the reference concentration (ii). The isorefractive line gives the refractive index difference between the droplet and its surrounding phase Δn (assuming a linear sum of contributions from the ribozyme and peptide). The point where the isorefractive line meets the tie-line gives the condensed-phase concentrations of ribozyme and peptide (iii) for each of the coacervate systems. **B**. **HH_{min}/Peptide concentration measurement in the dilute phase.** HH_{min} concentration (transparent bars) measured in the dilute phase by fluorescence spectroscopy with 10% FAM-labelled HH_{min} mixed with unlabelled HH_{min}. Dilute phase concentration of the peptides (grey bars) were measured via BCA assay. Error bars obtained from at least 3 repeats (see methods). **C**. **Quantitative phase images** of peptide-HH_{min} coacervates (selected with radii between $2.67 \pm 0.04 \mu\text{m}$). The colour bar shows the optical phase shift in radians, which is proportional to the product of local droplet thickness and the refractive index difference (Δn) between the coexisting phases. Scale bar $5 \mu\text{m}$. **D**. **Tie-lines for each of the peptide-RNA systems (Di).** **Log-log plot** showing (i) the dilute phase binodal directly measured for each peptide system. (ii) Open red circle denotes the reference concentration of RNA (250 μM) and peptide (500 μM) or used in the kinetic assays. The open symbols denote the average concentrations used to prepare each system for QPM. (iii) Concentrated phase binodal obtained by QPM. Error bars on dilute and total compositions represent the standard deviation of repeat measurements, while error bars on coacervate composition represent error propagation via Jacobians (see supplementary Information). **Dii**. **Tie-lines shown on linear scales** to visualize the condensed-phase compositional differences between systems. **E**. **Molar concentrations of HH_{min} (white) and peptide (grey) inside the condensed phase** for each system. Error bars are the same as in B but rescaled by net polymer charge. **F**. **Fraction of the coacervate phase volume occupied by polymers.** The remaining volume is occupied by solvent and salt ions. Error bars are the same as in B and D, rescaled by the partial specific volume of each polymer. **G**. **Plot showing the correlation between the rate constant (k) and the concentration of HH_{min}** (in molar concentration, from E) within the coacervate for each system.

The description of the phase diagrams based on text book knowledge has been moved to the Supplementary information:

4.1. Determination of HH_{\min} and peptide concentrations and tie line:

To determine the concentration of HH_{\min} and peptide within the coacervate phase, we employed an approach based on thermodynamic and optical considerations that has been recently described to define the RNA and peptide concentrations within phase separated droplets²³. This method uses a tie-line analysis (see Supplementary figure 13) in combination with quantitative phase microscopy and makes it possible to obtain the concentrations of RNA and peptides within the coacervate phase.

Figure S13: Cartoons to illustrate generic expectations for a ternary mixture in which the solutes p and r undergo associative phase separation. **A.** Bold line denotes the binodal, the concentrations at which the system transitions from the one phase region (homogeneous solution, white) to a two-phase region (phase coexistence, shaded). Within the two-phase region, tie lines (dotted line) connect the average system composition (e.g. point A) to the concentrations in the coexisting phases (dilute phase in pink, condensed phase in red). **B.** As one proceeds along the tie-line from A to F, the condensed phase occupies an increasing fraction of the total system volume. **C.** At each point along the tie-line, the concentrations of p and r in the dilute phase remain constant. **D.** At each point along the tie-line, the concentrations of p and r in the condensed phase remain constant.

The tie line in the 2-phase region of a phase diagram is given by two points; the first is the total concentration of RNA and peptide used for our experiments (250 μM RNA and 500 μM peptide), the second is the point at the dilute phase binodal determined by measuring the concentration of RNA and peptide in the supernatant or dilute phase by standard calibration methods.

The figure legend for the new supplementary figure 10 was modified:

Editorial note: Figure S14A adapted from McCall, P.M. et al. Label-free composition determination for biomolecular condensates with an arbitrarily large number of components. *bioRxiv* (2023).

Figure S14: A. Schematic representation showing the geometry of a ternary phase diagram for determination of polymer concentrations in the coacervate. The method uses a combination of tie-line analysis (purple line), where the gradient is given by $m_{TL} = \frac{\bar{c}_r - c_r^{dil}}{c_p - c_p^{dil}}$, with isorefractive line analysis (red line) where the gradient is given by $m_{IRL} = -\frac{dn}{dc_p}$. Satisfying the equation 41 (given above) gives the peptide and RNA concentrations in the coacervate phase (c_p^{cond} , c_r^{cond}). The tie-line is determined by measuring the dilute-phase concentrations of ribozyme, c_r^{dil} , and peptide, c_p^{dil} (point A) and the average concentrations of ribozyme \bar{c}_r and peptide \bar{c}_p at which the system was prepared (point B). The isorefractive line depends on the refractive index difference Δn between a coacervate droplet and its surrounding phase and is calculated assuming a linear sum of contributions from ribozyme and peptide. **B.** Co-existing phases and tie-lines for each peptide system in the concentration plane on log scales. This is the same data as in Figure 4B in the main text, but now plotted in the molar concentration of the polymers rather than charge concentration. **C.** Same data as shown in B, now on linear scales.

References to the new figures have been amended within the text.

L397 and many other instances: I suggest replacing the word 'tune' to describe the influence of the coacervate on ribozyme activity because 'tuning' usually requires an agent with an intention such as an experimenter or engineer. I would find 'influence' more neutral and appropriate.

The word "tune" is replaced by "influenced" or "modulated" in those places.

L369: there should be a space between 'RNase' and 'A'

Space has been added between RNase and A in all instances.

Figure 6: Please remove sub-panel 6D because it adds no information and makes the figure more confusing / less clear. The introduction of the gray scale is redundant with the height of columns in A9 of panel C. If the authors don't deem the columns in sub-panel C clear enough then the sub-panel C should be increased in size. The averages and error bars shown in sub-panel D are unnecessary because they are represented in the columns and their error bars in sub-panel C.

Sub-panel 6D is removed in figure 6.

L402: I suggest refraining from the use of the word 'library' because this implies that the peptides were used as a mixture. I suggest calling them 'seven peptides'.

We have changed the text from “In brief, we show that our small library of peptides with random sequence will form coacervate droplets with ribozyme and that the ribozyme is functionally active within the coacervate dispersion (albeit with much slower rates compared to the buffer solution)” To: “In brief, we show that each peptide from our small library of seven arbitrary peptides will form coacervate droplets with the minimal hammerhead ribozyme. The ribozyme is functionally active within the coacervate dispersion, albeit with much slower rates compared to the buffer solution.”

And from “Significantly, between the library of 7 short peptides” to “Significantly, between the 7 short peptides”

From “that we have used a small library of peptides” to “we have used seven peptides with a small”
From “Despite the modest library size” to “Despite only investigating seven peptides”

L407: This is usually called a '15-fold difference'.

15-times is changed to 15-fold in the modified text.

L447: Please clarify what 'total RNA' means. Is that total RNA from an organism?

Here the total RNA means RNA isolated from human induced pluripotent stem cells (iPSCs). The following text has now been added to the main text: “from human induced pluripotent stem cells; iPSCs,”

L450: Please add a few words to explain what is meant by 'chemically diverse molecules', perhaps by adding 'such as x, y, z.'"

We have included some examples. “For example, pools of chemically diverse molecules such as metabolites, peptides, nucleic acids, or lipids”

Reviewer #2 (Remarks to the Author):

In this report different combinations of RNA hammerhead ribozyme and a small peptide library are studied for their coacervation performance and catalytic activity. Because the peptide library is cationic in charge, all combinations give coacervates. There are differences in density, which is also related to the activity of the ribozyme.

This work is a continuation of earlier work by the group and the findings are to be expected, also in line with for example the work by the Keating group. As such, the results are rather incremental. The peptide library used is moderate of size and although some design results are extracted from the results in the discussion, these aspects are not further corroborated. Besides this general reservation about the novelty of the research, a number of technical comments should be addressed.

In response to the reviewers remarks and in line with reviewer 1 we have added additional references from the Keating group and included additional text in the abstract and introduction which highlight the novelty of our study. Specifically, we have acknowledged the small size of the peptide library in the discussion. “Despite only investigating seven peptides, our results also show that activity within the

coacervate dispersions are modulated by the distributed charge patterning of heterogeneous peptides interacting with hammerhead ribozyme.”

On the microscopic level, coacervates, can induce the secondary structure of peptides³ or alter the tertiary structure of RNA⁴ which can affect ribozyme activity. Indeed, previous studies have shown that increasing the length of the polyanion which forms the coacervate will increase the fraction of product generated by ribozymes⁵. Furthermore, it has been shown that changing the peptide sequence from polyarginine to polylysine and (RGG)_n and modifying the length of the peptide can lead to differences in material properties of the droplet which then impact the ribozyme activity⁶⁻⁸. Oligopeptide based coacervates have additional advantages to homo (poly)peptide systems as the properties of the different amino acids can impart different physical coacervate environments. For example, increasing hydrophobic moieties in peptides will increase the hydrophobicity of the internal environment and partitioning of hydrophobic molecules into the core,⁹ while increasing the strength of the interactions will affect the diffusion dynamics and material properties of the droplet⁸.

with new references:

38. Cakmak, F.P., Choi, S., Meyer, M.O., Bevilacqua, P.C. & Keating, C.D. Prebiotically-relevant low polyion multivalency can improve functionality of membraneless compartments. *Nature communications* **11**, 1-11 (2020).
39. Poudyal, R.R. et al. Template-directed RNA polymerization and enhanced ribozyme catalysis inside membraneless compartments formed by coacervates. *Nature communications* **10**, 1-13 (2019).
40. Iglesias-Artola, J.M. et al. Charge-density reduction promotes ribozyme activity in RNA-peptide coacervates via RNA fluidization and magnesium partitioning. *Nat Chem* **14**, 407-416 (2022).
41. Baruch Leshem, A. et al. Biomolecular condensates formed by designer minimalistic peptides. *Nat Commun* **14**, 421 (2023).
42. Akahoshi, Y. et al. Phase-Separation Propensity of Non-ionic Amino Acids in Peptide-Based Complex Coacervation Systems. *Biomacromolecules* **24**, 704-713 (2023).

Furthermore, alternative explanations for the observed variations in reaction rate have been provided. “However, deconvoluting the effect of coacervates on enzymatic activity is non-trivial as coacervates have been shown to tune reactions in a variety of ways. For instance, a coacervates’ ability to support primitive RNA reactions can lead to the slowing down of cleavage rates¹⁰, increased ligation yields¹¹, reduction in yields for template-directed polymerisation¹² and preferential ligation of linear RNA over circular RNA⁶.. ‘

We have added additional text at the end of the introduction. The text has been revised to provide a better description of the new insights from the study taking into account previous studies, from:

“This system provides a versatile and robust model for determining how peptide chemistry can change the physicochemical properties of the coacervate and how these subsequently impact reaction kinetics. Our study provides a physical framework for understanding how emergent properties of biological condensates can tune biochemistry within the cell and marks a step toward unravelling how coacervates formed within a prebiotic soup can affect compartment properties and reaction outcomes across a population of primitive cells. “

To:

We exploit RNA/peptide coacervates as a versatile and robust model for determining how amino acid sequence can change the physicochemical properties of the coacervate and how these subsequently impact reaction kinetics. Significantly, our study extends our understanding of the effect of compartmentalization on enzymatic reactions by correlating coacervate properties to ribozyme activity. Furthermore, it marks a step toward unravelling how coacervates formed within a prebiotic soup can affect compartment properties and reaction outcomes across a population of primitive cells.

First of all, the coacervates are made in 10 mM Tris-136 HCl, 1 mM MgCl₂, pH 8.1. How sensitive are the coacervates to varying pH and electrolyte concentration?

It has been well established that the properties of coacervate droplets can be affected by pH and electrolyte concentration²⁴⁻²⁸. To directly address the reviewers comments we undertook additional experiments to characterise the sensitivity of the peptide/RNA coacervates to pH and electrolyte concentration, we imaged the coacervate droplets prepared from P-2 and P-3 under the following conditions:

1. 10 mM Tris- HCl, 1 mM MgCl₂, pH 7.1
 2. 10 mM Tris- HCl, 1 mM MgCl₂, pH 9.1
- A. 50 mM Tris- HCl, 1 mM MgCl₂, pH 8.1

Figure R2.1: Optical microscopy of coacervate droplets. Peptide (P-2 and P-3)-RNA coacervate droplets were prepared under different buffer conditions. (1) 10 mM Tris- HCl, 1 mM MgCl₂, pH 7.1 (2) 10 mM Tris- HCl, 1 mM MgCl₂, pH 9.1 (A) 50 mM Tris- HCl, 1 mM MgCl₂, pH 8.1 and imaged under confocal microscopy to determine their stability under different pH and salt conditions. Scale bar 10 um.

Figure R2.2: Left: Raw confocal micrograph of coacervates. Right: Binarized and inverted image with detected coacervates. The radii were used to calculate the volumes with the assumption that the droplets are spherical. Scale bar 10 μm

Along with the previously obtained data for P-2 and P-3 coacervates in 10 mM Tris-HCl and 1 mM MgCl_2 at pH 8.1 we analysed the 2D confocal images of the droplets using the `imfindcircles` function in MATLAB (Figure R2.2) to obtain a distribution of volumes. The average value and standard deviation of the volumes were calculated directly from these distributions. (Figure R2.3).

Figure R2.3: Distribution of droplet volumes. Top: volume distribution of coacervates at different pH for a) the P-2/HH_{min} system and b) the P-3/HH_{min} system. Bottom: Volume distribution of coacervates at different buffer conditions c) the P-2/HH_{min} system and d) the P-3/HH_{min} system. Frequency is the count of the number of droplets.

For both systems, we find that the mean droplet volume varies with pH. In the case of P-2/HH_{min} coacervates, we observe a volume maximum at pH 8.1 while for the P-3/HH_{min} coacervates, we see a volume maximum at pH 7.1. This indicates that the buffer pH does impact the coacervate volume, and by extension, the phase diagram. It is interesting to note that the pH affects P-2 and P-3 coacervates in different ways. Plots of the mean droplet volume versus buffering agent (Tris) concentration show a 2-fold decrease in the volume for the P-2/HH_{min} system and a 200-fold decrease for the P-3/HH_{min} system upon a 5-fold increase in Tris.

Together our results show that pH and salt effects on the volume fraction of the coacervate which is a strong indication that the biophysical properties of the droplet will also be affected which are in agreement with previous studies¹⁸.

Figure R2.4: Left: The total droplet volumes for both the P-2/HH_{min} and P-3/HH_{min} systems show that pH affects the total droplet volume between pH 7.1-9.1. Right: Total droplet volume changes with increasing buffer concentration.

What is the pH in the microenvironment of the different coacervates ? This could also have an effect on the catalytic activity of the ribozyme.

This is an important comment. It is possible that pH differences within the microenvironment can have an effect on the catalytic activity of the ribozyme. Previous studies have shown that the rate of the ribozyme has a linear dependence on the pH^{29, 30}. It is possible to measure the pH of buffers using pH sensitive dyes. However, care should be taken when using these dyes within coacervate droplets as the environment of the droplet can change the fluorescence properties of the fluorophores due to changes in rotation or the dissipation of energy between excited states. Furthermore, it can be very challenging to calibrate the fluorophore in these complex environments. Due to these concerns regarding the use of fluorescence to measure the pH inside the droplet we sought to develop a theoretical approach to approximate the pH inside the coacervate droplet. Our theoretical approach gives a predicted internal pH difference between P-2 and P-3 coacervates of -0.0301 ± 0.0006 (see supplementary information note 5 and below) with less than 2 % chance for a pH difference between the dense-phase pH between the P-2 and P-3 to be large enough to be responsible for the 15-fold reaction rate difference observed in the experiments. However, from the previous reviewers' comment, we do observe small changes in the volume of the coacervates with a change in pH between 9 and 7. This indicates that the pH could have an effect on the physico-chemical properties of the droplet which could further impact the ribozyme activity. We have included new text into the discussion to highlight that pH differences within the coacervate droplet could contribute to the observed differences in ribozyme activity: **Therefore, rate of ribozyme cleavage could be affected by the substrate partitioning and diffusion through the coacervate as well as to changes in ribozyme activity, for instance from interactions with peptide. In addition, changes to local pH and salt can directly affect ribozyme rate as well as modulating the properties of the coacervate which could contribute to changes in ribozyme rates (See supplementary information note 5).**

The following text has been included as an additional supplementary note to describe the theoretical model to estimate the internal pH.

Supplementary Information note 5: determination of pH within the coacervate droplet:

Given that ribozyme activity is linearly dependent on the pH^{15,16}, we sought to determine if differences in pH could explain the 15 x difference in the rate between the P2 and P3 coacervates that we observed. The literature indicates that a 15x difference in the rate correlates to a difference of 1.5 pH units. We chose to use a theoretical approach to estimate the pH of the dense phase of the peptide/HH_{min}

coacervate due to experimental challenges that arise from direct measurements of small volume fractions of the condensed-phase. Typically, the volume fraction of the condensed phase, ϕ^{II} , is approximately 2×10^{-4} , such that a 1-mL sample contains only 0.2 μL condensed phase, see Table S12). In addition, pH-probes can have different fluorescent properties in a macromolecularly dense coacervate phase (with solvent volume fraction ϕ_s between 0.5 and 0.6) compared to dilute aqueous solutions. This makes it challenging to obtain the true pH in the condensed phase from calibrations obtained in the dilute aqueous phase (with $\phi_s \cong 1$). In the theoretical approach, developed here, we assume that the proton number (N^{tot}) is conserved upon phase separation to estimate the pH in the dense phase (pH^{II}) from the pH measured in the dilute phase (pH^I) and the starting pH of the coacervate components.

Prior to mixing, the total number of (hydrated) protons in solution (N^{tot}) is set by the pH of the Tris HCl buffer (8.1). In our system, we assume that N^{tot} does not change upon mixing of the RNA and peptide. We believe that this is a reasonable assumption as the peptides are added as a protonated chloride salt and the RNA as a deprotonated sodium salt, each with pK_a values far from the pH of the buffer. Thus, the presence of these polyelectrolytes in solution does not appreciably alter N^{tot} . If the protonation equilibrium of a species in one or both of the demixed phases is significant, one may expect a change in the N^{tot} . This could arise if phase-specific shift in protonation pK_a of a species away from the standard dilute-solution value and/or a shift in the local pH within a phase away from pH^{tot} and towards a solute pK_a . In this case we assume that there are no pK_a shifts and solve for the local pH in the limit that the values are sufficiently far from the pK_a values relevant for our peptides and RNA such that N^{tot} is unchanged by phase separation. As we shall see, the local pH values we obtain are self-consistent with this limit. Consequently, the number of (hydrated) protons in the dense phase (N^{II}) is given by $N^{II} = N^{\text{tot}} - N^I$, where N^I is the number of (hydrated) protons in the dilute phase. Converting to molar proton concentration ($c = \frac{N}{V}$) and assuming no change in solution volume upon phase separation ($V^{\text{tot}} = V^I + V^{II}$), this becomes

$$c^{II} = \frac{1}{\phi^{II}} c^{\text{tot}} + \left(1 - \frac{1}{\phi^{II}}\right) c^I, \quad (59)$$

where $\phi^{II} = \frac{V^{II}}{V^{\text{tot}}}$. This equation relates the concentration of (hydrated) protons in the condensed-phase to that in the dilute phase and the total protons prior to phase separation.

Coacervates have a high charge density, the electrostatic interactions can cause ion behavior that deviate from that of ideal solutes. Here, the pH is conventionally defined in terms of the activity rather than concentration of (hydrated) protons: $pH \equiv -\log_{10}(a_{H^+})$

To calculate pH^{II} from equation 59, the concentrations are rewritten in terms of activities. The activity of a species in phase α may be written in terms of the molal proton concentration in that phase (b^α) as $a^\alpha = \frac{\gamma^\alpha b^\alpha}{b^\circ}$, where γ^α is a phase-specific activity coefficient equal to 1 in the ideal limit and $b^\circ = 1 \text{ mol/kg}$ solvent is a phase-independent constant that defines the molality scale. The phase-specific molal concentration is related to the local molar concentration (c^α) by $b^\alpha = \frac{c^\alpha}{\rho_s \phi_s^\alpha}$, where $\rho_s \approx 1 \text{ kg/L}$ is the solvent mass density and ϕ_s^α is the solvent volume fraction in phase α .

Combining these relations together, the dense-phase pH is given by

$$pH^{II} = -\log_{10} \left[\frac{\gamma^{II}}{\phi_s^{II}} \right] - \log_{10} \left[\left(\frac{1}{\phi^{II}} \right) \left(\frac{\phi_s^{\text{tot}}}{\gamma^{\text{tot}}} \right) 10^{-pH^{\text{tot}}} + \left(1 - \frac{1}{\phi^{II}} \right) \left(\frac{\phi_s^I}{\gamma^I} \right) 10^{-pH^I} \right]. \quad (60)$$

This expression contains nine parameters, of which two (pH^I and pH^{tot}) are measured directly and two are inferred from measurements of condensed-phase composition (ϕ_s^{II}) and the phase diagram (ϕ^{II}). ϕ_s^{tot} and γ^{tot} can be approximated to 1 as the (poly)electrolyte concentrations in the dilute phase which are averaged over the entire system are small giving $\phi_s^{tot} \cong \phi_s^I \cong 1$ and $\gamma^{tot} \cong \gamma^I \cong 1$. This leaves a single undetermined parameter, the dense-phase proton activity coefficient γ^{II} , from which the pH of the condensed phase can be determined.

We treated the charge of the protons as being non-ideal due to the high charge concentration of protons within the dense-phase. To do this, we therefore applied Debye-Hückel theory to generate a rough numerical estimate for γ^{II} . The Debye-Hückel limiting law¹⁷ is commonly used to estimate the mean activity coefficient applicable to a pair of electrolytes with charges z_+ and z_- in a solution of ionic strength I as $\log_{10}(\gamma_{\pm}) = -A|z_+z_-|\sqrt{I}$. We take $z_+ = 1$ for the proton and select Cl^- as the anion such that $z_- = -1$. In the dense phase, the prefactor is given by

$$A^{II} = \frac{F^3}{4\pi N_A \ln(10)} \left(\frac{\rho_s \phi_s^{II} b^\circ}{2(\epsilon^{II})^3 R^3 T^3} \right)^{1/2}, \quad (61)$$

where F is the Faraday constant, N_A is Avogadro's number, ϵ^{II} is the local dielectric constant, R is the gas constant and T is the temperature. We have included the factor ϕ_s^{II} to account for the reduced solvent volume in the dense phase compared to typical dilute solutions. We estimate the dense-phase dielectric constant as a weighted sum of contributions from solvent, peptide and nucleic acid as

$$\epsilon^{II} \approx \epsilon_0 \sum_i \epsilon_i \phi_i^{II} \quad (62)$$

using measured dense-phase volume fractions (ϕ_i^{II} , Table S12) and literature values of the relative permittivities (ϵ_i) for viral protein and DNA³¹, (see Table S13). ϵ_0 is the static permittivity of free space. To estimate the activity coefficient for protons in the dense-phase from the Debye-Hückel expression, we need to estimate the remaining parameter, the ionic strength in the dense phase.

While all charged species contribute to the ionic strength, we expect the high concentrations of peptide and HH_{\min} polyelectrolytes in the dense phase to dominate. Accordingly, we make the simplifying approximation that contributions to the ionic strength from all other charges species are negligible. Ionic strength calculations for polyelectrolyte solutions are generally complex^{19,20,21}, owing in large part to the many spatial correlations among charges, such as those from chain connectivity and ion pairing. However, the molar-scale charge-equivalent concentrations we measure in these coacervate systems (Figure 4) suggest that the length-scale for electrostatic screening is comparable to or even smaller than the other relevant correlation lengths. We therefore calculate the ionic strength in the limit that all spatial ion correlations (including those from chain connectivity) are negligible which can be considered a mean-field limit.

In this limit, the equation for the dense-phase ionic strength takes the usual form for disconnected low-valence ions:

$$I^{II} = \frac{1}{2b^\circ} \sum_j b_j^{II} (z_j^{mon})^2, \quad (63)$$

where the index j accounts for peptide and HH_{\min} only. b_j^{II} is the dense-phase molality of polyelectrolyte species j in units of charge equivalents and determined experimentally (Figure 4). z_j^{mon} is the punctate charge of a monomer along the backbone, and is +1 for cationic amino acid sidechains and -1 for phosphates along the RNA backbone. The proton activity coefficient γ^{II} is then given by the Debye-Hückel limiting law using equations 61-63 above. We report the values of ϵ^{II} , I^{II} , and γ^{II} estimated from the equations in Table S15. With γ^{II} in-hand, the dense-phase pH may be computed from equation 60.

Generation of theoretical pH_{II} data via Monte Carlo

Simulated distributions of internal pH presented in Figure S21 were generated from Equation 62 using the Monte Carlo method²² to obtain values of pH^I and pH^{tot} from Gaussian distributions constrained by measurements. For pH^I , the measured values are 8.17 ± 0.02 for the P-2 system and 8.13 ± 0.02 for the P-3 system. For both systems, $pH^{tot} = 8.10 \pm 0.02$. In the Monte Carlo simulation, the mean and standard deviations of the Gaussian distributions were set to the measured central value and uncertainty, respectively. Other parameter values were treated as constant and taken from Table S12 (ϕ^{II}), Table S13 (ϕ_s^{II}), Table S14 (γ^{II}), or taken as 1 for ϕ_s^{tot} , ϕ_s^I , γ^{tot} , and γ^I . For each run, $n_{draw} = 1 \times 10^7$ independent samples were drawn for each of pH^I and pH^{tot} and then randomly paired. Individual pairs for which pH^I was below the threshold

$$pH^{thresh} = pH^{tot} - \log_{10} \left[\left(\frac{\gamma^I}{\gamma^{tot}} \right) \left(\frac{\phi_s^{tot}}{\phi_s^I} \right) \left(\frac{1}{1-\phi^{II}} \right) \right] \quad (64)$$

were discarded, as they would correspond to the unphysical situation of negative proton numbers in the dense phase. Typically, more than 8.5×10^6 pairs remained. pH^{II} was then calculated from Equation 60 for each surviving pair to populate the distributions in Figure S21a. The distribution of ΔpH^{II} in Figure S21b was generated by pairing samples from the synthetic pH^{II} distributions for the P-3 and P-2 systems and taking differences. To assess uncertainty in the most likely value of ΔpH^{II} as well as uncertainty in the probability that $\Delta pH^{II} \geq 0.5$, 1.0, or 1.5, the entire process was replicated 100 times. Uncertainties reported for these quantities represent the standard deviation observed over the 100 replicates.

The probability that ΔpH^{II} i.e. the pH difference between the condensed and dilute phase is 0.5, 1.0 or 1.5 are 0.1449 ± 0.0001 , 0.03951 ± 0.00007 , and 0.01183 ± 0.00003 respectively. However, 100 independent simulated datasets, the most likely difference between the internal pH of P-3/HH_{min} and P-2/HH_{min} condensates is -0.0301 ± 0.0006 (red vertical line). Given this, our results suggest a probability of less than 2% for a pH difference between the dense-phase pH between the P-2 and P-3 to be large enough to be responsible for the 15-fold reaction rate difference observed in the experiments.

Self-consistency assessment of physical limits applied to pH-related calculations

We briefly check for self-consistency between our results and the two physical limits taken in the calculations. First, the dense-phase ionic strength I^{II} was estimated in the limit that the electrostatic screening length (Debye length) is smaller than or comparable to the length-scale associated with ion correlations from chain connectivity. To check this, we use the I^{II} estimates to calculate the corresponding Debye length in the dense phase as

$$\lambda_D = \sqrt{\frac{\epsilon^{II} k_B T}{2 \rho_s \phi_s^{II} F^2 I^{II} b^{\circ}}} \quad (65)$$

and report the resulting values in Table S16. For each coacervate system, we find the screening length is $\lambda_D \leq 0.23$ nm. This is smaller than the typical distance between adjacent monomers for polypeptides (3.65 \AA ,²³ and also the phosphate-phosphate distance on nucleic acids (3.4 \AA), consistent with the limit taken.

The second important limit is that local pH values remain sufficiently far from polyelectrolyte pK_a values such that the total number of hydrated protons in solution does not change appreciably upon phase separation. To check this, we compare the local pH values for the P-2/HH_{min} and P-3/HH_{min}

systems to tabulated pK_a values of the charge groups in the polyelectrolytes. We take the sidechain pK_a values of lysine and arginine to be 10.9 and 13.8²⁴, respectively, and the phosphate pK_a in the RNA backbone to be ~ 2 . In all cases, these pK_a values are several pH units away from the pH measured in the dilute phase (8.17 and 8.12) and also from the most likely pH value in the dense phase obtained via simulation (~ 6.3 in both cases). These local pH values are thus far from the standard pK_a values of the polyelectrolyte charge groups, consistent with the limit taken.

Finally, we note that any errors introduced through the limits and approximations employed here are systematic rather than statistical and will impact estimates for each peptide/HH_{min} system similarly. In particular, systematic errors in γ^{II} will modulate the pH distributions in Figure S21 quantitatively. However, we expect the general distribution shapes and our conclusion that the most likely difference in internal pH between the P-3 and P-2 systems is $0 < \Delta pH^{II} \ll 1.5$ are both insensitive to correlated variation in γ^{II} from such systematic errors.

Table S12: Condensed-phase volume fraction^a

System	ϕ^{II}
P-1/HH _{min}	1.842×10^{-4}
P-2/HH _{min}	1.858×10^{-4}
P-3/HH _{min}	2.209×10^{-4}
P-4/HH _{min}	1.789×10^{-4}
P-5/HH _{min}	2.195×10^{-4}
P-6/HH _{min}	1.883×10^{-4}
P-7/HH _{min}	1.493×10^{-4}

^a Condensed-phase volume fraction represents the fraction of the system volume occupied by the condensed-phase and depends on the total concentrations added. Values presented here are for 500 μ M peptide and 250 μ M HH_{min}.

Table S13: Molecular composition (volume fraction) within condensed phase^a

System	ϕ_p^{II}	ϕ_r^{II}	ϕ_s^{II}
P-1/HH _{min}	0.2510	0.2036	0.5454
P-2/HH _{min}	0.2349	0.2404	0.5246
P-3/HH _{min}	0.2115	0.1926	0.5960
P-4/HH _{min}	0.2314	0.2490	0.5196
P-5/HH _{min}	0.2252	0.1979	0.5769
P-6/HH _{min}	0.2521	0.1589	0.5890
P-7/HH _{min}	0.1971	0.3020	0.5009

^a Solvent volume fraction in condensed phase is calculated as $\phi_s^{II} = 1 - \phi_p^{II} - \phi_r^{II}$. Deviation of row sums from 1 represent rounding error.

Table S14: Approximate dielectric constants of primary coacervate molecular species

Species	Symbol	Value
peptide	ϵ_p	3 ^a
HH _{min}	ϵ_r	8 ^b

water	ϵ_s	78.3 ^c
-------	--------------	-------------------

^a Typical value for protein based on measurements of shell and tail proteins from T7 virus. Ref: (2)

^b We use the value measured for packaged viral DNA (Ref: (2)) as a proxy for RNA.

^c Measured value for water at 25 °C. Ref:²⁵.

Table S15: Estimated dense-phase proton activity coefficients and associated parameters^a

System	ϵ^{II}	I^{II}	γ^{II}
P-1/HH _{min}	45.09	2.617	0.0401
P-2/HH _{min}	43.71	2.688	0.0351
P-3/HH _{min}	48.84	1.835	0.0824
P-4/HH _{min}	43.37	2.713	0.0338
P-5/HH _{min}	47.43	1.961	0.0704
P-6/HH _{min}	48.15	1.959	0.0728
P-7/HH _{min}	42.23	3.458	0.0201

^a All parameters are dimensionless.

Table S16 Estimated Debye-length in dense-phase^a

System	λ_D (nm)
P-1/HH _{min}	0.1933
P-2/HH _{min}	0.1914
P-3/HH _{min}	0.2298
P-4/HH _{min}	0.1907
P-5/HH _{min}	0.2227
P-6/HH _{min}	0.2221
P-7/HH _{min}	0.1698

^a Calculated based on the ionic strengths given in Table S15.

Figure S21: A. Simulated pH of internal condensate: Distribution of plausible condensed-phase (i.e. internal) pH values for the P-2/HH_{min} (top) and P-3/HH_{min} systems. Distributions were calculated from Equation 62 using values of pH^{tot} and pH^I drawn via the Monte Carlo method from Gaussian distributions parameterized by the experimentally-measured mean and standard deviation of the buffer pH (8.10 ± 0.02) and dilute phase (8.17 ± 0.02 and 8.13 ± 0.02 for P-2 and P-3, respectively). Values of other parameters are held constant. For the P-2/HH_{min} system: $\gamma^{II} = 0.1351$, $\phi_s^{II} = 0.5246$, and $\phi^{II} =$

1.858×10^{-4} . For the P-3/HH_{min} system: $\gamma^{II} = 0.1934$, $\phi_s^{II} = 0.5960$, and $\phi^{II} = 2.209 \times 10^{-4}$. For both systems, we take $\frac{\phi_s^{tot}}{\gamma^{tot}} \cong 1 \cong \frac{\phi_s^I}{\gamma^I}$. Histograms represent $\sim 8.5 \times 10^6$ independent samples. Bin-width is 0.05 pH-units. The most probable values of the internal pH are near ~ 6.3 for both systems. The distributions are highly asymmetric with pronounced tails extending to pH 12 (though difficult to see on linear scales). **B.** Distribution of plausible differences in internal pH between the P-3/HH_{min} and P-2/HH_{min} systems: $\Delta pH^{II} = pH_{P-3}^{II} - pH_{P-2}^{II}$. The ΔpH^{II} distribution was populated from differences between internal pH values drawn from the distributions in a. The ΔpH^{II} distribution is peaked near zero and non-Gaussian, displaying instead a narrower peak and also more pronounced tails. A Gaussian distribution (blue line) with the same mean (μ) and standard deviation (σ) as the synthetic ΔpH^{II} distribution is shown for reference. The difference in the tails is most obvious when viewed on a semi-log plot (inset). Bin-width is 0.1 pH-units.

References:

15. Hendry, P., McCall, M.J., Santiago, F.S. & Jennings, P.A. In Vitro activity of minimised hammerhead ribozymes. *Nucleic acids research* **23**, 3922-3927 (1995).
16. Canny, M.D. et al. Fast Cleavage Kinetics of a Natural Hammerhead Ribozyme. *Journal of American Chemical Society* **2004**, 10848-10849 (2004).
17. Atkins, P., de Paula, J. & Keeler, J. *Physical Chemistry*, Edn. 11. (Oxford University Press, 2018).
18. Cuevo, A. et al. Direct measurement of the dielectric polarization properties of DNA. *Proc Natl Acad Sci U S A* **111**, E3624-E3630 (2014).
19. Muthukumar, M. 50th Anniversary Perspective: A Perspective on Polyelectrolyte Solutions. *Macromolecules* **50**, 9528-9560 (2017).
20. de Alcântara Pessôa Filho, P. & Maurer, G. An extension of the Pitzer equation for the excess Gibbs energy of aqueous electrolyte systems to aqueous polyelectrolyte solutions. *Fluid Phase Equilibria* **269**, 25-35 (2008).
21. Manning, G.S. Limiting laws and counterion condensation in polyelectrolyte solutions: IV. The approach to the limit and the extraordinary stability of the charge fraction. *Biophysical Journal* **7** (1977).
22. G.M, A. Error propagation by the Monte Carlo method in geochemical calculation. *Geochimica et cosmochimica acta* **40**, 1533-1538 (1976).
23. Hanke, F., Serr, A., Kreuzer, H.J. & Netz, R.R. Stretching single polypeptides: The effect of rotational constraints in the backbone. *Europhysics Letters* **92**, 53001 (2010).
24. Fitch, C.A., Platzer, G., Okon, M., Garcia-Moreno, B.E. & McIntosh, L.P. Arginine: Its pKa value revisited. *Protein Science* **24**, 752-761 (2015).
25. Malmberg, C.G. & Maryott, A.A. Dielectric Constant of Water from 0° to 100°C. *Journal of Research of the National Bureau of Standards* **56** (1956).

Furthermore, there is a sharp change between P3 and P5 in fig 4F when changing the ribozyme concentration. This cannot be merely explained by the concentration.

Yes, the reviewer correctly points out the ribozyme rates have a 4x difference between P3 and P5 indicating that the rate is not affected by concentration alone. Indeed the weak correlation between the rate constant and the concentration of the ribozyme within the droplet agrees with this observation. Further, our results show that there is a strong linear dependence of the natural log of the rate constant with the natural log of the diffusion coefficient indicating that the rate is better correlated to the diffusion coefficient vs. the concentration, suggesting that these differences in rates could be attributed to the diffusion coefficient. Despite these results, it is important to state that these “simple” systems are far from simple and the emergent properties of the coacervate environment can contribute to tuning ribozyme rate constant in a myriad of ways. For example, concentration can be a weakly contributing factor but other factors including the diffusion of the substrate and product, the structure of the ribozyme

as well as pH can all contribute to changing the rates. Indeed, our studies shows our steps to defining and understanding how emergent properties of the coacervates can contribute to changing ribozyme rates.

We have included an additional paragraph do discuss this point: Furthermore, a weakly correlated decrease in the reaction rate with increasing ribozyme concentration inside the coacervates contradicts our general expectation of enzyme kinetics by mass action, where increased enzyme concentration is expected to increase the reaction rate. Our results agree with recent theoretical studies which shows that this expectation could break down within the droplet environment⁴⁵ and some of the ribozyme within the droplet could be rendered inactive by coacervate formation. Further, this decrease in rate with increasing ribozyme concentration, could be due to molecular or mesoscopic effects. For example, on the molecular level, RNA-RNA interactions driven by high local concentrations could lead to substrate binding between two neighboring ribozymes which reduces ribozyme activity. Increased packing (increased density and concentration) of the ribozyme within the coacervate can lead to changes in the secondary structure that could lead to loss of activity. In addition, peptide-RNA interactions could block RNA substrate binding sites. Here stronger binding between the ribozyme and peptide can lead to denser droplets with increased concentrations but reduced activity. On the mesoscopic level, molecular diffusion is slower in denser droplets, which can reduce reaction rates. Therefore, rate of ribozyme cleavage could be affected by the substrate partitioning and diffusion through the coacervate as well as to changes in ribozyme activity, for instance from interactions with peptide. In addition, changes to local pH and salt can directly affect ribozyme rate as well as modulating the properties of the coacervate which could contribute to changes in ribozyme rates (See supplementary information note 5).

How does the particle size distribution affect the results obtained?

We compared the ribozyme concentration, diffusion coefficient and reaction rate with droplet size distribution (Figure R2.5). The plots show that there is no correlation between the average size of the droplet and the concentration of HH_{min} with a potential but not highly correlated trend between the droplet size and the diffusion coefficient and the reaction rate. It is important that care should be taken when interpreting these results as the coacervate size is a kinetic parameter and can vary depending on the time frame of imaging and the ability for the droplet to coalesce. Consequently, we have not included these results in the revised manuscript or supplementary information. In depth characterisation of the particle size distribution on the thermodynamic parameters is important but goes beyond the scope of our study.

Figure R2.5: Comparison between droplet size with ribozyme concentration, ribozyme diffusion coefficient and reaction rate.

Reviewer #3 (Remarks to the Author):

In the manuscript "RNA-peptide co-operativity tunes the activity of coacervate microdroplets dispersions," Ghosh et al. present that coacervates composed of randomly generated sequences of peptides can alter ribozyme kinetics. The authors randomly generated small variation in sequences of 13-mer peptides with a fixed percentage of amino acid types and used them with hammerhead (HH) ribozyme to produce the coacervate phase. While this peptide library is rather modest size, they showed that the overall ribozyme rate constant in coacervate phase is anti-correlated to the ribozyme concentration and diffusion coefficient of ribozyme.

Although discussions have arisen regarding the testing of randomly generated peptides in the emergence of life, no reported works have attempted to use such peptides for coacervation, to my knowledge, due to potential difficulties and limitations in coacervation and interpretation. In this context, this work is intriguing, offering valuable insights into the ability of small randomly generated variation in sequence of peptides, devoid of biased design or inspiration from biological proteins, to produce coacervates with RNA that vary in composition and chemical properties, influencing RNA reactivity within. The manuscript is original and logically ordered, supported by experimental results. However, there are some concerns regarding substrate diffusion, local ribozyme reactivity, and the rationale behind choosing a 10x higher initial concentration for polymer content estimation. Given the fact that this manuscript can potentially have a broader impact by establishing a linkage between the physicochemical properties of membrane-free compartments and their molecular reactivity, I would recommend publishing in Nature Communications after addressing these concerns. Details are provided below:

1. While I understand the authors' intent to maintain cationic amino acid composition for coacervation with HH ribozyme, I couldn't fully follow the reasoning for choosing these composition ratios in amino acid types. For example, in the provided Python code in SI, each amino acid for acidic, polar, and hydrophobic amino acids has only a 0.03 probability each, while hydrophobic amino acids have a 0.1 probability each. As this could provide some bias in amino acid selection from amino acids pools given to the code, provide any reasons to support the fact that these are “randomly” generate peptide sequences.

In short, this balance was chosen to ensure that coacervate formation could take place. The probability of long-chain hydrophobic amino acids V, L and I was kept ~0.1 to enhance the chances of coacervate formation among non-uniform positive charge peptides, while G, A and P and other polar amino acids were kept 0.03 to balance an overall contribution. Recent studies have shown that the presence of hydrophobic amino acids (G,A,L,I,V) along with positively charged amino acids (K,R) facilitate coacervate formation^(1, 2). Particularly, long-chain hydrophobic amino acids (V, I, L) with increased aliphatic index were found to be more efficient in coacervate formation compared to short-chain hydrophobic amino acids⁽²⁾. We agree that providing an initial criterion for the generation of the sequence will provide some bias to the peptides and here the bias was towards coacervate formation. The code was used to place these amino acids into a random/ arbitrary sequence. We have now modified the text in the supplementary information to provide additional information rational of the weighting for peptide sequence generator code to provide further explanation.

A random sequence generator script (shown below) was used to generate seven peptide sequences from 12 amino acids. The sequences include 50% positive charge (R and K), 29% long chain aliphatic amino acids (I, V and L) and 21% acidic, polar and hydrophobic amino acids (D, E, S, T, G, A and P). The probability contribution of the different amino acids was chosen to increase the chances of coacervate formation between the peptides and RNA. Recent studies have shown that the presence of hydrophobic amino acids (G,A,L,I,V) along with positively charged amino acids (K,R) facilitate coacervate formation^{32 33}. Particularly, long-chain hydrophobic amino acids (V, I, L) with increased aliphatic index were found to be more efficient in coacervate formation compared to short-chain hydrophobic amino acids³³.

Additional references:

1. Saha B, Chatterjee A, Reja A, Das D. Condensates of short peptides and ATP for the temporal regulation of cytochrome c activity. *Chemical Communications* 55, 14194-14197 (2019).
2. Tabandeh S, Leon L. Engineering Peptide-Based Polyelectrolyte Complexes with Increased Hydrophobicity. *Molecules* 24,(2019).

We further, clearly state that a small selection of peptides places some limitation on our study.

Despite only investigating seven peptides, our results also show that activity within the coacervate dispersions are modulated by the distributed charge patterning of heterogeneous peptides interacting with hammerhead ribozyme.

2. The cited papers in the main text (ref [36] – [39]) seem like they do not mention arginine nor lysine. Please cite papers producing amino acid analogues containing amine groups to justify arginine and lysine. Longo et al (“Primordial emergence of a nucleic acid-binding protein via phase separation and statistical ornithine-to-arginine conversion”*PNAS* (2020) 117, 27, 15731-15739) and Plankensteiner et al (“Amino acids on the rampant primordial Earth: Electric discharges and the hot salty ocean”, *Molecular Diversity* (2006) 10, 3–7) may be helpful.

We thank the reviewer for the comment and have added the following references which discuss synthetic routes to lysine or arginine and thus demonstrate the plausibility of the lysine or arginine within a prebiotic context.

47. Longo, L.M. et al. Primordial emergence of a nucleic acid-binding protein via phase separation and statistical ornithine-to-arginine conversion. *Proc Natl Acad Sci U S A* 117, 15731-15739 (2020).

48. Plankensteiner, K., Reiner, H. & Rode, B.M. Amino acids on the rampant primordial Earth: electric discharges and the hot salty ocean. *Mol Divers* **10**, 3-7 (2006).
49. Thoma, B. & Powner, M.W. Selective Synthesis of Lysine Peptides and the Prebiotically Plausible Synthesis of Catalytically Active Diaminopropionic Acid Peptide Nitriles in Water. *J Am Chem Soc* **145**, 3121-3130 (2023).
50. Blanco, C., Bayas, M., Yan, F. & Chen, I.A. Analysis of evolutionarily independent protein-RNA complexes yields a criterion to evaluate the relevance of prebiotic scenarios. *Current Biology* **28**, 526-537. e525 (2018).

3. A HH ribozyme mutant is a great control for the FRET experiment.

We thank the reviewer for the comment.

4. I agree that the overall reactivity rate of the ribozyme can be limited by the diffusion of substrates, as it takes ~1 hour to fully reach the center of droplets. However, in Fig 3 and S7, the product is well colocalized with the substrate in the snapshots of 5 mins and 20 mins, although the overall reaction rates from gel analysis in Fig 1 are extremely low (e.g., 0.006/min for P3). A discussion of substrate partitioning and diffusion in relation to local vs overall ribozyme reactivity and coacervate properties needs addressing, linking it to the comparison of interactions between peptide/ribozyme versus RNA/substrate if possible.

We thank the reviewer for this comment and agree that the overall rates can be attributed to a number of factors including, substrate partitioning and diffusion, ribozyme activity and coacervate properties. As suggested by the reviewer, we have included additional text into the discussion to discuss the substrate partitioning and diffusion in relation to the local vs overall ribozyme activity:

Further, this decrease in rate with increasing ribozyme concentration, could be due to molecular or mesoscopic effects. For example, on the molecular level, RNA-RNA interactions driven by high local concentrations could lead to substrate binding between two neighboring ribozymes which reduces ribozyme activity. Increased packing (increased density and concentration) of the ribozyme within the coacervate can lead to changes in the secondary structure that could lead to loss of activity. In addition, peptide-RNA interactions could block RNA substrate binding sites. Here stronger binding between the ribozyme and peptide can lead to denser droplets with increased concentrations but reduced activity. On the mesoscopic level, molecular diffusion is slower in denser droplets, which can reduce reaction rates. Therefore, rate of ribozyme cleavage could be affected by the substrate partitioning and diffusion through the coacervate as well as to changes in ribozyme activity, for instance from interactions with peptide. In addition, changes to local pH and salt can directly affect ribozyme rate as well as modulating the properties of the coacervate which could contribute to changes in ribozyme rates (See supplementary information note 5).

5. Peptide and RNA composition analysis via quantitative phase microscopy is interesting and has great potential in this community. Perhaps citing the paper by Kim et al. (*Nat Commun* **14**, 2425 (2023)) as another example of this method would be appropriate.

We thank the reviewer for this comment and have added the reference, as suggested:

53. Kim, T. et al. RNA-mediated demixing transition of low-density condensates. *Nat Commun* **14**, 2425 (2023).

6. However, on pg. 7, the last paragraph, the reasoning to use 10x higher peptide and ribozyme concentration is unclear. On a phase diagram, it is not guaranteed that a 10x higher initial concentration of peptide and ribozyme would lead to the same trend as the ones used for ribozyme reactivity experiments. This is critical for comparing their relationship in Figure 5D.

Two suggestions:

1) If the increase in 10x peptide and ribozyme concentration was to increase the number of droplets and droplet sizes, consider making a larger sample size and letting them sit for a longer time to produce more coacervates on the cover glass (assuming that coacervate droplets would get bigger via gravity and coalescence at a fixed surface area of the cover glass).

2) While I appreciate the raw data in S11, consider showing the raw data from 1x peptide and ribozyme concentration and state in the SI the limitations of this measurement. Try 2x or 5x of two extreme cases of peptides (perhaps P6 and P4) to ensure that the peptide and ribozyme concentration in the coacervate phase changes somewhat linearly with their initial concentration.

We thank the reviewer for this comment and agree that increasing the concentration by 10x would not provide coacervate systems with the same thermodynamic properties as the ones used for the ribozyme assays. For all kinetic assays, samples were prepared at 500 μM peptide and 250 μM HH_{min} . In contrast, samples were prepared at 8-fold higher peptide concentrations (4 mM) for QPI experiments to increase the typical droplet size. To ensure that droplets formed under these two conditions have the same internal concentrations, we determined the concentration of HH_{min} to combine with 4 mM of each peptide using each of the corresponding tie-lines. This was described in the following text which has now been modified to provide more clarity. From:

“To ensure that these larger droplets had the same thermodynamic properties (density) as those used for our previous experiments (RNA: peptide 250 μM : 500 μM), we used the measured tie-lines to determine the concentration of HH_{min} required for each system to coacervate with 4 mM of each peptide (Supplementary Table S4).”

To:

“To ensure that these larger droplets had the same thermodynamic properties (density) as those used for our previous experiments (RNA: peptide 250 μM : 500 μM), we used the measured tie-lines to determine the concentration of HH_{min} which, when combined with 4 mM peptide, would produce coacervates of the same properties as those generated at 8-fold lower concentrations for the kinetic measurements. (Supplementary Table S4 and supplementary information note 4).”

Table S8: concentrations of RNA and peptide used for quantitative phase imaging:

Peptide	[Peptide] (μM)	[HH_{min}] (μM)
P-1	4000	2426.7
P-2	4000	3213.2
P-3	4000	3397.4
P-4	4000	3414.4
P-5	4000	3358.7
P-6	4000	2307.5
P-7	4000	3337.0

Please also refer to Figure S13 in the supplementary information which summarizes the relationship between tie-lines, concentration and volumes.

To further address the reviewer’s comment, we now provide additional data to demonstrate that our procedure for preparing systems along a measured tie-line indeed produces coacervate droplets of modulated size but comparable composition. This has now been provided in the supplementary note with a full description and an additional figure:

Supplementary information note 4: Invariance of phase coexistence and increase in droplet size along tie-lines: To demonstrate that our procedure for preparing systems along a measured tie line indeed produces coacervate droplets of modulated size but comparable composition, we prepared P-6/ HH_{min} samples at different concentrations of peptide (3 mM, 4 mM and 5 mM) and HH_{min} (1.7 mM, 2.3 mM and 2.9 mM, respectively, (shown in Supplementary Figure 15A as points a,b,c). These average compositions were chosen to lie on the tie line that was determined for P-6 coacervates as described

previously (Supplementary table S7). We next used QPI to measure the refractive index difference, Δn , of coacervate droplets prepared at points a, b, and c. Analysis of at least 900 droplets for each sample reveals that the distribution of Δn is comparable between samples. This indicates that the concentrations in the coacervate are comparable across the samples prepared at different concentrations on the same tie-line (Supplementary Figure 13). In contrast, P-6/HH_{min} coacervates prepared at a 2:1 charge ratio and offset from the determined tie line (4 mM peptide: 2 mM HH_{min}, point d in Supplementary Figure 15A) show a shift in the refractive index distribution relative to those prepared at points a, b, and c (Supplementary Figure 15B, blue). This confirms that the latter sample (Point d) was prepared on a different tie-line compared to the other three samples, consistent with our expectation (Supplementary Figure 15A). Taken together, these measurements demonstrate that our procedure to measure tie-lines is sufficiently accurate to predict tie-line location at concentrations at least 10-fold farther into the two-phase regime.

From basic theoretical considerations, we expect the volume of coacervate phase to increase as the average composition along a tie-line and away from the dilute binodal (Supplementary Figure 13). We hypothesized that this increase in total volume of coacervate phase would be accompanied by an increase in the typical droplet size. To check this, we examined the size distribution obtained through our QPI experiments for P-6/HH_{min} coacervates prepared at points a, b, and c (Supplementary Figure 15C). Consistent with our expectation, we find that the number of larger droplets increases as the average composition increases along a fixed tie-line. Note that only coacervates with a high-quality fit (Adj. $R^2 \geq 0.98$) in the QPI analysis were included in these distributions. As we show below, this requirement excludes many smaller droplets. While the size distributions in Supplementary Figure 15C therefore do not capture all droplets in the samples, they do suffice to demonstrate the increased numbers of droplets at larger sizes for samples prepared deeper in the two-phase regime.

To illustrate the influence of droplet size on QPI analysis, we examined the refractive index measured by QPI as a function of droplet size for P-6/HH_{min} samples prepared with 4 mM peptide and 2.3 mM HH_{min} (Supplementary Figure 15D). These data are colored according to two thresholds for fit quality, with Adj. $R^2 \geq 0.94$ in grey and Adj. $R^2 \geq 0.98$ in red. For droplets with a high fit-quality, Δn shows a small spread and is independent of droplet size, as expected for droplets of equivalent composition. Notably, all droplets with high fit-quality are relatively large ($R > 1 \mu\text{m}$). Using a lower threshold on fit-quality reveals that smaller droplets show a much larger spread in Δn estimate. For coacervates smaller than $\sim 1 \mu\text{m}$ in radius, the assumptions made in our image analysis pipeline are not fulfilled, yielding poor fit quality and a wide range of refractive index estimates. We expect that this large variation at small sizes comes from optical effects, such as Mie scattering, that feature prominently in this size range and are neglected in our analysis.

For the peptide/HH_{min} coacervates studied for our kinetic assays (500 μM peptide: 250 μM RNA), we find that droplets are typically too small for robust analysis by QPI when prepared at the conditions used for kinetic measurements. Preparing samples for QPI at higher concentrations (i.e. 4 mM peptide instead of 0.5 mM) and on the measured tie-line ensures that the coacervate composition is equivalent to that present in samples prepared at lower concentrations and that a suitable number of droplets are large enough for measurement. Taken together, our results show that preparing samples at higher concentrations along the same tie-line provides droplets of equivalent refractive index difference and larger size that provide more accurate analysis of QPI data.

Figure S15: Invariance of phase coexistence along tie-lines. **A. schematic of phase-diagram** for the P-6/HH_{min} system at low concentrations. Points a, b, and c (red) denote the average composition of samples prepared along a single tie-line (black), while point d (blue) represents a sample prepared offset from that tie line. The tie line shown was calculated from two points: the dilute-phase concentrations (filled black circle) measured from a sample prepared at a system-average concentration of (0.5 mM peptide, 0.25 mM HH_{min}) and the system-average concentration (red circle). Note that the boundary between regions of homogeneous solution and phase coexistence is only approximate and was drawn by hand as a visual aid. **B. Refractive index distributions** measured by QPI for samples prepared at Points a, b, c, and d. The distributions contain measurements from N = 999, 1603, 2146 and 655 individual droplets, each fitted with an Adj. R² ≥ 0.98. The distributions measured for these three points are very similar, as expected for points on the same tie-line. As this tie-line was constructed to pass through the conditions used for kinetic measurements (red circle), the composition of coacervates formed at points a, b, and c are all equivalent to those formed at 0.5 mM peptide and 0.25 HH_{min}. In contrast, the refractive index distribution measured at point d deviates significantly from the others, confirming that point d is on a separate tie-line and corresponds to a different pair of coexisting phases. **C. Droplet size distribution measured** by QPI for samples prepared at Points a, b, and c. All droplets retained in this analysis had Adj. R² ≥ 0.98. The number of large droplets increases with increasing concentration along the tie-line. **D.** The refractive index difference measured for individual coacervate droplets from a sample prepared at Point b. For droplets with a high fit-quality (Adj. R² ≥ 0.98, red), Δn shows a small spread and is independent of droplet size, as expected for droplets of equivalent composition. Notably, all droplets with high fit-quality are relatively large (R > 1 μm). Using a lower threshold on fit-quality (Adj. R² ≥ 0.94, grey) reveals that smaller droplets show a much larger spread. Preparing samples for QPI at higher concentrations (i.e. 4 mM peptide instead of 0.5 mM) and on the measured tie-line ensures that the coacervate composition is equivalent and that a suitable number of droplets are large enough for robust measurement with QPI.

7. The interpretation of comparing the A9 site of HH ribozyme cleavages in the presence of different peptides seems valid. However, to support the assumption that RNase cannot cleave the ribozyme sites if they interact with peptides or proteins in the RNaseA footprinting assay, it is important to include some comparison of total uncut RNA between the negative control of RNaseA reaction of HH ribozyme without peptides (as in Fig S15) and the positive controls of

RNaseA reaction in the presence of peptides (as in Fig S6). Although they are not in the same gel, this should be addressed.

We agree with the reviewer and have included new data which provides a direct comparison between the RNase A reaction on HH_{min} with and without peptides. For this experiment we used 100 nM of RNase A and incubated the samples for 12 mins. The gel shows cleavage of the RNA (in the absence of peptides) at all of the restriction sites except the last site A9. In comparison, RNase A incubation with RNA with peptides shows distributed cleavage at the restriction sites. The new gel has been included into SI Figure 20.

Figure S20: **A.** Gel image showing band intensities of RNase A cleavage sites on HH_{min}. HH_{min} ladder generated by alkaline hydrolysis of HH_{min} with individual degraded nucleotides (37 visible out of 39). untreated HH_{min}, 100nM, 10nM and 1nM RNase A treated HH_{min}. We note that there is no band intensity at A4; it is possible that this site was uncleavable because of the folded region of the ribozyme. **B.** Gel image showing band intensities of RNase A cleavage sites on HH_{min}. Lanes show untreated HH_{min}; a ladder generated by alkaline hydrolysis of HH_{min} with individual degraded nucleotides (37 visible out of 39); HH_{min} with and without peptide subjected to 100 nM RNase A treatment for 12 mins. HH_{min} charge concentration was 25 mM, peptide charge concentration was 50 mM.

8. In the last paragraph on pg. 11, "sequence-specific between peptide and ribozyme interaction" could imply a specific sequence motif of the protein (in this case, peptide) interacting with a specific sequence of ribozymes coordinated by hydrogen bonding, cation-pi interaction, etc., which is not supported by the experiment here. I suggest rephrasing this.

We thank the reviewer for his suggestion and have rephrased the text from: “Despite the modest library size, our results also show that activity within the coacervate dispersions is tuned by the sequence specific interactions between the peptide and ribozyme.”

To: “Despite **only investigating seven peptides**, our results also show that activity within the coacervate dispersions is **modulated** by the **distributed charge patterning of heterogeneous peptides interacting with hammerhead ribozyme**.”

9. Additionally, it is unclear why peptides with a distribution of hydrophobic residues between positively charged amino acids lead to increased charge density. The pattern of charged amino acids seems similar (e.g., 1-1-4-2 for P2 vs. 1-3-4 for P4). Do you mean the local hydrophobicity by hydrophobic amino acid residues can change the effective charge? Please clarify this.

What we mean is that the distribution of the hydrophobic groups can change the distribution of charge leading to regions of increased charge density from the accumulation of charges together. For clarity, we have added additional text.

“This could lead to patches of increased charge density and stronger charge-charge interaction with HH_{min} as seen in P-2 and P-4 peptides **if, consequently, the number of charged residues are accumulated together** (Figure 6).”

Manuscript: NCOMMS-23-42302B

Reviewers comments:

Reviewer #1 (Remarks to the Author):

The manuscript is much improved over the first submission, and I support publication after one critical alteration have been made. I do not support publication with the current title.

Specifically, the title of the manuscript is still misleading by using the term 'RNA-peptide cooperativity tunes...'. The term 'RNA-peptide cooperativity' would be understood that ribozyme activity is enhanced by the peptides. However, the major effect of the peptides is that ribozyme activity is inhibited by 2-3 orders of magnitude, in peptide coacervates as compared to the free ribozymes. The minor effect is that some peptides are up to 15-fold less inhibitory than others. The effect of less inhibition by some peptides' is still interesting but we should not present a misleading picture. I request that the title is modified to "RNA-peptide interactions tune the ribozyme activity...".

Reviewer #2 (Remarks to the Author):

The authors have extensively answered to the comments raised by the reviewers. Because of the answers my initial concerns regarding novelty and interpretation of the results have been lifted. I feel the manuscript is now suitable for publication

Reviewer #3 (Remarks to the Author):

The authors have done substantial works to address major concerns of the reviewers, making this manuscript suitable for publication in Nature Communications.

Response to reviewers comments

We thank the reviewers for their comments and appreciate the comments from Reviewers 2 and 3 that state that the manuscript is suitable for publication in Nature Communications.

We specifically thank reviewer 1 for their comments with respect to the title of the manuscript and have now modified the title to "RNA-peptide interactions tune ribozyme activity within coacervate microdroplets".